# Epigenetic inactivation of the autophagy–lysosomal system in appendix in Parkinson's disease

Juozas Gordevicius [1,2✉], Peipei Li[1], Lee L. Marshall[1], Bryan A. Killinger[1,3], Sean Lang [1], Elizabeth Ensink [1], Nathan C. Kuhn[4], Wei Cui[5], Nazia Maroof[6], Roberta Lauria[6], Christina Rueb[6], Juliane Siebourg-Polster[7], Pierre Maliver[7], Jared Lamp[4,8], Irving Vega[4,8], Fredric P. Manfredsson [4,9], Markus Britschgi [6] & Viviane Labrie [1,10]

The gastrointestinal tract may be a site of origin for α-synuclein pathology in idiopathic Parkinson's disease (PD). Disruption of the autophagy-lysosome pathway (ALP) may contribute to α-synuclein aggregation. Here we examined epigenetic alterations in the ALP in the appendix by deep sequencing DNA methylation at 521 ALP genes. We identified aberrant methylation at 928 cytosines affecting 326 ALP genes in the appendix of individuals with PD and widespread hypermethylation that is also seen in the brain of individuals with PD. In mice, we find that DNA methylation changes at ALP genes induced by chronic gut inflammation are greatly exacerbated by α-synuclein pathology. DNA methylation changes at ALP genes induced by synucleinopathy are associated with the ALP abnormalities observed in the appendix of individuals with PD specifically involving lysosomal genes. Our work identifies epigenetic dysregulation of the ALP which may suggest a potential mechanism for accumulation of α-synuclein pathology in idiopathic PD.

[1] Center for Neurodegenerative Science, Van Andel Institute, Grand Rapids, MI, USA. [2] Institute of Biotechnology, Life Sciences Center, Vilnius University, Vilnius, Lithuania. [3] Graduate College, Rush University Medical Center, Chicago, IL, USA. [4] Department of Translational Neuroscience, College of Human Medicine, Michigan State University, Grand Rapids, MI, USA. [5] Center for Epigenetics, Van Andel Institute, Grand Rapids, MI, USA. [6] Roche Pharma Research and Early Development, Neuroscience Discovery, Roche Innovation Center, Basel, F. Hoffmann-La Roche Ltd, Basel, Switzerland. [7] Roche Pharma Research and Early Development, Pharmaceutical Sciences, Roche Innovation Center Basel, F. Hoffmann-La Roche Ltd, Basel, Switzerland. [8] Integrated Mass Spectrometry Unit, College of Human Medicine, Michigan State University, Grand Rapids, MI, USA. [9] Parkinson's Disease Research Unit, Department of Neurobiology, Barrow Neurological Institute, Phoenix, AZ, USA. [10] Division of Psychiatry and Behavioral Medicine, College of Human Medicine, Michigan State University, Grand Rapids, MI, USA. ✉email: juozas.gordevicius@vai.org

Parkinson's disease (PD) is a common neurodegenerative movement disorder affecting approximately 1% of individuals over the age of 60[1]. Clinically, PD is characterized by both motor and non-motor symptoms[2]. Non-motor symptoms, such as gastrointestinal (GI) dysfunction, may appear as many as 20 years prior to motor symptoms, during the prodromal phase of the disease[3]. Intraneuronal alpha-synuclein (α-syn) aggregates, the pathological hallmark of PD, have been detected in GI tract neurons of individuals in the prodromal stage of PD[4]. Development of pathological α-syn inclusions in neurons and loss of neuronal function is hypothesized to occur by a neuron-to-neuron propagation of presumably aggregated toxic species of α-syn[5,6]. Because α-syn aggregates are detected in the GI tract early in disease[4,7] and, in animal models of synucleinopathy, the pathology is capable of propagating to the brain via the vagus nerve[8,9], it has been suggested that the GI tract may be a point of origin for idiopathic PD[10,11]. We recently demonstrated that the appendix hosts an abundance of aggregated α-syn in healthy individuals and PD patients, with the PD appendix showing a greater amount of insoluble α-syn[12]. Hence, the appendix may be a source of α-syn pathology that contributes to the initiation of PD. However, the presence of α-syn aggregates in the healthy human appendix suggests that these aggregates in the gut are not implicitly disease-causing. Rather, there must be another molecular system whose dysfunction enables the accumulation of α-syn aggregates and their propagation to the brain.

Dysfunction of the autophagy–lysosomal pathway (ALP) may play a key role in the pathogenesis of PD. The ALP is responsible for the degradation of intracellular structures such as organelles or aggregated proteins, including α-syn[13,14]. The ALP is compromised in PD[15,16], and its disruption in neurons causes an accumulation of aggregation-prone α-syn and subsequent neurodegeneration[17–20]. Inhibition of the ALP also promotes the extracellular release of α-syn aggregates via exosomes that are capable of interneuronal travel, thereby facilitating the spread of α-syn pathology[21,22]. Many genes implicated in PD by GWAS and transcriptomic studies are linked to ALP function[23–25], including the most common monogenetic causes of PD[26,27]. Furthermore, decreasing activity of the ALP is closely related to aging, which itself is the strongest risk factor for idiopathic PD[28]. Taken together, the apparent relationships among PD, the ALP, and the development and spread of α-syn pathology suggest that disruption of the ALP in the aging appendix could be an important mechanism underlying the transfer of α-syn aggregates from the appendix to brain in PD. However, little is known about the mechanisms underlying the development of α-syn pathology in the gut or what triggers its spread to the brain.

There is evidence that epigenetic perturbation of the ALP could be responsible. DNA methylation is an epigenetic mark that is key to the regulation of gene expression, cellular function, and phenotypic outcome[29,30]. DNA methylation is known to regulate the activity of ALP genes[31,32]. Epigenetically mediated gene silencing has been shown to control autophagic flux and the supply of autophagy-related proteins[33]. Furthermore, changes in DNA methylation facilitate the decline in autophagic activity with age[34,35]. Epigenetic remodeling is known to occur with aging[30,36,37], and genes linked to neurodegeneration show accelerated DNA methylation changes with age[38]. Thus, DNA methylation changes at ALP genes could be responsible for the impaired clearance, accumulation, and spread of aggregated α-syn in PD.

In this study, we perform a comprehensive fine-mapping of DNA methylation across all genes involved in the ALP (521 genes reported in publicly available human autophagy[39] and lysosomal[40] databases as well as PD risk genes). We first determine DNA methylation abnormalities at ALP genes in the appendix of PD cases relative to controls. We then identify whether similar changes are mirrored in neurons of the prefrontal cortex, a region affected in later disease stages, and the olfactory bulb, another proposed starting point for PD[41,42]. Next, we examine the effects of age on DNA methylation at the 521 ALP genes in the appendix and prefrontal cortex neurons of PD cases and controls. We also determine the relevance of gut inflammation and α-syn pathology in the mouse cecal patch to the ALP disruption observed in PD. We find a widespread epigenetic silencing of the ALP in the PD appendix, particularly among genes implicated in lysosomal function. DNA methylation-mediated inactivation of ALP genes may contribute to the impaired clearance of pathogenic α-syn in the appendix of PD patients.

## Results

**Epigenetic dysregulation of the ALP in the PD appendix.** To determine whether epigenetic disruption of the ALP in the appendix could contribute to PD, we profiled DNA methylation at 521 genes involved in the ALP. The genes profiled were those reported on the Human Autophagy Database[39] and Human Lysosome Gene Database[40], as well as PD risk genes, which have been implicated in ALP dysregulation[23,43,44]. DNA methylation was comprehensively examined at each ALP gene and the surrounding area (± 300 kb) in the appendix of 24 PD patients and 19 controls (Supplementary Fig. 1). Fine-mapping of DNA methylation at ALP genes was performed using a targeted bisulfite sequencing approach involving a library of 67,789 unique probes, which after data preprocessing interrogated a total of 182,024 CpG methylation sites. Our analysis of differential methylation in the PD appendix was controlled for sample age, sex, postmortem interval, batch, and other sources of variation using vectors from a surrogate variable analysis. The surrogate variable analysis is a benchmark approach that corrects for cell-type heterogeneity and effectively removes other sources of variation[45–47].

DNA methylation abnormalities were detected at 928 cytosine sites affecting 326 ALP genes in the PD appendix relative to the control appendix (2.8 differentially methylated sites per affected ALP gene with 7% average methylation change; $q < 0.05$, robust linear regression; 192 and 134 genes had more hypermethylated and hypomethylated cytosines, respectively; Fig. 1a; Supplementary Data 1). Assessment of all differentially methylated cytosines demonstrated a widespread hypermethylation of ALP gene regions in the PD appendix compared to controls (odds ratio (OR), measuring the magnitude of enrichment = 1.39, $p = 1.05 \times 10^{-6}$, Fisher's exact test; Fig. 1b). Interestingly, we observed an epigenetic dysregulation of genes implicated in PD risk by large meta-analyses of genome-wide association studies (GWAS)[23]. This included a hypomethylation (associated with epigenetic activation) of α-syn (SNCA) ($q < 0.05$, robust linear regression; Fig. 1a). There was also a hypermethylation (associated with epigenetic silencing) of Toll-like receptor 9 (TLR9), which controls host immune responses to pathogens via the lysosome[48]; a hypermethylation of GTP cyclohydrolase 1 (GCH1), which regulates lysosomal function, dopamine synthesis, and infection[49,50]; and a hypermethylation of glycoprotein non-metastatic melanoma protein B (GPNMB), which attenuates inflammation[51] ($q < 0.05$, robust linear regression; Fig. 1a).

Since we profiled ALP genes and extended genomic areas that include enhancers, CpG islands, and other regulatory elements, we sought to determine the types of genomic regions with the most prominent epigenetic dysregulation. Promoters and CpG islands within ALP gene areas were most epigenetically perturbed in the PD appendix (OR = 3.55, $p = 1.71 \times 10^{-45}$ and OR = 5.79,

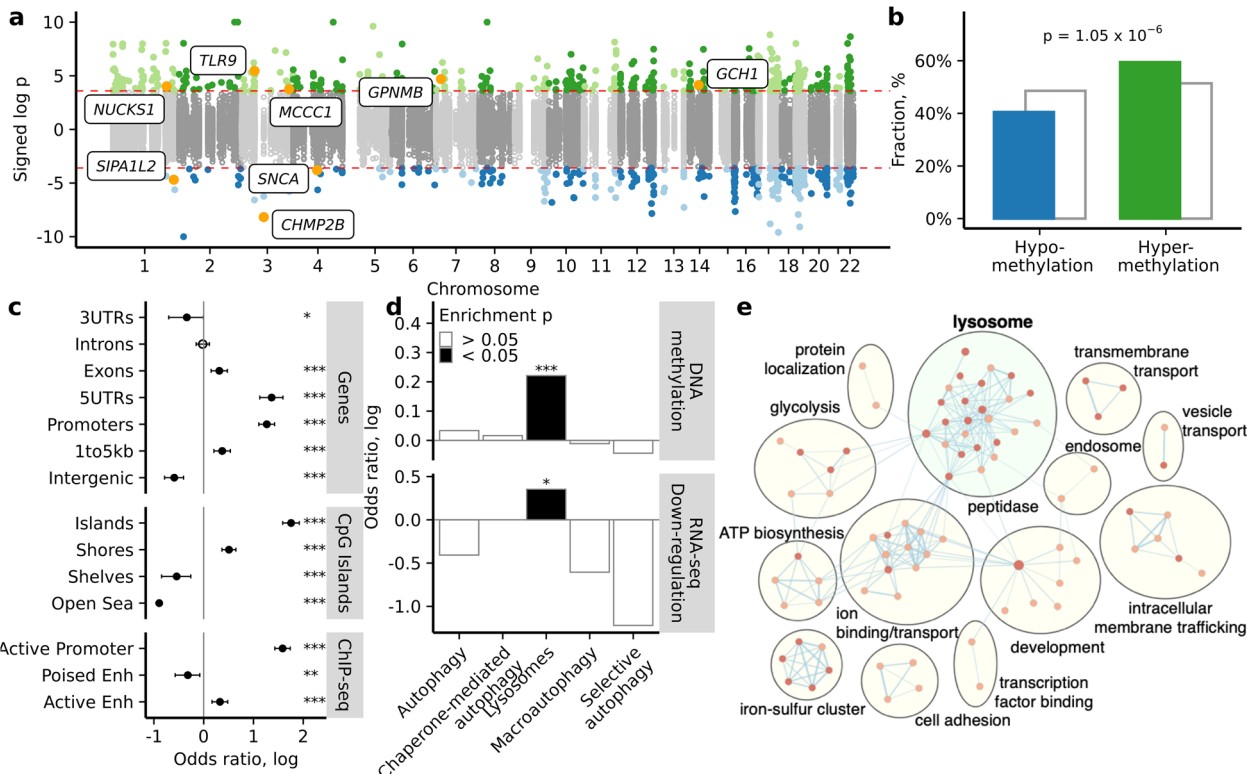

**Fig. 1 Epigenetic dysregulation of the ALP in the PD appendix, particularly at lysosomal genes.** DNA methylation was fine-mapped at 521 genes in the ALP in the appendix of PD patients and controls ($n = 24$ and 19 individuals, respectively). **a** Manhattan plot demonstrating differential methylation in PD appendix. There were 326 genes in the ALP exhibiting differential methylation in PD relative to controls (928 cytosine sites at $q < 0.05$, robust linear regression). Highlighted are genes implicated in both our study and PD GWAS[23]. Signed log p refers to the significance of differentially methylated cytosines, with sign corresponding to the direction of DNA methylation change (hypermethylation, green or hypomethylated, blue). **b** Fraction of significantly hypomethylated (blue) and hypermethylated (green) sites in the PD appendix, as compared to all sites (gray). *P*-value represents two-sided Fisher's exact test for enrichment of hypermethylated sites among significant versus non-significant cytosines. **c** Genomic elements enriched with significant differentially methylated sites in the PD appendix. Filled circles represent *$p < 0.05$, **$p < 0.01$, and ***$p < 0.001$, two-sided Fisher's exact test on $n = 182,024$ cytosines comparing the odds of observing significant cytosines within versus outside of the genomic element. Error bars indicate 95% confidence intervals. **d** ALP pathways enriched with differential methylation and differential gene expression. Filled bars represent *$p < 0.05$ and ***$p < 0.001$, two-sided Fisher's exact test comparing the odds of observing significant cytosines at genes belonging to the selected pathway versus the other targeted ALP genes. **e** Pathways affected by transcriptomic changes in the PD appendix. Pathways for the differentially expressed genes in the PD appendix were obtained using DAVID and visualized with EnrichmentMap and AutoAnnotate in Cytoscape (v3.7.1). Node colors represent normalized enrichment score, where darker red signifies greater enrichment. Actual p values are reported in Supplementary Data 21.

$p = 3.20 \times 10^{-70}$, respectively; Fisher's exact test; Fig. 1c). Furthermore, we identified active enhancers, poised enhancers, and active promoters in the healthy appendix by chromatin immunoprecipitation of histone H3K27ac and H3K4me1 marks ($n = 3$ individuals per mark). In ALP gene regions, DNA methylation abnormalities strongly affected active promoters and to a lesser extent affected active enhancers (OR = 4.88, $p = 1.24 \times 10^{-69}$ and OR = 1.39, $p = 5.98 \times 10^{-5}$; Fisher's exact test; Fig. 1c). We also confirmed that promoters at ALP genes had the strongest enrichment in hypermethylated cytosines (OR = 9.78, $p = 7.45 \times 10^{-111}$; Fisher's exact test; Supplementary Fig. 2). Hence, there is an epigenetic silencing of ALP gene areas that primarily impacts promoter activity.

We next sought to determine the ALP pathways that exhibited the most prominent epigenetic dysregulation in the PD appendix. We stratified the profiled ALP genes according to their corresponding pathways (gene ontology classifications) and found that lysosomal genes were enriched with hypermethylated cytosines in the PD appendix (OR = 1.33, $p = 9.80 \times 10^{-4}$; Fisher's exact test; Fig. 1d; Supplementary Fig. 3). Conversely, we found enrichment of hypomethylated cytosines among genes involved in autophagy (OR = 1.25; $p = 0.04$; Supplementary

Fig. 3). Furthermore, there was an enrichment of DNA methylation alterations within 20 kb of lysosomal gene start sites in the PD appendix (OR = 2.12, $p = 3.67 \times 10^{-8}$; Fisher's exact test; Supplementary Fig. 4). DNA methylation changes in the PD appendix were specific to lysosomal genes, as no enrichment was observed for non-lysosomal genes profiled in this study (Supplementary Fig. 4). Hence, DNA methylation abnormalities in the ALP of the PD appendix particularly affect lysosomal genes.

To determine whether differential methylation of ALP genes corresponded to functionally relevant changes in gene transcript levels, we performed a transcriptomic analysis of the PD and control appendix by RNA sequencing (RNA-seq; $n = 12$ PD and 16 controls; Supplementary Fig. 1). Differentially expressed genes in the PD appendix were identified, after adjusting for sample age, sex, postmortem interval, RIN, and other sources of variation (including cell-type heterogeneity) using vectors from a surrogate variable analysis. We identified 246 genes with differential transcript levels in the PD appendix ($q < 0.05$, robust linear regression; Supplementary Data 2). There was no dominant direction of change transcriptome-wide, but ALP gene transcripts showed an overall downregulation (OR = 2.95, $p = 0.002$, Fisher's

exact test). Next, we found that changes in averaged DNA methylation at ALP genes were significantly associated with changes in corresponding ALP gene transcript levels in the appendix of PD patients relative to controls ($r = 0.21$, $p = 1.58 \times 10^{-5}$, weighted Pearson correlation of fold changes). Consistent with our DNA methylation analysis, we found that among ALP genes, there was a specific downregulation of genes in the lysosomal pathway in the PD appendix ($OR = 1.42$, $p = 0.02$; Fisher's exact test; Fig. 1d). Transcriptome-wide pathway analysis indicated that there was a significant enrichment of lysosomal abnormalities in the PD appendix ($q < 0.05$; DAVID 6.8; Fig. 1e). Hence, epigenetic changes at ALP genes are associated with transcriptomic abnormalities in the PD appendix that involves a prominent dysregulation of lysosomal genes.

**Epigenetic dysregulation of the ALP is recapitulated in the PD brain.** We investigated the extent to which DNA methylation abnormalities affecting the PD appendix are observed in the brain. In addition to the gut, the olfactory bulb has also been proposed as an initiation site for PD[41,42]. We fine-mapped DNA methylation in the olfactory bulb of PD patients and controls ($n = 9$ PD and 14 controls) at the same 521 ALP genes and surrounding genomic area ($\pm$ 300 kb) as for the appendix study and using the same bisulfite padlock probe approach (Supplementary Fig. 1). In olfactory bulb tissue, we interrogated DNA methylation at 143,553 CpG sites at ALP genes. We also profiled DNA methylation at ALP genes in neurons isolated from the prefrontal cortex of PD patients and controls ($n = 52$ PD patients and 42 controls; Supplementary Fig. 1). In isolated neurons, DNA methylation occurs at both CpG and CpH (i.e., CpA, CpT, CpC) locations[36], and thus in neurons of the prefrontal cortex, we investigated a total of 130,733 CpG and 696,665 CpH sites at ALP genes. Our analysis identified DNA methylation differences in the PD brain after adjusting for sample age, sex, postmortem interval, and, for prefrontal cortex neurons, the proportion of glutamate neurons to GABAergic neurons[52].

In the olfactory bulb, we identified 1,142 differentially methylated cytosines affecting 353 genes in PD relative to controls (3.2 differentially methylated sites per affected ALP gene with 18% average methylation change; $q < 0.05$, robust linear regression; Fig. 2a; Supplementary Data 3). As observed in the PD appendix, there was a significant hypermethylation of ALP genes in the PD olfactory bulb ($OR = 1.18$, $p = 0.006$; Fisher's exact test; Fig. 2b, OFB). In prefrontal cortex neurons, we observed 70 differentially methylated sites affecting 58 genes in PD (1.2 differentially methylated sites per affected ALP gene with average methylation change 8% in CpG and 6% in CpH sites; $q < 0.05$, robust linear regression; Fig. 2a; Supplementary Data 4), which again were mostly hypermethylated ($OR = 2.41$, $p = 7.13 \times 10^{-4}$, Fisher's exact test; Fig. 2b, PFC). Specifically, we observed a strong hypermethylation trend among CT and CA dinucleotides ($OR = 5.01$, $p = 0.005$, $OR = 13.41$, $p = 0.0004$, respectively; Fisher's exact test). We found a significant overlap between the ALP genes epigenetically disrupted in the PD appendix and those altered in the PD olfactory bulb and prefrontal cortex neurons ($n = 264$ overlapping genes, $OR = 3.97$, $p = 2.39 \times 10^{-11}$ and $n = 49$ overlapping genes, $OR = 3.20$, $p = 0.001$, respectively, Fisher's exact test). Similar disruptions of PD risk genes to those identified in the PD appendix included aberrant *TLR9* methylation, *GPNMB* hypermethylation, and *SNCA* hypomethylation in the analyzed PD brain regions (Supplementary Figs. 5 and S6). We also examined prefrontal cortex neurons of PD patients that did not have Lewy pathology (Braak stage 3–4) and identified 110 differentially methylated cytosines affecting 87 genes relative to controls ($q < 0.05$, robust linear regression; $n = 20$ PD Braak stage

3–4, 42 controls; Supplementary Data 5). These changes significantly overlapped with those identified with the full cohort ($OR = 349.36$, $p = 1.24 \times 10^{-7}$, Fisher's exact test), suggesting that many of the DNA methylation changes observed precede the onset of Lewy pathology.

We further confirmed our findings in the PD brain using a separate cohort of neurons isolated from the prefrontal cortex. Since this was a smaller cohort of 13 PD patients and 15 controls, we limited our analysis to CpG sites ($n = 110,397$; Supplementary Fig. 1). There were 1,131 differentially methylated cytosines affecting 341 ALP genes in neurons of PD patients relative to those of controls (3.3 differentially methylated sites per affected ALP gene with 13% average methylation change; $q < 0.05$, robust linear regression; Fig. 2a; Supplementary Data 6). This cohort replicated the significant hypermethylation of ALP genes in PD neurons ($OR = 1.32$, $p = 5.08 \times 10^{-6}$; Fisher's exact test; Fig. 2b, PFCII). There was also a robust overlap between differentially methylated ALP genes in this cohort of prefrontal cortex neurons and the appendix of PD patients ($OR = 4.21$, $p = 2.33 \times 10^{-12}$; Fisher's exact test). Overall, the epigenetic misregulation of the ALP in the PD appendix is recapitulated in the PD brain.

To further investigate the extent to which epigenetic alterations in the PD brain are similar to those in the PD appendix, we investigated different types of genomic elements. We determined whether DNA methylation changes within genomic elements were consistent between the PD brain and PD appendix. In PD patients, DNA methylation differences found in the appendix, prefrontal cortex neurons, and the olfactory bulb were largely concordant across genomic elements, especially within introns and intergenic regions ($p < 0.001$, Fisher's exact test; Fig. 2c, Fig. S19). Regions most distal to CpG islands (referred to as "open sea") also exhibited concordant epigenetic changes in the PD appendix and brain ($p < 0.001$, Fisher's exact test; Fig. 2c). Next, we identified active enhancers, poised enhancers, and active promoters that had similar genomic coordinates ($\pm 1$ kb) in the human brain and appendix (based on PsychEncode data of prefrontal cortex neurons ($n = 9$ individuals) and the ChIP-seq data in the appendix ($n = 3$ individuals) described above). We found that the PD appendix and brain had a strong concordant epigenetic dysregulation of active and poised enhancers and active promoters ($p < 0.05$, Fisher's exact test; Fig. 2c). In sum, our analyses of DNA methylation demonstrate that the PD appendix and brain show a common epigenetic dysregulation of the ALP that is shared across several types of genomic elements.

We compared the differentially methylated sites at ALP genes identified in the PD prefrontal cortex neurons to those identified in a genome-wide analysis[53]. The genome-wide dataset used profiled DNA methylation at all brain enhancers and promoters, examining 904,511 methylated cytosine sites in prefrontal cortex neurons of 57 PD patients and 48 controls. We found that there was a significant overlap of the top loci identified in our ALP dataset with those of the genome-wide dataset ($OR = 4.49$, $p = 1.23 \times 10^{-6}$; top 5000 loci examined in each dataset). Thus, comparison with a genome-wide analysis supports an epigenetic disruption of ALP gene elements in PD brain neurons.

**ALP genes and proteins disrupted in the PD appendix and brain.** We determined specific ALP genes that were the most consistently epigenetically disrupted across the PD appendix and brain. The ALP genes were ranked based on their enrichment for significant DNA methylation changes in the PD appendix, olfactory bulb, and prefrontal cortex neurons, relative to controls. ALP genes exhibiting the strongest epigenetic misregulation across PD tissues were then determined using the cumulative

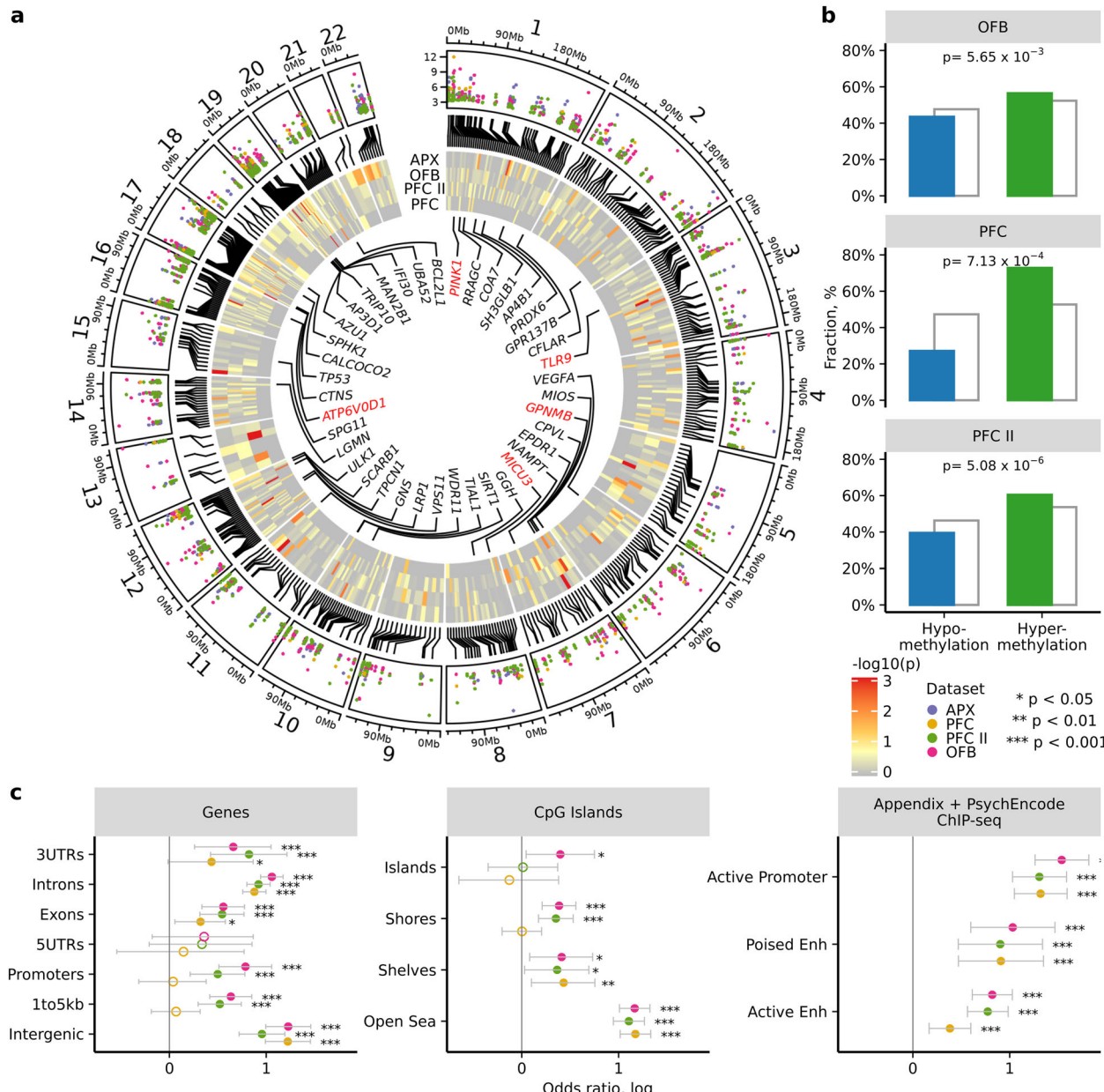

**Fig. 2 DNA methylation alterations at ALP genes in the PD appendix are mirrored in PD brain.** Fine-mapping of DNA methylation changes at ALP genes was performed in the olfactory bulb (OFB, $n = 9$ PD and 14 controls) and prefrontal cortex neurons ($n = 52$ PD and 42 controls in primary cohort (PFC); 13 PD and 15 controls in replication cohort (PFCII)). Concordance between DNA methylation changes in the PD brain and PD appendix (APX) is shown. **a** Distribution of differentially methylated sites at ALP genes in the appendix, olfactory bulb, and prefrontal neurons of PD patients, relative to controls. Colored dots represent log $p$-values for each dataset. Rectangles represent ALP genes with fill color corresponding to −log $p$-value of gene enrichment with differentially methylated cytosines (determined by two-sided Fisher's exact test). The top 40 ALP genes that are consistently differentially methylated across the PD appendix, olfactory bulb, and prefrontal cortex neurons are highlighted (determined by robust rank aggregation). Known PD risk genes and genes identified in PD GWAS[23] labeled in red. **b** Fraction of significantly hypomethylated and hypermethylated sites in PD brain cohorts (blue and green, respectively), in comparison to all sites (gray) investigated. $P$-value represents two-sided Fisher's exact test for enrichment of hypermethylated sites among significant versus non-significant cytosines. **c** Genomic elements exhibiting similar changes in DNA methylation in the PD brain and PD appendix. For each category, the overlap of genomic elements with differentially methylated cytosines between PD appendix and brain datasets was determined. Poised and active enhancers, and active promoters were identified using appendix ChIP-seq ($n = 3$ individuals each for H3K27ac and H3K4me1) and prefrontal cortex ($n = 9$ individuals, PsychENCODE data). Filled circles represent *$p < 0.05$, **$p < 0.01$, and ***$p < 0.001$, two-sided Fisher's exact test examining overlap with PD appendix. Error bars indicate 95% confidence intervals. Actual p values are reported in Supplementary Data 21.

ranked score in the robust rank aggregation algorithm[54] (Fig. 2a; Supplementary Data 7). In this analysis, genes are ranked according to their enrichment in differentially methylated cytosines for each tissue, followed by a rank integration approach[54] to identify consistently disrupted ALP genes across the PD appendix

and brain tissues. The top 40 genes included *TLR9*; *GPNMB*; PTEN-induced kinase 1 (*PINK1*), a well-established familial PD gene[55]; Unc-51 like autophagy activating kinase 1 (*ULK1*), the initiating enzyme in autophagy[56]; and sphingosine kinase 1 (*SPHK1*), nicotinamide phosphoribosyltransferase (*NAMPT*),

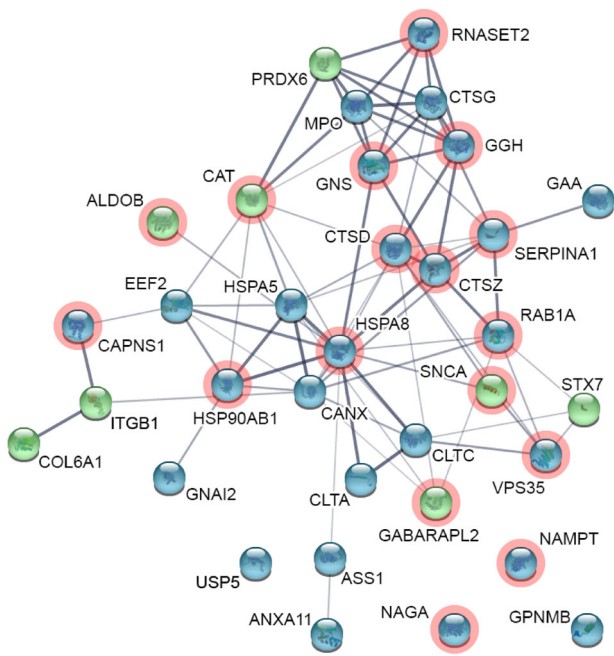

**Fig. 3 Epigenetically dysregulated ALP genes exhibit differential proteins levels in the PD appendix and brain.** Proteomic analysis of the appendix and brain of PD patients and controls ($n = 3$ PD and 3 controls per tissue). STRING-db representation of ALP genes showing DNA methylation changes and altered protein abundances in the PD appendix. Node color indicates increase (green) or decrease (blue) in protein abundance in the PD appendix. Genes marked in red have altered protein abundances in the PD prefrontal cortex. Thickness of lines connecting nodes corresponds to the strength of data support for interactions between proteins.

and sirtuin 1 (*SIRT1*), which impact neurodegeneration and aging processes[57,58].

To further investigate the functional consequences of epigenetic dysregulation of the ALP, we performed an unbiased, quantitative proteome analysis of the appendix and brain of PD patients and controls ($n = 3$ PD and 3 controls per tissue; Supplementary Fig. 1). Overall, 1,341 and 1,743 proteins were quantified in the appendix and prefrontal cortex, respectively (Supplementary Data 8 and 9). Protein abundance changes in the PD appendix and brain were significantly correlated ($r = 0.26$, $p < 2.2 \times 10^{-16}$; Pearson correlation). Among the altered proteins were 34 ALP proteins encoded by genes with an epigenetic dysregulation in the PD appendix (Fig. 3; Supplementary Table 1). These ALP proteins showed a marked concordance in fold change in the PD appendix and prefrontal cortex ($r = 0.60$, $p = 0.008$; Pearson correlation), signifying that the PD appendix and brain exhibit similar abnormalities in ALP protein levels. The appendix of PD patients had increased α-syn protein levels (SNCA), along with decreases of GPNMB, NAMPT, vacuolar protein sorting 35 ortholog (VPS35) and heat shock 70 kDa protein 8 (HSPA8/HSC70), which are all linked to PD risk and affect neuronal survival[23,57,59–62]. Consistently, several of these proteins had abnormal levels in the PD prefrontal cortex (Fig. 3), including an upregulation of α-syn (SNCA) and a downregulation of NAMPT, HSPA8, and VPS35 (Supplementary Table 1). Overall, we found that the majority of ALP proteins altered in the PD appendix and prefrontal cortex are also epigenetically altered in their respective tissues (76%, 70%, 85.7% of ALP proteins altered in the PD appendix, prefrontal cortex, or both tissues, respectively, were epigenetically altered). Consequently, there is concordance between the epigenetic and proteome-wide analysis, which together point to a prominent decrease in ALP

function and an overabundance of α-syn in the PD appendix and brain.

In addition, we performed a replication study using a tandem mass tag (TMT) quantitative proteomic analysis to further validate our findings in a lysosome-enriched fraction of appendix tissue ($n = 5$ control, 5 PD). We identified 2084 proteins including 132 (27%) of the proteins encoded by ALP genes interrogated in this study (Supplementary data 10). We found 175 differentially abundant proteins (FDR $q < 0.05$, robust linear regression), including 7 ALP proteins encoded by genes that were also epigenetically dysregulated in the human appendix. The overlap of differentially abundant proteins identified in both datasets was higher than expected by chance (OR = 1.88, $p = 0.04$; Fisher's exact test), and the proteins identified as differentially abundant in label-free approach had a concordant fold change in the replication dataset ($r = 0.40$, $p = 5.99 \times 10^{-25}$; weighted Pearson correlation). Upregulation of SNCA and downregulation of NAMPT, HSPA8, GPNMB and VPS35 was confirmed by fold change estimates in the replicate data. Overall, this analysis confirms the validity of our previous findings.

**Epigenetic age effects indicate disrupted ALP in the PD appendix and brain.** Advanced age is universally considered the greatest risk factor for PD. Why aging exerts such strong risk for PD is not well understood, particularly at a mechanistic level. We investigated the epigenetic age dynamics of the ALP in the healthy human appendix, and determined its relevance to ALP changes occurring in PD. For this analysis, we examined DNA methylation at the 521 ALP genes in 51 healthy (ages 18–92) and 24 PD (ages 62–91) appendix samples and profiled a total of 181,151 CpG sites (Supplementary Fig. 1). As a comparison, we also examined age-related changes in DNA methylation in two brain sample datasets. In prefrontal cortex neurons of 42 control and 52 PD individuals, ages 55–93, we profiled a total 130,733 CpG and 696,665 CpH sites. In olfactory bulb tissue of 14 control and 9 PD individuals, ages 53–92 we profiled a total of 143,553 CpG sites.

In the healthy appendix, there were 285 cytosine sites at 170 ALP genes showing chronological age-related changes in DNA methylation (1.7 differentially methylated sites per affected ALP gene with 0.44% average methylation change per year; $q < 0.05$, robust linear regression after adjusting for sex, postmortem interval, batch, and other sources of variation by surrogate variables factor analysis; Supplementary Data 11). Though DNA methylation in the aging appendix did not show an overall directional shift at ALP genes (Fig. 4a), there was a prominent hypermethylation of poised enhancers, promoters, and CpG islands and shores with age ($p < 0.05$, Fisher's exact test; Supplementary Fig. 7). In prefrontal cortex neurons of controls 304 sites affecting 200 ALP genes changed methylation with age (1.5 differentially methylated sites per affected ALP gene with 0.42% average methylation change per year; $q < 0.05$, robust linear regression adjusting for sex, postmortem interval, and neuron subtype proportion; Supplementary Data 12). ALP genes in prefrontal cortex neurons largely became hypermethylated with increasing age (OR = 1.65, $p = 2.55 \times 10^{-5}$, Fisher's exact test; Fig. 4a). Interestingly, age-associated genes in appendix and prefrontal cortex neurons overlapped (OR = 2.89, $p = 0.5 \times 10^{-7}$; Fisher's exact test) and their absolute age effect magnitudes correlated ($r = 0.31$, $p = 0.002$; Spearman correlation). In olfactory bulb of controls there were 853 sites epigenetically changing by age affecting 325 ALP genes (2.6 differentially methylated sites per affected ALP gene with 0.75% average methylation change per year; $q < 0.05$, robust linear regression adjusting for sex and post mortem interval; Supplementary Data 13). No dominant

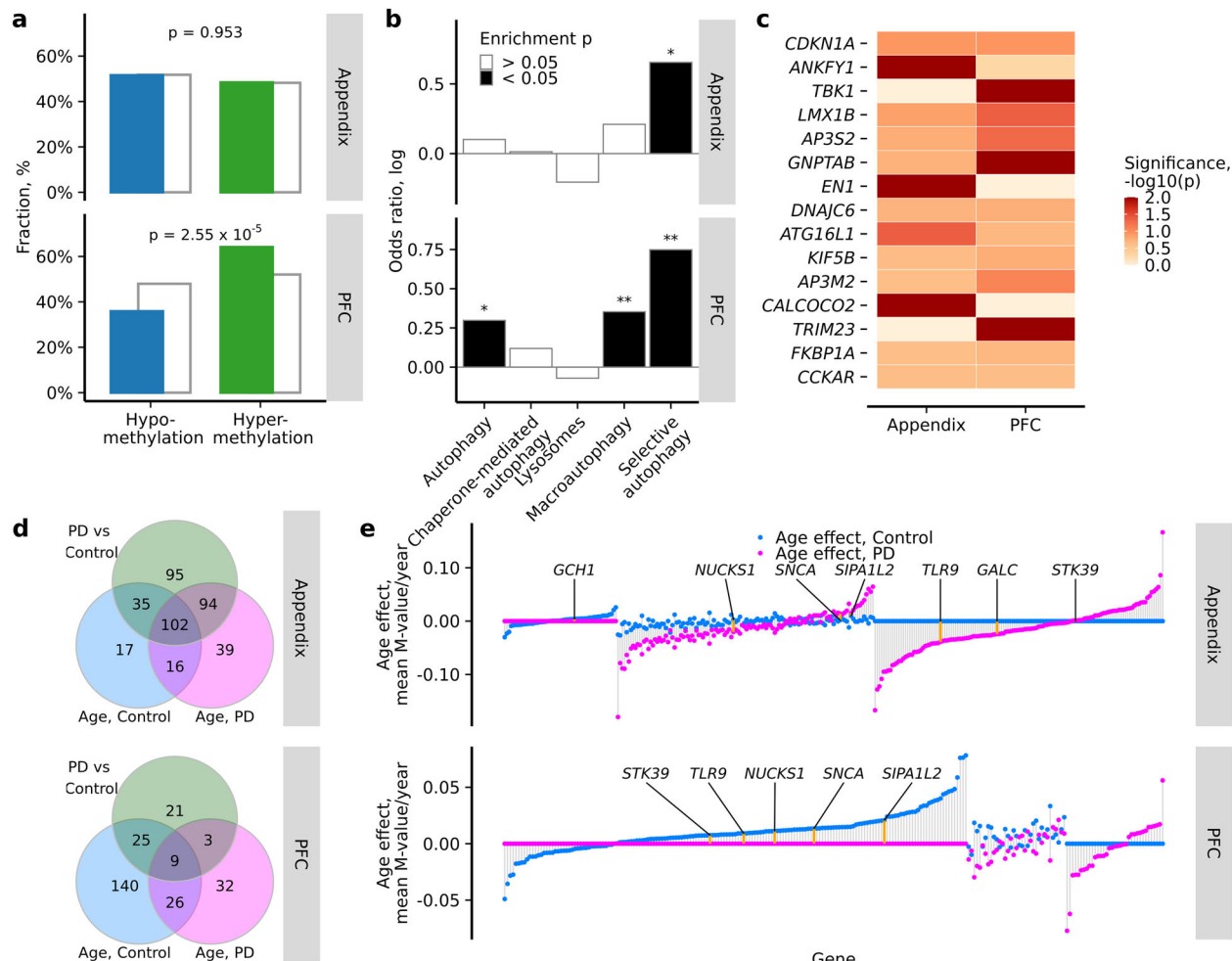

**Fig. 4 Age-related changes in DNA methylation affecting ALP genes in the appendix and brain. a** Distribution of significantly hypomethylated (blue) and hypermethylated (green) cytosine sites in the healthy aging appendix and prefrontal cortex neurons, compared to all cytosine sites investigated (gray). *P*-values represent two-sided Fisher's exact test for enrichment of hypermethylated sites among significant versus non-significant cytosines. **b** ALP pathways altered in the healthy aging appendix and prefrontal cortex neurons. Pathway enrichment for ALP genes with significant epigenetic aging changes in the healthy appendix (*n* = 51) and prefrontal cortex neurons (*n* = 42). Filled bars represent \**p* < 0.05 and \*\**p* < 0.01, two-sided Fisher's exact test comparing the odds of observing significant cytosines at genes belonging to selected pathway versus the other targeted ALP genes. **c** Top 10 genes most consistently altered in the healthy aging appendix and neurons of the prefrontal cortex (determined by robust rank aggregation). *P*-value is ALP gene enrichment in differentially methylated cytosines determined by two-sided Fisher's exact test. **d** Overlap of the ALP genes affected by mean methylation differences between control and PD samples or age-related methylation changes in healthy control or PD samples in the appendix (upper panel) and prefrontal cortex neurons (lower panel). A gene is affected if it has a significantly differentially modified cytosine (FDR *q* < 0.05). **e** Epigenetic aging rates of ALP genes in the appendix (upper panel) and prefrontal cortex neurons (lower panel) of healthy controls (blue points) and PD cases (purple points). For each ALP gene, the aging rate was computed as mean aging rate of cytosines pertaining to that gene weighted by log transformed *p* value of aging model fit. The aging rate was set to zero for ALP genes that did not have any aging cytosines with FDR *q* < 0.05. Genes were sorted by the sum of their aging rates in control and PD samples. PD-related genes implicated in GWAS studies are marked by orange segments and labeled. Appendix *n* = 51 controls, 24 PD; prefrontal cortex *n* = 42 controls, 52 PD. Actual p values are reported in Supplementary Data 21.

direction of methylation change could be established (OR = 1.06, *p* = 0.41; Fisher's exact test).

ALP pathway analysis revealed an overrepresentation of age-related hypermethylation of cytosines among selective autophagy genes in both the appendix and prefrontal cortex neurons, and among macroautophagy genes in the prefrontal cortex (OR = 2.42, *p* = 0.02; OR = 1.39, *p* = 0.10 in appendix and OR = 2.06, *p* = 0.04; OR = 1.42, *p* = 0.04 in prefrontal cortex neurons, for selective autophagy and macroautophagy, respectively, Fisher's exact test; Fig. 4b; Supplementary Fig. 8). There was also a strong correlation in the enrichment of all tested pathways between appendix and prefrontal cortex neurons (*r* = 0.97, *p* = 0.006, Pearson's correlation). Together, this suggests that the machinery for degrading aggregated proteins and invading

pathogens may be compromised in the appendix and brain with advanced age.

We determined the ALP genes most epigenetically affected by age in the healthy appendix and brain. We ranked ALP genes according to their enrichment in significant DNA methylation changes with chronological age in the appendix and prefrontal cortex, and then consolidated the rankings (using the robust rank aggregation algorithm, as above; Fig. 4c; Supplementary Data 14). The top genes exhibiting convergent DNA methylation abnormalities in the appendix and brain included TANK-binding kinase 1 (*TBK1*), which plays a key role in initiating selective autophagy[63]; engrailed homeobox 1 (*EN1*) and LIM homeobox transcription factor 1 beta (*LMX1B*), which are neuroprotective and affect the functioning of dopamine neurons[64–66]; calcium-

binding and coiled-coil domain-containing protein 2 (CAL-COCO2), which has been implicated as a key regulator of inflammation in Crohn's disease[67]; and auxilin (DNAJC6), mutations of which are associated with early-onset PD[68] (Fig. 4c). Hence, the ALP genes most affected by age in the human appendix and cortical neurons are related to selective autophagy, inflammation, and physiological neuronal activity and survival.

We then investigated whether the ALP genes that exhibit age-related changes in healthy controls also differ between PD cases and controls. ALP genes affected by significant age-related DNA methylation changes observed in the control group were compared to the ALP genes affected by significant DNA methylation differences between control and PD patients, examining the appendix and prefrontal cortex neurons separately. We found a significant overlap of age and PD affected ALP genes in the appendix and prefrontal cortex neurons (OR = 3.19, $p < 2 \times 10^{-16}$ and OR = 2.31, $p = 0.004$, respectively, Fisher's exact test; Fig. 4d). Absolute cytosine methylation age effect rates were significantly positively correlated to absolute methylation changes manifesting in PD, for the appendix, olfactory bulb, and prefrontal cortex neurons ($r > 0.70$, $p < 10^{-15}$; Pearson correlation; Supplementary Fig. 9). Hence, the ALP genes that are altered with age are also vulnerable to PD disease processes.

Next, we compared magnitudes of epigenetic age effects among ALP genes in PD patients to that in healthy controls. In the appendix of PD samples there were 561 cytosines changing with age and involving 251 ALP genes. In prefrontal cortex neurons of PD samples there were 77 cytosines changing with age and involving 70 ALP genes. For each ALP gene, we computed the mean magnitude of the age effect weighted by the corresponding $p$ value of each cytosine pertaining to that gene and set the rate to zero if there were no cytosines significantly changing with age (Fig. 4e). In the appendix, we found 118 ALP genes changing with age among both healthy control and PD samples, more than expected by chance (OR = 3.43, $p < 2.2 \times 10^{-16}$; Fisher's exact test), and among them there was an agreement of age effect direction (OR = 2.21, $p = 0.05$; Fisher's exact test). The absolute age effect magnitudes of ALP genes were higher in PD than in control samples ($p = 1.58 \times 10^{-15}$; paired $t$-test), suggesting accelerated methylation change with age. In prefrontal cortex neurons, 35 ALP genes were affected with age in both healthy control and PD samples (OR = 1.58, $p = 0.09$; Fisher's exact test); there was no significant agreement of age effect direction nor magnitude (OR = 0.31, $p = 0.55$ and $p = 0.35$; Fisher's exact test and paired $t$-test, respectively).

To ascertain that our findings are not influenced by the wide age range of appendix controls, we repeated the same analysis using only appendix samples older than 62 years. We confirmed hypermethylation of promoters and again found absolute age effects to be stronger among PD samples (Supplementary Data 20, Supplementary Fig. 18). Taken together, this suggests that in patients, PD disease processes are associated with disrupted normal age-related changes in ALP function.

**α-syn accumulation exacerbates responses to gut inflammation in vivo**. Gut inflammation is thought to play a role in the development of PD, and inflammatory bowel diseases, including ulcerative colitis, are linked to increased PD risk[69,70]. Therefore, we explored the effects of gut inflammation on epigenetic regulation of the ALP in mice. DNA methylation changes at ALP genes were examined in the mouse equivalent of the appendix, the cecal patch. To induce gut inflammation, we performed a chronic dextran sodium sulfate (DSS) administration paradigm, a widely used model of ulcerative colitis[71]. Wild-type mice and transgenic mice that overexpress human α-syn with the hemizygote A30P mutation (A30P α-syn mice), a PD-relevant model of synucleinopathy, were chronically exposed to increasing concentrations of DSS over four weeks, followed by a recovery period of four weeks (Fig. 5a). By the end of the recovery period, the colon of the mice had returned to a histologically normal appearance, as determined by pathological analysis of leukocyte infiltration and tissue integrity (Supplementary Fig. 11). We and others have previously reported that wild-type and A30P α-syn mutant mice develop α-syn aggregates in enteric neurons triggered by this chronic DSS colitis paradigm[72,73], though processes beyond α-syn aggregation (i.e., synaptic transmission, immunological responses) are also altered in these models[71,74].

We fine-mapped DNA methylation changes at 571 ALP genes in the mouse cecal patch. We first identified the mouse homologs for the human ALP genes profiled above and examined DNA methylation across the entire gene and neighboring genomic area (±50 kb), using 87,605 bisulfite padlock probes. In total, we profiled 240,366 CpG sites at ALP genes in the cecal patch of wild-type and A30P α-syn mice that experienced DSS colitis or remained on normal water ($n = 9$–11 mice/group; Supplementary Fig. 1).

DSS colitis induced DNA hypomethylation at ALP genes that were far more severe in the PD α-syn overexpression model. In wild-type mice, previous DSS colitis resulted in 1,104 differentially methylated cytosines affecting 397 ALP genes, relative to mice that did not experience colitis (2.8 differentially methylated sites per affected ALP gene with 8.4% average methylation change; $q < 0.05$, robust linear regression; Supplementary Fig. 12a; Supplementary Data 15). In A30P α-syn mice, there were 1378 differentially methylated cytosines affecting 408 ALP genes (3.4 differentially methylated sites per affected ALP gene with 9% average methylation change; $q < 0.05$, robust linear regression; Supplementary Fig. 12b). DSS colitis in both wild-type and A30P α-syn mice induced ALP hypomethylation (OR = 1.54, $p = 2.55 \times 10^{-12}$ and OR = 2.76, $p = 3.08 \times 10^{-55}$, respectively, Fisher's exact test; Supplementary Fig. 12a, b). However, DNA methylation changes mediated by DSS colitis were greatly exacerbated in the A30P α-syn mice relative to wild-type mice (OR = 1.70, $p = 7.48 \times 10^{-34}$, Fisher's exact test). The A30P α-syn mutation alone did not induce significant ALP hypomethylation (Supplementary Fig. 12c). These results suggest that α-syn overexpression strongly sensitizes mice to an intestinal inflammatory insult.

We observed that DSS colitis resulted in the epigenetic disruption of numerous ALP genes associated with PD risk in GWAS (Supplementary Fig. 12a and Supplementary Fig. 12b). Notably, we observed significant hypomethylation of the Snca and Gpnmb genes in the A30P mice given DSS, but not in the wild-type mice given DSS ($q < 0.05$, robust linear regression; Supplementary Fig. 12a and Supplementary Fig. 12b); these genes were both epigenetically perturbed in the PD appendix (Fig. 1a).

We then sought to determine whether α-syn aggregation in enteric neurons was a key contributor to epigenetic dysregulation of the ALP induced by intestinal inflammation. In this experiment, we injected into the cecal patch of wild-type C57BL/6J mice a recombinant adeno-associated virus (rAAV) vector that overexpressed either human α-syn or GFP as control[74] (Fig. 5a) using direct subserosal delivery to the ENS. This targeted approach results in transduction of neurons per se, with no off-target transduction of support cells such as muscle or glia[74,75]. At one month post-injection, there was an abundance of aggregated α-syn positive for phospho-Ser129 in enteric plexuses of the mouse cecal patch (Supplementary Fig. 13). No phospho-Ser129 immunoreactivity was observed in mice treated with the control vector. We measured the effects of α-syn aggregation on DNA methylation changes at the same 571 ALP genes as above, profiling a total of 236,363 cytosine sites in the mouse cecal patch

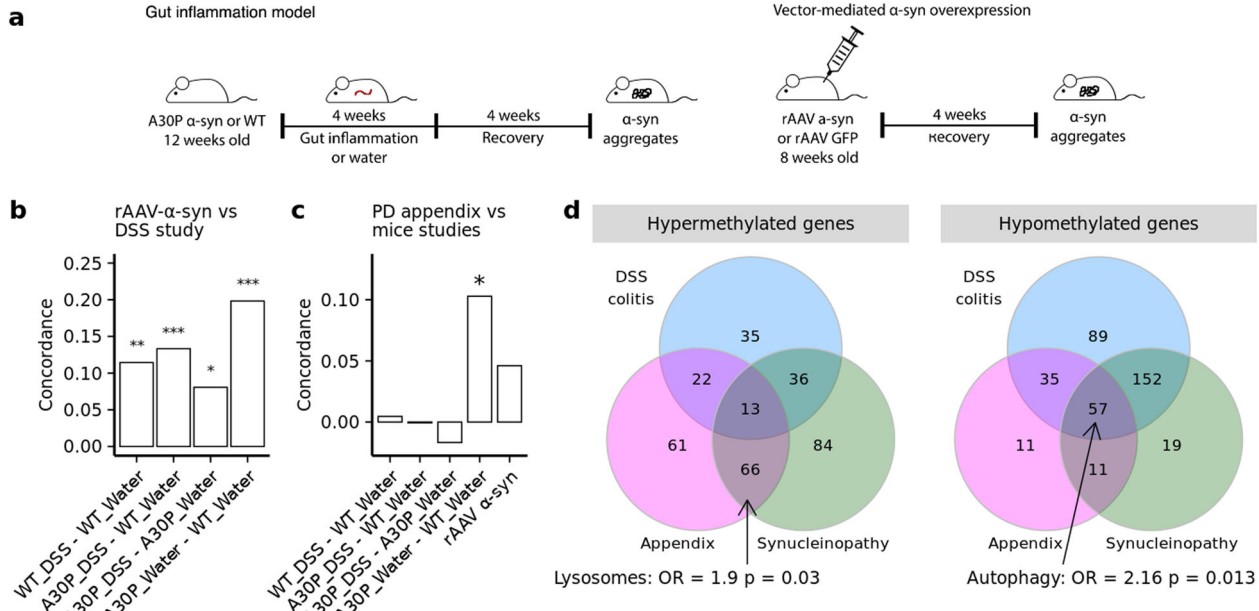

**Fig. 5 ALP changes in DNA methylation in response to experimental gut inflammation and α-syn aggregation.** DNA methylation was fine-mapped in the cecal patch of mice exposed to chronic DSS colitis, examining both wild-type mice and a mouse model of synucleinopathy, A30P α-syn mice ($n = 40$ mice: 9 A30P/DSS colitis, 10 A30P/Water, 10 WT/DSS colitis, 11 WT/Water). DNA methylation changes in response to α-syn aggregation were examined using a rAAV-mediated α-syn overexpression mouse model ($n = 5$ control vector and 5 α-syn overexpression vector in wild-type mice). **a** Schema of experimental design for the gut inflammation (left panel) and vector-mediated α-syn aggregation (right panel) studies. **b** Comparison of ALP changes induced by rAAV vector-mediated α-syn aggregation to those mediated by gut inflammation in wild-type and A30P α-syn mice. ALP gene enrichment in differentially methylated cytosines was determined. Plot shows concordance of epigenetic changes at ALP genes between rAAV-mediated α-syn aggregation mice and the wild-type and A30P α-syn mice in the DSS colitis study. **c** Comparison of ALP changes occurring in the PD appendix to those mediated by α-syn aggregation or gut inflammation. Concordance of epigenetic changes at ALP genes between PD appendix study and mouse studies. *$p < 0.05$, **$p < 0.01$, ***$p < 0.001$ Kendall's rank correlation coefficient, two-sided test. **d** Overlap of hyper- and hypo- methylated genes differentially methylated in PD appendix, rAAV a-syn and A30P water treated mice with synucleinopathy and A30P and wild-type mice with DSS colitis. Genes in each overlaping domain were tested for enrichment by ALP pathway genes using two-sided Fisher's exact test. Actual p values are reported in Supplementary Data 21.

(Supplementary Fig. 1). We found 896 differentially methylated cytosines affecting 365 ALP genes in response to α-syn aggregation (2.5 differentially methylated sites per affected ALP gene with 10.85% average methylation change; $q < 0.05$, robust linear regression; Supplementary Fig. 14; Supplementary Data 16). We next determined whether similar ALP genes were epigenetically altered in mice with rAAV-mediated α-syn overexpression and mice exposed to DSS-mediated intestinal inflammation. We observed that α-syn overexpression and aggregation by the rAAV vector paralleled the effects of A30P α-syn overexpression on the ALP epigenome in the mouse cecal patch ($r = 0.20$, $p = 3.65 \times 10^{-6}$, Kendall's rank correlation coefficient; 246 overlapping differentially methylated genes; Fig. 5b). Interestingly, we found a significant positive correlation between the epigenetically altered ALP genes in response to inflammation and those altered in response to α-syn aggregation by rAAV-mediated overexpression (wild-type and A30P α-syn mice: $r = 0.11$, $p = 3.25 \times 10^{-3}$ and $r = 0.13$, $p = 4.76 \times 10^{-4}$, Kendall's rank correlation coefficient; 298 and 303 overlapping differentially methylated genes, respectively; Fig. 5b). Thus, ALP genes epigenetically altered by α-syn aggregation are also affected by intestinal inflammation.

We determined whether ALP genes altered by α-syn aggregation and gut inflammation in the cecal patch are consistent with those differentially affected in the appendix of PD patients. We found that differentially methylated ALP genes were concordant between the A30P α-syn overexpression mouse model and the PD appendix ($r = 0.10$, $p = 0.04$, Kendall's rank correlation coefficient; Fig. 5c). Of the 326 differentially methylated ALP genes in

the PD appendix, there were 176 gene homologs that were differentially methylated in the mice overexpressing A30P α-syn relative to wild-type mice. Further, pathway analysis revealed that α-syn overexpression in the A30P mouse model recapitulated the abnormalities in lysosome function in the PD appendix and affected autophagy (OR = 2.63, $p = 0.001$ and OR = 1.85, $p = 0.04$, respectively, Fisher's exact test; Supplementary Fig. 15). In addition, we examined the overlap of the genes exhibiting predominant hypermethylation or hypomethylation in the PD appendix, in mice with synucleinopathy, and in mice in response to DSS colitis (Fig. 5d; Supplementary Data 17). We performed a pathway analysis for overlapping genes, and again found that the PD appendix and mice with synucleinopathy shared a hypermethylation of lysosomal genes (OR = 1.9, $p = 0.03$; Fisher's exact test). Meanwhile, there was a hypomethylation of genes involved in autophagy that overlapped between PD appendix, mice with synucleinopathy and following DSS colitis (OR = 2.16, $p = 0.01$; Fisher's exact test). Therefore, even though there is a predominant hypermethylation of lysosomal genes in the PD appendix and induced by synucleinopathy, there are autophagy genes exhibiting hypomethylation that are in common between the PD appendix, mice with synucleinopathy, and mice exposed to gut inflammation.

## Discussion
Our highly detailed deep-sequencing maps of 521 ALP genes in the healthy and PD human appendix and in neurons of the prefrontal cortex and olfactory bulb serve as a rich resource for

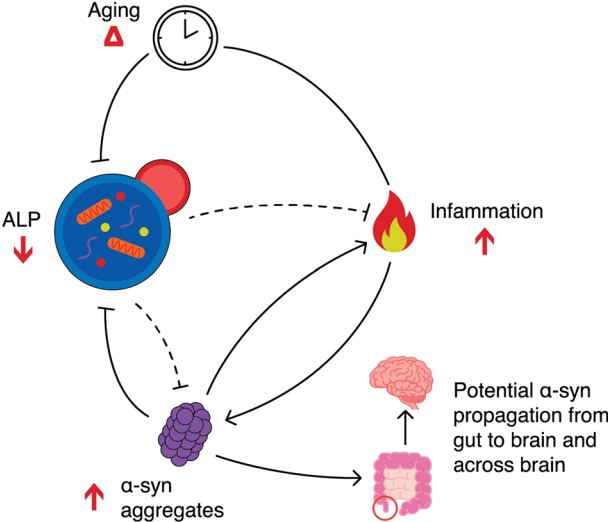

**Fig. 6 Proposed model of ALP changes in the PD appendix and brain.**
Model based on our study and the literature[14,101] illustrating the interplay between the ALP, aging, inflammation, and α-syn aggregates, and their contribution to the development and progression of PD. The healthily functioning ALP is responsible for the breakdown of physiological and aggregated α-syn[14]. In PD, widespread epigenetic silencing of ALP genes leads to decreased lysosomal functioning. This promotes an accumulation of α-syn aggregates, which reciprocally furthers ALP dysfunction in PD. In aging, there is an epigenetic inactivation of macroautophagy and selective autophagy genes, with concomitant decline in ALP activity, which places individuals of advanced age at greater risk of developing PD. The ALP also moderates inflammatory responses[101]. PD patients may exhibit heightened responses to inflammation as result of α-syn accumulation and ALP dysregulation. Loss of ALP function in PD also enables the secretion and cell-to-cell transfer of aggregated α-syn[85]. Hence, epigenetic disruption of the ALP in the gut and brain may contribute to the development and progression of α-syn pathology. While the causative factors of the epigenetic dysregulation of the ALP in PD remain unclear, joint contribution of genetic risk factors[23,25] and α-syn accumulation triggering epigenetic disruption of the ALP, environmental agents[93] and abnormal shifts in the microbiome[94] may play a role, especially because they can impact gut inflammation. The appendix, a potential initiation site for synucleinopathy in idiopathic PD, is circled in red. Red arrows indicate direction of change in PD relative to controls. Dotted lines indicate interactions that are weakened by the epigenetic dysregulation of the ALP in PD.

exploring the dysregulation of ALP genes in PD. First, we demonstrate a pronounced hypermethylation of ALP genes in the PD appendix. Specifically, genes implicated in lysosomal function exhibit significant downregulation. Second, we confirm that the pattern of hypermethylation in the PD appendix is recapitulated in the PD brain, in the olfactory bulb and in prefrontal cortex neurons, with a significant overlap of the specific ALP genes affected. Third, we observe a gradual hypermethylation of selective autophagy genes with age in the healthy appendix and brain, as well as a hypermethylation of macroautophagy genes in the brain. Fourth, we demonstrate that the normal epigenetic age effects of ALP genes are disrupted in PD. Fifth, and finally, we show that overexpression of α-syn sensitizes to the effects of gut inflammation in vivo. Together, our translational study demonstrates that the ALP, and specifically the lysosomal pathway, is epigenetically disrupted in the PD appendix, making it a potential culprit for PD initiation and progression (Fig. 6).

We found a widespread epigenetic silencing of the ALP in the PD appendix and brain, particularly for lysosomal genes, with

concomitant transcriptional downregulation of lysosomal pathways. Our human studies do not discern whether epigenetic changes at ALP genes are causal to PD or a consequence of PD pathophysiology, though we used a computational approach (SVA) and follow-up studies in mice in an effort to address potential confounding factors (i.e., non-shared environment, cell-type heterogeneity). Moreover, an epigenetic inactivation of the lysosomal system is in line with PD GWAS identifying a role for lysosomal genes in PD[23]. It is also consistent with studies identifying considerable genetic risk overlap between PD and lysosomal storage diseases, such as Gaucher's disease[26] and Niemann-Pick disease[76]. Lending further evidence to the centrality of lysosomal function to PD, human postmortem studies indicate abnormalities in lysosomal proteins in idiopathic PD[17,77]. Transcriptomic studies of the PD brain and of induced dopaminergic neurons from PD patients support lysosomal dysregulation in this disease[25,78–80]. A recent study also demonstrated an abundance of lysosomal markers and vesicles in PD Lewy pathology[81]. The genetic and/or chemical manipulation of lysosomes in model systems contribute to the accumulation of α-syn[82], while aggregated α-syn itself appears to impede lysosomal function[83,84]. There is also evidence that compromised lysosomal function, leading to a decrease in autophagic flux, contributes to the secretion and cell-to-cell transmission of α-syn aggregates[18,85], which may occur via exosomes[22], endocytosis[86], or tunneling nanotubes[87]. Here, we present a new line of evidence suggesting that epigenetic misregulation of the ALP, particularly of lysosomes, is responsible for its dysfunction in the PD gut and brain.

Epigenetically induced dysfunction of the ALP in the PD appendix may contribute to the accumulation of aggregated α-syn in the gut and brain. In the PD appendix, widespread epigenetic inactivation of the ALP signifies that this tissue site exhibits molecular changes capable of promoting synucleinopathy; however, this study does not delineate whether the ALP dysregulation in the PD appendix preceded that of the PD brain. Nonetheless, imaging studies of prodromal PD patients support that pathology can occur in the gut prior to the brain[11]. Aggregated α-syn has also been detected in enteric neurons of individuals in the prodromal stage of PD[7,11]—in some cases as early as 20 years prior to the onset of motor symptoms[4]. Studies using animal models have demonstrated that α-syn pathology is capable of propagating from the gut to the brain via the vagus nerve[8,9], though there is also evidence for bidirectional transfer of aggregated α-syn between the gut and brain[88,89]. Transfer of α-syn pathology from the enteric nervous system to the CNS has been observed in a variety of α-syn models[72,90,91]. While it is unclear, what triggers the enteric α-syn pathology and its propagation to the brain, recent studies in α-syn transgenic mice demonstrate that α-syn pathology develops in the brain 18 months post an experimental form of colitis but not at 6 months post colitis which was accompanied by dopaminergic neuronal loss in the substantia nigra[92]. In humans, colitis and the prodromal appearance of enteric α-syn pathology have also been implicated as a risk factor for PD as well as several genes related to immune function[92]. We previously reported that there can be an abundance of aggregated α-syn in both the healthy and PD appendix, although α-syn levels are up to three times greater in the PD appendix[12]. In combination with epigenetic perturbation of lysosomal function, hypomethylation of the α-syn gene in the PD appendix may propel α-syn pathology. Indeed, studies in the brain have found that endogenous α-syn levels influence the spread of synucleinopathy[93]. Thus, epigenetic changes in the PD appendix are consistent with an increased production and impaired clearance of α-syn pathology.

In patients in which PD has fully emerged, epigenetic changes at ALP genes in the PD appendix are similar to those occurring in

the PD brain. Our proteomic analysis also identified consistently increased levels of α-syn protein (SNCA) and decreased levels of NAMPT, HSPA8, and VPS35 in both the PD appendix and brain. Similarities across the PD gut and brain signify that the epigenetically dysregulated genes that enable the development, progression, and transport of α-syn pathology in the appendix could also facilitate the propagation of pathology within the brain. The appendiceal orifice is routinely identified during a total colonoscopic examination and can be biopsied[94], and as such, is more accessible than the brain. Our findings suggest that epigenetic changes in the ALP in the appendix may serve as a proxy for ALP status in the brain, though this would require further investigation across disease stages.

Typical epigenetic age effects on the ALP appear to be disrupted in PD. In the healthy appendix and prefrontal cortex, we found that the cytosines hypermethylated with age were overrepresented among genes in the selective autophagy pathway. In the brain, hypermethylated cytosines were also overrepresented at macroautophagy genes. Selective autophagy, which involves the targeting of specific cargoes by ubiquitin tagging, includes the clearance of intracellular pathogens, also referred to as xenophagy[95]. The accumulation of suppressive epigenetic marks affecting selective autophagy with age suggests that the brain may be more susceptible to infection in advanced age. Chaperone-mediated autophagy (CMA) and macroautophagy are thought to be key pathways through which physiological α-syn and aggregated α-syn, respectively, are cleared from the cell[14,83]. The hypermethylation of macroautophagy genes in the healthy aging brain may consequently render older individuals more vulnerable to the accumulation of α-syn aggregates. Though advanced age places individuals at greater risk for PD, the ALP in PD patients fails to exhibit normative epigenetic changes with age. It may be the case that, in PD, various ALP pathways (e.g., macroautophagy, selective autophagy) do not undergo the same extent of hypermethylation as in healthy aging, in a futile attempt to compensate for the decrease in autophagic flux induced by lysosomal dysfunction.

For this study we selected prefrontal cortex neurons because of their relevance in later stages of PD and because prefrontal cortex neurons still exist in the analyzed early stage (Braak stage 3–4) postmortem PD brain. In contrast, substantia nigra neurons have largely degenerated and therefore are insufficiently present or are too advanced in the degenerative process for isolation[96]. On a technical level, neuronal nuclei isolation has been fully optimized for the prefrontal cortex, and the prefrontal cortex yields sufficient numbers of neurons for DNA methylation analysis[52,97]. Epigenetic (DNA methylation) changes in prefrontal cortex neurons can occur early in neurodegenerative diseases[52,97,98]. Hence, for our study of molecular changes in the early PD brain neurons, the prefrontal cortex offers a disease-relevant and available source of neurons, in combination with technical feasibility. Our study of normal age-related methylation changes in the appendix is somewhat limited by potential confounding factors in the individuals from which it was obtained. Although the control appendix tissue was confirmed to be histologically normal, the patient diagnosis of intestinal cancer leading to incidental appendix removal may have impacted some of the methylation changes observed, particularly those related to inflammatory pathways. Nevertheless, we verified that age-related genes observed in these appendices overlap with those seen in prefrontal cortex neurons from other control individuals, suggesting that the control appendix samples exhibit healthy aging, at least in part. Follow-up studies in a second cohort of the normal appendix would further validate our results and strengthen our understanding of normal age-related changes of methylation in the appendix.

The causative factors of the epigenetic dysregulation of the ALP in PD remain unclear, although there is evidence for a bidirectional relationship between α-syn and the ALP[19,21]. Decreased autophagic flux results in an accumulation of α-syn[19,21], and misfolded α-syn itself appears to play an active role in suppressing the ALP[83,84]. In this way, genetic and/or epigenetic defects in the ALP leading to α-syn accumulation could lead to further (epigenetic) dysregulation of the ALP. This is supported by our finding that the same ALP genes are disrupted in a mouse model with α-syn overexpression as in the human PD appendix. In addition to the joint contribution of genetic risk factors[23,25] and α-syn accumulation triggering epigenetic disruption of the ALP, environmental agents[99] and abnormal shifts in the microbiome[100] may play a role, especially because they can impact gut inflammation.

There is a robust interplay between inflammation and the ALP[101]. Autophagy is involved in the induction and suppression of inflammation[101,102]. It regulates the development, homeostasis, and survival of inflammatory cells and affects the transcription, processing, and secretion of cytokines[101]. Loss of autophagy has proinflammatory consequences, and in the gut, ALP suppression exacerbates the inflammatory effects of DSS colitis[102,103]. In our study, gut inflammation mediated by DSS largely induced a hypomethylation of the ALP. This is consistent with evidence that activation of autophagy is important for resolving inflammation[101,102]. Furthermore, DSS colitis effects on the ALP were amplified by synucleinopathy. Synucleinopathy in mice induced a hypermethylation of lysosomal genes which was consistent with that observed in PD. Thus, the exaggerated response to an inflammatory event may be an increased activation of autophagy in an attempt to overcome deficient lysosomal function mediated by α-syn accumulation. Indeed, we find that autophagy genes exhibiting hypomethylation are in common between the PD appendix and mice models of synucleinopathy and gut inflammation, while lysosomal genes are hypermethylated in the PD appendix and by synucleinopathy. Excessive or prolonged autophagy stimulation is detrimental as it can lead to cell death, including in the GI tract (known as autophagic cell death and autosis)[104–107]. Thus, our study suggests that epigenetically mediated ALP abnormalities in the PD appendix may, in part, be due to an accumulation of α-syn. Given the evidence supporting that inflammation (including in the GI tract) plays a key role in PD pathogenesis[69,70], it is possible that normative ALP activation needed to resolve inflammation is incapacitated in PD.

Recent studies have also provided evidence for a link between gut inflammation and PD[108], including a higher likelihood of developing PD among individuals with inflammatory bowel disorders (IBD)[69,70]. Several of the ALP genes we found consistently epigenetically suppressed across the PD gut and brain have direct roles in inhibiting inflammation under physiologically normal conditions[51,67,109–115]. In addition, two of the top dysregulated ALP genes, *TIAL1* and *CALCOCO2*, have been implicated as key factors in ulcerative colitis- and Crohn's disease-related inflammation, respectively[67,112]. There is also evidence for the involvement of α-syn in rallying an immune response in the GI tract in response to infection[116], and that α-syn is upregulated in response to viral[117] and bacterial infections[118]. Inflammation, induced by DSS, activated the ALP in both wild-type and A30P α-syn overexpressing mice, with a significantly exaggerated response in the A30P α-syn mice. If α-syn pathology does originate in the gut for some individuals, then the heightened response in the A30P α-syn mice suggests that α-syn-dependent changes in the ALP may sensitize individuals to an increased risk of developing both PD and IBD. Hence, abnormalities in gut α-syn expression and the effects of α-syn on the ALP may contribute to the heightened co-occurrence of IBD and PD[69,70].

In sum, the ALP faces widespread epigenetic dysregulation in the PD appendix and brain. Our findings add to evidence suggesting the potential for the gut as an initiation site for idiopathic PD. Further study of the ALP abnormalities we have identified could provide key insights into gut-based methods for tracking disease progression and for preventative strategies.

## Methods
Within each dataset all measurements were taken from distinct samples. All data analyses were performed using R programming language (v 3.6.1), unless explicitly specified otherwise.

**Human tissue samples**. Postmortem appendix tissue from PD patients and controls was obtained from the Oregon Brain Bank. In our age effect analysis, we included surgically-isolated, histologically normal appendix tissue from control (non-PD) individuals obtained from the Spectrum Health Universal Biorepository and Cooperative Human Tissue Network (CHTN). Appendix surgical samples were from individuals undergoing a right hemicolectomy for intestinal cancer not involving the appendix (appendix incidentally removed and histologically confirmed to be normal). Prefrontal cortex tissue was obtained from the NIH NeuroBioBank, Parkinson's UK Brain Bank, Michigan Brain Bank (primary cohort), or the Oregon Brain Bank (replication cohort). Olfactory bulb tissue was obtained from the Oregon Brain Bank. For the study samples, we had information on demographics (age, sex), tissue quality (postmortem/surgical interval), and pathological staging (Supplementary Data 15). Appendix, prefrontal cortex, and olfactory bulb postmortem tissue from PD patients have evident brain Lewy pathology (PD Braak stages III-VI), whereas control individuals have no Lewy pathology in the brain. Sample information is detailed in Supplementary Data 18. All fresh tissues were snap-frozen and stored at −80 °C until time of processing. Neuronal nuclei from prefrontal cortex were isolated using an antibody (NeuN) and flow cytometry–based approach[52,97]. First, fresh frozen tissue was homogenized on ice in PBSTA buffer (0.3 M sucrose, 1 × Dulbecco's PBS (Gibco), 3 mM MgCl2) for three intervals of 5 s (BioSpec Tissue Tearor, on lowest setting), followed by incubation for 15 min with 0.2% Triton X-100. Next, the tissue homogenate was further homogenized with eight strokes in a dounce (Kimble). To remove debris, the homogenate was filtered through Miracloth (Calbiochem) onto a sucrose cushion (1.4 M sucrose, 1 × Dulbecco's PBS (Gibco), 0.1% TritonX-100, 3 mM MgCl2) and centrifuged at 3000 × g for 30 min at 4 °C. The pelleted nuclei were resuspended in blocking buffer (1 × PBS, 1.25% goat serum (Gibco), 3 mM MgCl2, and 0.0625% BSA (Thermo Fisher Scientific)) and incubated with anti-NeuN antibody (1:500, Abcam) for at least 30 min on ice. Immediately before sorting, 10 µL of 7-AAD or DAPI (Thermo Fisher Scientific, Sigma-Aldrich) was added to each sample, and nuclei samples were filtered through a 41-µm filter (Elko Filtering Co.). Samples were sorted on a MoFlo Astrios in the Flow Cytometry Core of the Van Andel Research Institute, using the gating strategy described in Supplementary Fig. 16. After sorting, nuclei were pelleted by bringing the total volume to 10 mL in 1 × PBS (Gibco) and adding 2 mL 1.8 M sucrose, 50 µL 1 M calcium chloride, and 30 µL of 1 M magnesium acetate. Samples were mixed by inverting and incubated on ice for 15 min before centrifuging at 2500 × g for 10 min. The remaining pellet was frozen at −80 °C until DNA isolation. The study complied with all relevant ethical regulations for work with human participants, informed consent was obtained and the protocol was ethically approved by the institutional review board at the Van Andel Research Institute (IRB #15025).

**DNA methylation fine-mapping with bisulfite padlock probes**. We comprehensively profiled DNA methylation at all genes involved in the ALP, as reported on the Human Autophagy Database ((http://autophagy.lu)[39] and Human Lysosome Gene Database (http://lysosome.unipg.it)[40]. To our panel, we also added genes from the PDGene website (http://www.pdgene.org/) because of the involvement of PD risk genes in the ALP[23,43,44]. DNA methylation was fine-mapped at 521 ALP genes in the appendix, prefrontal cortex neurons, and olfactory bulb of PD patients and controls, using a targeted bisulfite sequencing approach known as bisulfite padlock probe sequencing. We designed a library of 67,789 unique probes targeting the nonrepetitive genome of each human ALP gene and surrounding genomic area (± 300 kb). The ppDesigner software[119] was used to design padlock probes for both the forward and reverse DNA strand on a bisulfite-converted human GRC37/hg19 genome. Probes were synthesized using a programmable microfluidic microarray platform (CustomArray, Inc.), and probe preparation and purification was done as described[120]. Probes targeting the human ALP genes are listed in Supplementary Data 19.

DNA methylation fine-mapping was performed using the bisulfite padlock probe technique[120]. gDNA from the appendix, prefrontal cortex neurons, and olfactory bulb was isolated using standard phenol–chloroform extraction methods. Bisulfite conversion of gDNA and purification was done using the EZ DNA Methylation Kit (Zymo Research) and quantified using a Qubit 3.0 Fluorometer (Thermo Fisher Scientific). For each sample, the bisulfite-converted DNA (200 ng) was hybridized to the padlock probes (1.5 ng) at 55 °C for 20 h. Target regions flanked by the hybridized padlock probes were extended using PfuTurbo Cx

(Agilent Technologies) and circularization was completed using Ampligase (Epicentre). The non-targeted, non-circularized DNA was digested using exonuclease (Exonuclease I and III; New England Biolabs). The target circularized DNA was amplified using a common linker sequence within the padlock probe. Libraries were then PCR amplified, pooled in equimolar amounts, and purified by QIAquick Gel Extraction kit (Qiagen). Libraries were quantified by qPCR (Kapa Biosystems) on a ViiA 7 Real-time PCR system (Applied Biosystems). Next-generation sequencing of the libraries was performed on an Illumina HiSeq 2500 machine in HiOutput mode at the Epigenetics Lab at the Centre for Addiction and Mental Health in Toronto, Canada, which generated 30–50 million reads/sample.

For the studies in mice, we examined DNA methylation at 571 ALP genes, the orthologs of the human ALP genes. BiomaRt (v2.38.0) identified 571 ALP mouse gene homologs and their genomic coordinates. Padlock probes were designed to the bisulfite-converted GRCm38/mm10 genome. We designed 87,605 probes to target the mouse ALP genes and surrounding genomic area (±50 kb). Padlock probe synthesis and library preparation was performed as described above. gDNA from the mouse cecal patch was obtained using standard phenol–chloroform extraction methods. Probe design and DNA methylation fine-mapping with the bisulfite padlock probe technique was performed as described above. Probes for the mouse ALP genes are listed in Supplementary Data 16.

**Fine-mapping of DNA methylation by bisulfite padlock probe sequencing**. DNA methylation status at ALP genes in humans and mice was determined using a custom pipeline[52,120]. Trim Galore (v0.4.4) was used to remove low-quality reads and adapters. Bismark (v0.17.0)[121] was used to align bisulfite sequencing reads to a masked reference genome (GRCh37/hg19 for human data or GRCm38/mm10 for mouse data) and the base pair–specific methylation calls were extracted. Methylation calls overlapping padlock probe arms were removed using Bedtools (v2.25.0). Methylation calls were only considered for cytosines with at least a 30 × read depth. Bisulfite conversion efficiency was 99.60% ± 0.003% for the human data and 99.62% ± 0.006% for the mouse data (as measured by average CC methylation ± s.e.m.). This approach was applied to the seven DNA methylation datasets: 1) PD and control appendix, 2) aging healthy appendix, 3) the primary cohort of PD and control prefrontal cortex neurons, 4) the replication cohort of PD and control prefrontal cortex neurons, 5) PD and control olfactory bulb, 6) the gut inflammation mouse model, and 7) the rAAV-α-syn overexpression mouse model (Supplementary Fig. 1). Preprocessing was done jointly for the human appendix datasets (n = 77 unique samples, 16 technical replicates), as well as for the mouse datasets (n = 50 unique samples, 7 technical replicates). Brain datasets were each preprocessed independently: prefrontal cortex neurons (98 unique samples, 4 technical replicates in primary cohort; 31 unique samples in replication cohort) and olfactory bulb (24 unique samples, 3 technical replicates).

We confirmed that technical reproducibility exceeded biological variability in the human and mouse datasets. In the prefrontal cortex neuron datasets, there was one sample in each dataset excluded because of a low inter-sample cytosine methylation correlation (Pearson correlation coefficient r ≤ 0.9). For technical replicate samples, methylation calls and counts were averaged for each site. Sites with less than 70% of samples containing methylation calls as well as sites with all samples having a methylation call of 0 were removed from further analysis. Sites overlapping common SNPs in the population (minor allele frequency ≥ 0.05), as identified by the 1000 Genomes Project (phase 3 v5a 20130502 release for chr1-chr22, v1b 20130502 for chrX; all populations and European populations) were also removed from further analysis.

We next explored the data using principal component analysis (PCA) of beta values defined as $B = M / (M + U)$. PCA of all datasets was computed and sample projection onto the first two principal components was plotted, which served to identify outlying samples. Samples deviating from the center of either of the first two principal components by more than 3 standard deviations were deemed outliers and removed from further analysis. Next, the distribution of U and M read counts across experimental batches was used to determine whether the sample cohort should be normalized. Normalized B values for appendix datasets were obtained from quantile normalized and rounded M and U read count matrices. Sites with stable DNA methylation calls (0%, 100%, and/or missing) in 50% or more of the samples were excluded from further analysis.

After data processing we had the following datasets for downstream statistical analysis: (1) 182,024 CpGs in 24 PD and 19 control appendix samples, (2) 181,151 CpGs in 24 PD and 51 healthy appendix, (3) 130,733 CpGs and 696,665 CpHs in the primary cohort of 52 PD and 42 control prefrontal cortex neurons, (4) 110,397 CpGs in the replication cohort of 13 PD and 15 control prefrontal cortex neurons, 95) 143,553 CpGs in the 9 PD and 14 control olfactory bulb, (6) 240,366 CpGs in the gut inflammation model (n = 11 wild-type + water, 10 wild-type + DSS, 10 A30P + water, 9 A30P + DSS), and 7) 236,363 CpGs in the rAAV-α-syn overexpression mouse model (n = 5 rAAV-GFP, 5 rAAV-α-syn).

For each dataset, m values were computed from beta values. First, extreme B values were shrunk towards the center $B' = (B * (f−1) + 0.5)/f$, where f is the shrinkage factor set to 1000 for appendix and mice datasets, and, to accommodate CpHs it was set to 2000 for brain data. Next, m values were computed as $m = \log_2(B'/(1−B'))$. Robust linear regression was fitted to each cytosine site with a maximum number of model fitting iterations set to 100 using limma[122]. The

regression models for the human datasets included the following covariates: diagnosis (PD, control), age, sex, postmortem interval, and surrogate variables to correct for other sources of variation. Surrogate variables were computed using SmartSVA[123], a benchmark approach to control for cell-type heterogeneity and effective removal of other sources of variation[45–47]. The number of surrogate variables was determined by performing PCA on residuals of the $m$ value matrix after regression of all known covariates and counting the number of principal components explaining more than 5% of variability. In the prefrontal cortex neuron dataset, we also adjusted for neuronal subtype proportion, which refers to the proportion of glutamatergic to GABA neuronal subtypes, using cell-type deconvolution performed with CIBERSORT[124] (http://cibersort.stanford.edu; 100 permutations) and neuronal subtype-specific signatures[125]. For analysis of age effects, a diagnosis–age interaction variable was also added to the models. The robust linear regression models for the mouse DSS study included interaction of genotype and treatment, as well as sex, while the viral vector mouse study included treatment. Empirical Bayes correction was applied to model fits[126]. Limma's *contrasts.fit* function was used to perform group comparisons in the age effect and mouse DSS datasets. For each dataset and contrast, $p$ values were adjusted with a Benjamini-Hochberg correction for multiple testing and those with FDR $q < 0.05$ were deemed significant. Change in methylation percentage was computed by converting fitted $m$ values of each cytosine and each sample into beta values using formula $B = 2^m/(2^m + 1)$. Linear models were then fitted on the $B$ values and model coefficients were extracted.

No statistical methods were used to predetermine sample size. Sample sizes were selected based on the available appendix tissue, the prior publications using comparable methods in human postmortem neurons and mouse studies of neurodegenerative disease[52,53]. Subsequent to our DNA methylation analysis we confirmed that our DNA methylation analysis had sufficient statistical power (Fig. S17).

**Transcriptomic analysis by RNA sequencing**. The transcriptome was profiled in the PD and control appendix and in the healthy aging appendix by RNA sequencing (RNA-seq). RNA-seq was performed on a random subset of the same appendix samples used in DNA methylation analysis (Supplementary Fig. 1). Frozen appendix tissue (~20 mg) was homogenized using a Covaris cryoPREP pulverizer and then in 1 mL of TRIzol (Life Technologies) with a ceramic bead-based homogenizer (Precellys, Bertin Instruments). Total RNA was isolated according to the TRIzol manufacturer's instructions, treated with RNase-free DNase I (Qiagen) at room temperature for 30 min, followed by clean-up with the RNeasy Mini Kit (Qiagen). RNA yield was quantified using a NanoDrop ND-1000 (Thermo Fisher Scientific) and RNA integrity was verified with an Agilent Bioanalyzer 2100 system (Agilent Technologies). Libraries were prepared by the Van Andel Genomics Core from 300 ng of total RNA using the KAPA RNA HyperPrep Kit with RiboseErase (v1.16; for PD vs control appendix) or KAPA stranded mRNA-seq kit (v5.17; for aging appendix) (Kapa Biosystems). RNA was sheared to 300–400 bp. Prior to PCR amplification, cDNA fragments were ligated to NEXTflex dual adapters (Bioo Scientific). The quality and quantity of the finished libraries were assessed using a combination of Agilent DNA High Sensitivity chip (Agilent Technologies, Inc.), QuantiFluor dsDNA System (Promega Corp.), and Kapa Illumina Library Quantification qPCR assays (Kapa Biosystems). Individually indexed libraries were pooled, and 100-bp, single-end sequencing was performed on an Illumina NovaSeq6000 sequencer using an S1 100 cycle kit (Illumina Inc.), with all libraries run on a single lane to return an average depth of 37 million reads per library. Base calling was done by Illumina RTA3 and output of NCS was demultiplexed and converted to FastQ format with Illumina Bcl2fastq v1.9.0.

For the RNA-seq analysis, Trim Galore (v0.4.4) was applied to remove low-quality bases and for adapter trimming. STAR (v2.5.3a)[127] was used to index Ensembl GRCh37 p13 primary assembly genome with the Gencode v19 primary assembly annotation. STAR (v2.5.3a)[127] was then used for read alignment to the genome. Expected gene-level counts were obtained after processing of raw RNA-seq reads with RSEM[128]. The edgeR package[129] was used to read the counts of all samples and transform them into count per million values (CPM). Only the genes having more than 1 CPM in 80% of samples were retained. Next, transcript per million (TPM) estimates were loaded from RSEM output. TPM was used for subsequent model fitting following the same procedure as for padlock bisulfite sequencing. Genes falling into the group of 10% least variable genes were removed. Using voom[130], TPM count data were transformed to log2-counts per million (logCPM) and adjusted for mean-variance relationship. The resulting matrix was used to calculate surrogate value vectors and perform linear model fitting. The linear regression model approach is commonly used for RNA-seq analysis[52,131]. For the PD vs control appendix dataset, a robust linear regression model was used to examine the effects of diagnosis, after adjusting for age, sex, postmortem interval, RNA integrity (RIN), and surrogate variables (SmartSVA vectors). For the analysis of age effects in appendix, a robust linear regression model was used to examine age, diagnosis, and their interaction, after adjusting for sex, postmortem interval, batch, RIN, and surrogate variables (SmartSVA vectors). The age covariate was used to determine age effects. *P*-values for each tested contrast were adjusted using Benjamini-Hochberg false discovery rate correction, and genes with FDR $q < 0.05$ were deemed significant.

No statistical methods were used to predetermine sample size. However, subsequent to our RNA-seq analysis we confirmed that our RNA-seq analysis had sufficient statistical power (Fig. S17).

We examined the association between DNA methylation and transcriptome changes in the PD appendix. For DNA methylation, absolute weighted mean value was used to emphasize differentially methylated cytosines in the PD appendix. We then computed a weighted Pearson correlation comparing the mean DNA methylation fold change at ALP genes to their corresponding ALP transcript fold change in the PD appendix ($n = 422$ genes). Log transformed differential expression $p$-values were used as weights for the correlations.

**Identification of active enhancers and promoters in the appendix**. Chromatin immunoprecipitation (ChIP) was used to identify active enhancers and promoters in the healthy human appendix ($n = 3$ individuals). The ChIP library preparation protocol was based on that of the NIH Roadmap Epigenomics Project[132]. Appendix tissue was first cryoground into a fine powder using liquid nitrogen and a mortar and pestle. Frozen appendix tissue powder (~50 mg) for each sample was quickly suspended into 5 mL of ice-cold PBS and 0.5 mL of fresh crosslinker buffer (0.1 M NaCl, 1 mM EDTA, 0.5 mM EGTA, 50 mM HEPES pH 8.0, and 11% formaldehyde), and the sample was nutated for 7 min at room temperature. Cross-linking was quenched in 125 mM glycine and rotation for 5 min at room temperature. Samples were then centrifuged for 15 min at 2,000 × $g$ and the pellet was washed with 5 mL of ice-cold PBS. Samples were then centrifuged again for 10 min at 2,000 × $g$ and the pellet was stored at −80 °C until further processing. Each cross-linked tissue sample was then lysed by adding 100 µl of lysis buffer (1% SDS, 50 mM Tris-HCl pH 8.0, 20 mM EDTA, Complete protease inhibitor cocktail (Roche), and 1 mM PMSF), passing the solution through a 26 g needle 10 times, followed by incubation on ice for 10 min. Samples were then placed into 1 mL of ice-cold TE buffer (1 mM EDTA, 10 mM Tris-HCl pH 8.0) and sonicated for 20 min using the E220 ultrasonicator (Covaris), which obtained a 200–500 bp sheared DNA length. To perform the ChIP, 200 µl of sheared DNA was exposed to 5 µg of anti-histone H3 acetyl K27 antibody (1:40, Abcam #ab4729), anti-histone H3 mono methyl K4 antibody (1:40, Abcam #ab8895), or control IgG (1:40, Cell Signaling). Sample tubes were incubated overnight at 4 °C with gentle nutation. Samples were then incubated with Dynabead Protein G beads (Thermo Fisher Scientific) for 10 min at room temperature. DNA-protein complexes on beads were washed 5 times with ice-cold modified RIPA (5 mM HEPES pH 8.0, 1% NP-40, 0.7% sodium deoxycholate, 0.5 M LiCl, Complete protease inhibitor cocktail (Roche), 1 mM EDTA) and then once with 150 µl of TE buffer. To elute DNA and reverse crosslinks, beads were incubated in 120 µl elution buffer (10 mM Tris-HCl pH 8.0, 1 mM EDTA, 1% SDS) at 65 °C overnight. For input DNA, 20 µl of sheared chromatin was added to 130 µl elution buffer and now processed in parallel with the ChIP samples. Following overnight incubation, 250 µl of TE was added to each sample with 0.2 mg/mL RNase A (Thermo Fisher Scientific) and incubated at 37 °C for 1 h. Then 0.4 mg/mL proteinase K (Thermo Fisher Scientific) was added to each sample and incubated at 55 °C for 1 h. DNA was then purified with phenol–chloroform extraction and suspended in 10 mM Tris-HCl pH 8.0. Target fold enrichment relative to IgG control (205.0 for H3K27ac, 44.5 for H3K4me1) was confirmed by qPCR on an ABI StepOne Plus system (primers in Supplementary Data 16).

Libraries for appendix ChIP (H3K27ac, H3K4me1) and input samples were prepared by the Van Andel Genomics Core from 10 ng of input material and all available immunoprecipitated material using the KAPA Hyper Prep Kit (v5.16) (Kapa Biosystems). Prior to PCR amplification, end-repaired and A-tailed DNA fragments were ligated to IDT for Illumina TruSeq UD Indexed Adapters (Illumina Inc.). The quality and quantity of the finished libraries were assessed using a combination of Agilent DNA High Sensitivity chip (Agilent Technologies, Inc.), and QuantiFluor dsDNA System (Promega Corp.). Sequencing (single-end 100 bp) was performed on an Illumina NovaSeq6000 sequencer producing ~99.6 million reads per sample. Base calling was done by Illumina RTA3, and output was demultiplexed and converted to FASTQ format with Illumina Bcl2fastq2 v2.19.

Adapter sequence from raw sequencing reads was removed using Trim Galore (v0.4.4). Sequenced reads from H3K27ac and H3K4me1 immunoprecipitated samples and matched input controls from the human appendix were mapped to the human reference genome (GRCh37/hg19) with BWA (v0.7.15)[133]. A combination of Picard and Samtools (v1.9) was used to mark and remove PCR duplicates, respectively. Deeptools (v2.3.1)[134] was used for quality controls and narrow peaks were called using MACS2[135] for each sample. Consensus peaks were called using MACS2[135] combined peak calls (removing blacklist regions) and IDR, following ENCODE ChIP-seq guidelines[136], which yielded 26,381 H3K27ac peaks and 37,727 H3K4me1 peaks.

**Enrichment and genomic region analysis**. All enrichment and overlap analyses were performed by computing odds ratios (a measure of the magnitude of enrichment) and evaluating their significance with Fisher's exact test. The odds of observing hypermethylated loci among the significantly differentially methylated cytosines (foreground) were compared to that of the rest of the interrogated loci (background). The same approach was used to examine the direction of change in the transcriptome. To examine the enrichment of differentially methylated

cytosines in types of genomic elements in the PD appendix, the Bioconductor annotatr package[137] was used to determine the coordinates of 3'UTRs, introns, exons, 5'UTRs, promoters, 1 to 5 kb upstream regions, intergenic regions, as well as CpG islands, shores, shelves, and open sea. Appendix ChIP-seq peaks were used to define active promoters as well as active and poised enhancers. To determine enrichment of differentially methylated cytosines in genomic elements, the odds of observing a differentially methylated cytosine within an element type was compared to that of the rest of the interrogated sites. Genomic element enrichment analyses for all, hypermethylated, and hypomethylated cytosines were performed similarly. We also examined the proximity of differentially methylated cytosines in the PD appendix to the transcription start site of ALP genes. Transcription start coordinates of ALP genes targeted in the DNA methylation analysis were obtained using Bioconductor package TxDb.Hsapiens.UCSC.hg19.knownGene. We examined all ALP genes targeted in the DNA methylation analysis, and we also stratified the ALP genes into those belonging to the lysosome pathway and the rest. Every cytosine was assigned a 20 kb-wide bin based on the distance to the transcription start site of the nearest ALP gene. For each bin and gene strata, the odds of observing differentially methylated cytosines within a given bin and strata were compared to the odds of observing differentially methylated cytosines outside of the bin in the same gene strata.

We also determined whether there was a significant overlap in ALP genes between PD appendix and brain datasets. Disrupted ALP genes had at least one differentially methylated site, and we determined the enrichment of overlap of disrupted ALP genes across datasets relative to the remaining genes. To determine the types of genomic elements with DNA methylation changes overlapping between the PD appendix and PD brain datasets, we examined the top 5000 most significant differentially methylated cytosines in each dataset. For each genomic element category, we evaluated the odds of observing differentially methylated cytosines in the same genomic element in the PD appendix and PD brain, compared to the odds of having differentially methylated genomic elements in only one of the tissues. In addition, the overlap of differentially methylated active promoters, and active and poised enhancers between PD appendix and PD brain was performed using the Appendix ChIP-seq ($n = 3$ individuals) and PsychENCODE data ($n = 9$ individuals, prefrontal cortex neurons)[138]. Regions belonging to the same category (active or poised enhancer, active promoter) and overlapping within 1 kb were examined.

In the Manhattan plots displaying the ALP gene analysis for each dataset, PD risk genes identified by GWAS[23] and having at least one differentially methylated cytosine are denoted. For any ALP gene the direction of DNA methylation change was determined by the direction of the majority of significant cytosines, with ties resolved by the direction of the weighted mean fold change of all cytosines pertaining to the gene. Log transformed $p$ value was used as weight putting the most emphasis on the most significant cytosine.

**Human pathway analysis**. We determined ALP pathways enriched with differentially methylated cytosines and differentially expressed genes. Gene lists belonging to ALP Gene Ontology (GO) pathways were obtained from http://geneontology.org (retrieved on 2019-07-01) with GO IDs: GO:0006914 for Autophagy, GO:0061684 for Chaperone-mediated autophagy, GO:0005764 for Lysosomes, GO:0016236 for Macroautophagy, and GO:0061912 for Selective Autophagy. For the DNA methylation data, pathway enrichment was determined by comparing the odds of differentially methylated cytosines within a pathway relative to the odds of differentially methylated sites among the rest of the interrogated sites in this study. For the RNA-seq data, nominally significant differentially expressed genes were used to test the enrichment of up- and down-regulated ALP pathways using Fisher's exact test. Transcriptome-wide functional annotation of differentially expressed genes was performed using DAVID (v6.8; https://david.ncifcrf.gov/summary.jsp). In DAVID, differentially expressed genes ($q < 0.05$) were the foreground list and all genes in this RNA-seq study was the background. Resulting functional annotations were clustered with EnrichmentMap[139] (default setting of Jaccard and overlap = 0.375) in Cytoscape (v3.7.1) and visualized with AutoAnnotate[140].

**Identification of the top differentially methylated ALP genes**. For the PD appendix, prefrontal cortex, and olfactory bulb datasets, the odds ratio of enrichment of ALP genes with significant differentially methylated cytosines was determined. To calculate an odds ratio, we compare the odds of observing a significantly differentially modified site within a gene versus the odds of observing a significantly differentially modified site genome-wide. Odds ratios were tested for statistical significance of enrichment using one-sided Fisher's exact test. Then, for each dataset, the genes were ranked according to their enrichment $p$-value. Using the robust rank aggregation algorithm[54] we obtained the ranking of ALP genes consistently disrupted across the PD appendix and brain tissues. The approach brings to the top the genes that rank consistently higher across all used datasets than expected by chance alone.

**Mass spectrometry and proteomics analysis**. Label-free quantitative proteomic analysis was performed on the appendix and prefrontal cortex of PD patients and controls ($n = 3$ PD and 3 controls per tissue). Mass spectrometry analysis of appendix and prefrontal cortex samples (~30 mg tissue) was performed by the Integrated Mass Spectrometry Unit at Michigan State University. The wet weight of each appendix and prefrontal cortex tissue was measured, and 5-fold lysate buffer (20 mM Tris Base (pH 7.4), 150 mM NaCl, 1 mM EGTA, 1 mM EDTA, 5 mM sodium pyrophosphate, 30 mM NaF, 1x Halt Protease inhibitor Cocktail (Thermo Fisher Scientific)) was used to homogenize the tissue on ice with a tissue grinder (Tissue Master 125, Omni International). The homogenate was centrifuged at $18,407 \times g$ for 10 min at 4 °C and the supernatant was transferred to a low-bind tube. Protein concentration was determined by BCA assay (Pierce BCA Protein Assay, Thermo Fisher Scientific). Protein lysate (10 μg) of the appendix and pre-frontal cortex was denatured using 25 mM ammonium bicarbonate/80% acetonitrile and incubated at 37 °C for 3 hr. The samples were dried and reconstituted in 50 μl of 25 mM ammonium bicarbonate/50% acetonitrile/trypsin/LysC solution (1:10 and 1:20 w/w trypsin:protein and LysC:protein respectively) and digested overnight at 37 °C. The samples were dried and reconstituted in 50 μl of 25 mM ammonium bicarbonate/5% acetonitrile.

Samples were loaded onto an UltiMate 3000 UHPLC system with online desalting. 10 μl of each sample was separated using a C18 EASY-Spray column (2 μm particles, 25 cm × 75 μm ID) and eluted using a 2 h acetonitrile gradient into a Q-Exactive HF-X mass spectrometer. Data-dependent acquisition for Full MS was set using the following parameters: resolution 60,000 (200 $m/z$), AGC target 3e6, maximum IT 45 s, scan range 300 to 1500 m/z, dynamic exclusion 30 s. Fragment ion analysis was set with the following parameters: resolution 30,000 (200 m/z), AGC target 1e5, maximum IT 100 ms, TopN 20, isolation window 1.3 m/z, NCE at 28. Each sample was run in triplicate. The mass spectra from each technical replicate were searched against the Uniprot human database (filtered-proteome_3AUP000005650) using LFQ method in Proteome Discoverer (v. 2.2.0.388, 2017) set as follows: at least 2 peptides (minimum length = 6, minimum precursor mass = 350 Da, maximum precursor mass 5000 Da), tolerance as set to 10 ppm for precursor ions and 0.02 Da for fragment ions (b and y ions only), dynamic modification was set for methionine oxidation (+15.995 Da) and N-terminus Acetylation (+42.011 Da), target FDR (strict minimum value 0.01), Delta Cn minimum value 0.05). Only proteins with more than 1 unique peptide were considered. LFQ intensity was calculated using the following parameters: pairwise ratios, maximum allowed fold change 100, ANOVA (background based). The technical replicates from each biological sample were pooled to perform diagnosis comparisons for the appendix and prefrontal cortex, using a non-nested test. Proteins were quantified using the pairwise peptide ratio information from extracted peptide ion intensities. Only proteins with abundances recorded in at least two of three samples per diagnosis group were considered. Proteins with absolute log fold change between groups exceeding 0.2 were considered as altered. Genes of altered proteins in the appendix dataset were intersected with those having at least one significantly differentially methylated cytosine in the PD appendix study. The resulting list of genes and their interactions were visualized using STRING-db version 11[141].

A replication cohort of human appendix tissue ($n = 5$ PD, 5 controls) was analyzed by the Whitehead Institute proteomics core facility using tandem mass tag (TMT) proteomics. Samples were prepared using the Minute Lysosome Isolation kit (Invent Biotechnologies) following kit directions with the following modifications: Approximately 35 mg of cryopulverized appendix tissue was dounce homogenized in 500 μL of buffer A with protease inhibitor cocktail (Roche). After initial filtration and centrifugation supernatant were centrifuged for 5 min at $11,000 \times g$. Remaining kit protocol steps were followed with the omission of the $2000 \times g$ spin before adding buffer B. Final lysosome pellet was resuspended in 50 μL PBS and quantified with BCA. Reduction, alkylation, proteolytic digestion, and isotopic labeling were carried out using Pierce TMT 10-plex (catalog number 90110) according to kit specifications. The resulting labeled peptides were washed, extracted and concentrated by solid-phase extraction using Waters Sep-Pak Plus C18 cartridges. Organic solvent was removed and the volumes were reduced to 80 μL using a speed vac for subsequent analyses. First dimension of chromatography fractionation of the labeled peptides was performed using Pierce High pH Reversed-Phase Fractionation Kit (catalog number 84868) according to the manufacturer's specifications. Each of the fractions were transferred to autosampler vials and reduced to a final volume of 20 μl by SpeedVac with eight fractions destined for subsequent analysis. These chromatographic fractions were analyzed by reversed-phase high performance liquid chromatography (HPLC) using Thermo EASY-nLC 1200 pumps and autosampler and a Thermo Exploris 480 Hybrid Quadrupole-Orbitrap mass spectrometer using a nanoflow configuration. Samples were loaded on a 2 cm × 75 μm Thermo Pepmap100 C18 trapping column and washed with 4 μL total volume to trap and wash peptides. These were then eluted onto the 15 cm × 75 μmThermo EASY-Spray C18 analytical column attached to a spray emitter with a 5 μm tip. The gradient initial condition was 1% A Buffer (1% formic acid in water) at 300 nl min⁻¹ with increasing B buffer (1% formic acid in acetonitrile) concentrations to 6% B at 1 min, 21% B at 42.5 min, 36% B at 63 min and 50% B at 73 min. The column was washed with high percent B and re-equilibrated between analytical runs for a total cycle time of approximately 97 min. The mass spectrometer was operated in a dependent data acquisition mode where the 12 most abundant peptides detected in the Exploris using full scan mode with a resolution of 120,000 were subjected to daughter ion fragmentation using a resolution of 60,000. A running list of parent ions was tabulated to an exclusion list to increase the number of peptides analyzed throughout the chromatographic run.

The raw mass spectrometry data were searched using PEAKS Studio (Bioinformatics Solutions Inc., Waterloo, ON, Canada, version 10.5) against a combined protein database of Refseq human entries and common MS contaminants. De novo sequencing of peptides, database search and characterizing specific PTMs were used to analyze the raw data; false discovery rate (FDR) was set to ≤0.5%, and $[-10 \times \log(P)]$ was calculated accordingly. The search parameters included a maximum of two missed cleavages; carbamidomethylation at cysteine and TMT10plex as fixed modifications with oxidation at methionine as a variable modification. The precursor tolerance was set to 10 ppm and MS/MS tolerance to ±0.05 Da for both de novo and database searches. Purity correction factors for isotopic distribution of reporter ions was incorporated in the quantification. Only proteins with more than one unique peptide were considered further. Measured abundance values were log transformed and standardized across each sample. Robust linear regression was used to determine the differences between PD cases and controls by adjusting for sample sex, age, and postmortem interval. Proteins with FDR adjusted $q < 0.05$ were considered differentially abundant.

**Age effect analysis**. Identification of ALP genes consistently altered in the healthy appendix and prefrontal cortex was performed using the enrichment $p$-value of ALP genes with differentially methylated cytosines and robust rank aggregation algorithm[54] approach described above. To determine whether the cytosine sites at ALP genes that were epigenetically altered with age were similarly altered in PD, a Pearson correlation of absolute age effects and absolute PD effects was computed using loci that were nominally significant in both datasets. We also compared epigenetic age effect patterns of the ALP genes in the healthy and PD appendix and prefrontal cortex. Summarized age effect trend for each gene was computed as weighted mean age effect of all cytosines pertaining to the gene where log transformed aging $p$ value was used as weight. The age effect trend was set to zero for the ALP genes that did not have a single cytosine significantly affected by age (FDR $q < 0.05$).

**Experimental gut inflammation in wild-type and PD mouse model**. We profiled epigenetic changes in the ALP of mice that were exposed to experimental colitis. We examined wild-type mice and a mouse model of synucleinopathy, hemizygous Tg(Thy1-SNCA*A30P)18Pjk mice (A30P α-syn)[142]. A30P α-syn mice express human α-syn with the A30P mutation under the neuron selective Thy1 promoter[142]. A30P α-syn mice have been maintained on a C57BL/6 background for more than 10 generations. Colitis was induced using dextran sodium sulfate (DSS), a widely used model of ulcerative colitis[71–73]. At 3 months of age, wild-type and A30P α-syn mice were exposed to a chronic DSS protocol, consisting of 4 cycles of increasing DSS concentration of 2.5%, to 3.0%, to 3.5%, to 4%. One cycle comprised of 5 days of DSS followed by 2 days of water (DSS: 160110, MP Biomedicals, LLC). Non-DSS groups were administered normal drinking water. Mice were then given a 4-week long recovery period during which they received normal drinking water, followed by tissue harvest. Mice were anesthetized with pentobarbital and trans-cardially perfused with PBS before tissue collection. Cecal patch tissue was snap frozen and stored at −80 °C until processing for DNA methylation analysis. A 1-cm-part of colon proximal to the cecum was post-fixed 24 h in 4% PFA followed by 30% sucrose in PBS until processing for paraffin embedding. Inflammation and tissue integrity was analyzed in 10 μm thick paraffin sections processed for hematoxylin/eosin staining. Approximately equal numbers of males and females were included in each genotype (wild-type, A30P α-syn) and treatment (water, DSS) group. To the extent possible, littermates were used in the experiments. The mice were generated and kept in filter cover cages under normal housing conditions with a 12 h dark/light cycle (light from 6am to 6 pm), constant ambient temperature at 22 °C (±2 °C) and air humidity ranging from 40 to 60%. The animal experiments complied with all relevant regulations for animal testing and research and were endorsed by a Roche internal review board and approved by the local animal welfare authorities in Canton Basel-Stadt, Basel, Switzerland.

**α-syn overexpression by viral vector in the mouse cecal patch**. We investigated epigenetic changes in the ALP in response to α-syn aggregation in the mouse cecal patch. α-syn aggregation in neurons of the mouse cecal patch was induced by recombinant AAV (rAAV 2/9) vector-mediated overexpression of full-length human α-syn or GFP as control, using a hybrid CMV/CBA promoter[74]. The vectors were generated as previously described[143]. To generate the vectors, plasmid transfection was performed in HEK 293 T cells, followed 72 h later by cell harvest, media collection, and concentration by tangential flow filtration. Viral particle purification was performed using an iodixanol gradient, followed by column chromatography. Titer was determined by digital droplet PCR by serially diluting the virus 1:500,000 and using the Biorad ddPCR system according to the manufacturer's instructions. The following oligonucleotides were used with an annealing temperature of 62 °C: Forward primer: 5′ CGG CCT CAG TGA GCG A 3′, reverse primer: 5′ GGA ACC CCT AGT GAT GGA GTT. To calculate the concentration of the vector batch, we multiplied the genomes/μL determined by the droplet reader by total reaction volume and divided this number by the volume of virus that was included in the reaction mixture, then adjusted for the dilution (x500,000) and normally used vector genomes/mL (x1000)[144].

Injections of the rAAV 2/9 α-syn and GFP control vector were performed on male C57BL/6 J mice obtained from Jackson Labs. Animals were housed in a 12 h light (on 6am to 6 pm). Ambient temperature is 69.9 to 73 F with a humidity ranging from 0 to 52%. At 8 weeks of age, mice were anesthetized using 2% isoflurane and a full laparotomy was performed to expose the cecal patch on the cecum. Each mouse cecal patch region received two 2 μl injections at a titer of $6 \times 10^{12}$ vg/ml of rAAV α-syn ($n = 10$) or control vector ($n = 10$). Injections were performed with an automated micropump at 1 μl/s. Following the injection, the needle was left in place for an additional 10–20 s to allow the injection pressure to dissipate. All mice administered rAAV-α-syn or control vectors to the cecal patch fully recovered from surgery, consistent with prior studies[74,145]. Following surgery, mice remained in their home cages for a 4-week-long incubation period, followed by tissue harvest. Digital droplet PCR analysis of rAAV genomes[144] confirmed the successful rAAV injection into the mouse cecal patch (Supplementary Fig. 13C). We used 5 rAAV-α-syn and 5 control mice for DNA methylation analysis at ALP genes and used 5 rAAV-α-syn and 5 control mice for immunohistochemistry analysis. The experiment using rAAV vectors in mice was performed in accordance and with the ethical approval of the Michigan State University Institutional Animal Care & Use Committee guidelines (AUF 08-16-148-00).

We confirmed that rAAV-mediated α-syn overexpression induces α-syn aggregation in the mouse cecal patch, as observed for other intestinal regions[74,143,146]. Mouse cecal patch sections were washed in TBS containing 0.5% Triton X-100, incubated in 3% peroxide solution and blocked in 10% normal goat serum. Neurons within enteric ganglia were identified using the pan-neuronal marker HuC/D (1:2000; Thermo Fisher Scientific). Sections were also probed for α-syn, using a pan-α-synuclein antibody (1:1000; BD Biosciences), and for phospho-serine 129 positive α-syn aggregates (1:10,000; 81 A, Abcam). Secondary antibodies used were goat anti-mouse IgG2a Alexa Fluor 488 (1:500; Thermo Fisher Scientific), goat anti-mouse IgG1 Alexa Fluor 488 (1:500; Thermo Fisher Scientific), goat anti-mouse IgG2b Alexa Fluor 594 (1:500; Thermo Fisher Scientific). Sections were coverslipped using Vectashield hardset mounting medium (VectorLabs).

**Mice datasets comparisons**. We tested whether the A30P α-syn mice have stronger epigenetic responses to DSS than wild-type mice. In this analysis, we compared the fold changes of differentially methylated cytosines in wild-type and/or A30P α-syn mice exposed to DSS. The odds of observing greater fold change in A30P α-syn mice (relative to wild-types) among the differentially methylated sites were compared to that of the non-significant loci. Fisher's exact test was used to compute the $p$-value. Next, we examined the concordance of DNA methylation changes at ALP genes between the mice groups and between the PD appendix and mice studies. In mice experiments, the enrichment of each ALP gene with significant differentially methylated cytosines was computed using Fisher's exact test as described above. Kendall's rank correlation coefficient, which does not make assumptions about the distribution of resulting odds ratios, was used to evaluate concordance between epigenetically altered ALP genes in the rAAV α-syn mice and the mouse groups in the DSS study. Similarly, odds ratios of gene enrichment with differentially methylated cytosines were computed for PD appendix dataset, and the odds ratios were compared with those of the homologous genes in the mouse studies using Kendall's rank correlation coefficient to determine concordance of epigenetically altered ALP genes in the PD appendix and mouse studies. Pathway enrichment in the mouse studies was performed by first identifying the genes having more than one differentially methylated cytosine. The genes from each mouse study were then intersected with corresponding human genes that had more than one differentially methylated cytosine in the PD appendix. Enrichment within human ALP pathways was determined using Fisher's exact test. Likewise, the genes from the rAAV α-syn study were intersected with the genes of each mouse comparison group in the DSS study, and their enrichment within mouse ALP pathways was tested using Fisher's exact test. Gene lists pertaining to the mouse ALP pathways were downloaded from Mouse Genome Informatics website (http://www.informatics.jax.org) using the GO accession numbers matched to the human pathway analysis. ALP genes differentially methylated in appendix were overlapped with those differentially methylated in synucleinopathy or DSS colitis. The genes differentially methylated in either rAAV a-syn mice or between A30P and wild-type mice treated with water were deemed as affected by synucleinopathy. Similarly, genes affected by DSS colitis were those that showed difference between DSS and water treatments in either A30P or wild-type mice. As before, the dominant direction of methylation for a gene was determined by majority significantly affected cytosines with ties resolved by weighted mean fold change.

**Reporting summary**. Further information on research design is available in the Nature Research Reporting Summary linked to this article.

## Data availability

All sequencing data generated in this study are freely available from the NCBI Gene Expression Omnibus (GEO) database under the accession number GSE135751. The mass spectrometry proteomics data have been deposited to the ProteomeXchange Consortium via the PRIDE[147] partner repository with the dataset identifiers PXD015079 and PXD021757.

## Code availability

Custom code needed for reproduction of all reported statistical results and figures pertaining to DNA methylation, RNA-seq and proteomics analysis is available at http://www.vugene.eu/VAI/pdappendixalp and through Zenodo https://doi.org/10.5281/zenodo.5059957.

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

## Acknowledgements
The authors thank Farmer Family Foundation, the Van Andel Research Institute Flow Cytometry, Genomics, Bioinformatics and Biostatistics, and Pathology Cores. We also thank the CAMH Sequencing Facility. We thank all organ donors and their families for their support of our research. We thank the Oregon Brain Bank, Spectrum Health Universal Biorepository, Parkinson's UK Brain Bank, the NIH NeuroBioBank, and the Michigan Brain Bank for the tissue provided. We are also grateful for the proteomics services provided by Dr. Eric Spooner at the Whitehead Institute Proteomics Core Facility. We appreciate the manuscript review and recommendations provided by Dr. Patrik Brundin. We thank L. Spycher and C. Zundel at Roche for their technical support with the DSS colitis paradigm and tissue processing. N.M. was funded by the Roche Postdoctoral Fellowship Program. Data were generated as part of the PsychENCODE Consortium supported by: U01MH103339, U01MH103365, U01MH103392, U01MH103340, U01MH103346, R01MH105472, R01MH094714, R01MH105898, R21MH102791, R21MH105881, R21MH103877, and P50MH106934 awarded to: Schahram Akbarian (Icahn School of Medicine at Mount Sinai), Gregory Crawford (Duke), Stella Dracheva (Icahn School of Medicine at Mount Sinai), Peggy Farnham (USC), Mark Gerstein (Yale), Daniel Geschwind (UCLA), Thomas M. Hyde (LIBD), Andrew Jaffe (LIBD), James A. Knowles (USC), Chunyu Liu (UIC), Dalila Pinto (Icahn School of Medicine at Mount Sinai), Nenad Sestan (Yale), Pamela Sklar (Icahn School of Medicine at Mount Sinai), Matthew State (UCSF), Patrick Sullivan (UNC), Flora Vaccarino (Yale), Sherman Weissman (Yale), Kevin White (UChicago) and Peter Zandi (JHU). VL is supported by grants from the Department of Defense (W81XWH1810512), the National Institute of Neurological Disorders and Stroke (1R21NS112614-01), and a Gibby & Friends vs. Parky Award. Vector work (FPM) was supported by a grant from the National Institute of Diabetes and Digestive and Kidney Diseases (5R01DK108798).

## Author contributions
J.G. contributed to experimental design and computational analyses. P.L. was involved in DNA methylation sequencing data preprocessing and RNA-seq analysis. L.M. designed the bisulfite padlock probes in humans and mice and contributed to DNA methylation data preprocessing. BK isolated nucleic acids from appendix, performed neuronal nuclei isolations from prefrontal cortex, and was involved in the bisulfite padlock probe library preparation. E.E. and W.C. contributed to bisulfite padlock probe library preparation. N.M.K. and F.P.M. contributed to the experiments involving rAAV α-syn overexpression in mice. J.L. and I.V. performed the mass spectrometry proteomic analyses. N.M., R.L., C.R., J.S.P., P.M., and M.B. were involved in the gut inflammation mouse model. V.L. was involved with study design and overseeing the experiments. The manuscript was written by V.L. and S.L. and commented on by all authors.

## Ethics approval
For the human tissue studies, the study protocol was ethically approved by the institutional review board at the Van Andel Research Institute (IRB #15025). For the gut inflammation mouse model, the animal experiments were endorsed by a Roche internal review board and approved by the local animal welfare authorities in Canton Basel-Stadt, Basel, Switzerland. The experiment using rAAV vectors in mice was performed in accordance and with ethical approval of the Michigan State University Institutional Animal Care & Use Committee guidelines (AUF 08-16-148-00).

## Competing interests
N.M., R.L., C.R., J.S.P., P.M., and M.B. are or were full-time employees at Roche and they may additionally hold Roche stock/stock options. No other authors have conflicts of interest.
