## [Peer Review File · Nature Communications]

Epigenetic inactivation of the autophagy–lysosomal system in appendix in Parkinson’s diseaseREVIEWER COMMENTS:

Reviewer #1 (Remarks to the Author):

The manuscript “Epigenetic inactivation of the autophagy–lysosomal system in the Parkinson's disease appendix” presents a comprehensive study of the methylation status of DNA in human appendices, olfactory bulbs and isolated neuronal nuclei from prefrontal cortex by comparing Parkinson's disease (PD) patients and controls. The methylation patterns associated to normal aging is also characterised and compared to the PD patterns. The functional relevance of hyper vs hypomethylation on mRNA expression and protein abundance is investigated in a limited semiquantitative proteomic analysis. Finally, the role of intestinal inflammation to the changes observed in rodent models of alpha-synuclein (AS) expression is investigated.

The main message that is carried by the paper is that epigenetic hypermethylation occurs in genes associated to the autophagic-lysosomal pathway (ALP) in PD appendix and CNS. These changes are present in neurons of the prefrontal cortex before apparent Lewy body pathology occurs suggesting they represents early effects. Additionally, a hypomethylation of the AS expressing SNCA gene occurs that may increase AS expression.

The results are exciting but some points need to be addressed to fully substantiate the conclusions

Major points:

1) The functional effect of the methylation patterns is investigated by transcriptomic analysis of appendix. To validate they also occur at the protein levels, a proteomic analysis is conducted in 3 samples of detergent extracts of appendix tissue from PD and ctl. The data are considered valid if 2 of 3 samples are suitable. These numbers are low. Moreover, there is not performed any real quantitative comparison using e.g. SILAC-like technology. They will have to validate their MS-based semi-quantitative results using Western blotting both for the ALP candidate proteins but also for AS that they hypothesise is upregulated.

2) The animal studies focuss on the relation of gut inflammation and DNA methylation. Here is observed a general hypomethylation in contrast to the general hypermethylation in PD. The authors need to discuss their models better. The A30P-AS transgenic mice may be suboptimal as this mutation changes the vesicle binding of AS and thereby may affect other pathways than aggregation as hypothesised in the paper. The other model using AAV mediated overexpression of AS in the mouse cecal paths (equivalent of human appendix) does not convincingly demonstrate aggregation of AS although this is stated in the text. Fig S13 demonstrates staining of total AS and pS129-AS. The pS129 is used as a proxy for aggregated AS. The total AS demonstrates AS in beaded structures arranged along neurite or axon-like structures but also larger positive cells. These cells may well be non-neurons. The pS129 staining occurs predominantly in such larger structures and are not convincingly localised in neuronal structures. The AAV vectors used for expression of AS is likely not neuron specific. Hence the effects of AS expression cannot be associated to aggregation and not even to neuronal AS expression. An AAV vector using a neuron specific promoter may help solve the issue or double immunofluorescence microscopy should be employed to prove the AS expression is restricted to neurons. As presented, data does not demonstrate a correlation to aggregation but only AS expression.

Interesting point that may be addressed:

In the Discussion, the potential as biomarkers are presented “epigenetic changes in the ALP in the appendix may be a potential biomarker for disease risk and progression by serving as a proxy for the epigenetic status of ALP genes in the brain”. It will be interesting to determine if such changes also occur on other sites in the intestinal tracts that are amenable for biopsy as this will improve their significance as biomarkers.

Reviewer #2 (Remarks to the Author):

The paper by Gordevicius et al., performs a tour de force analysis of epigenetic and expression changes in the appendix olfactory bulb and prefrontal cortex in PD patients. Using padlock bisulfite sequencing the authors measure at high resolution at 521 genes belonging to the autophagy lysosome pathway (ALP) as well as genes associated with PD by GWAS studies. They show changes in methylation in 326 of these genes. Changes are enriched in lysosomal genes in promoters and enhancers. Examination of olfactory bulb and a replicate study in the prefrontal cortex shows parallel changes in DNA methylation in the brain. Transcriptome analysis of PD and control appendix shows downregulation of ALP genes and enrichment in lysosomal genes. Proteomic analysis shows as well downregulation of ALP genes as well as downregulation of several genes that are epigenetically regulated. The authors show that many of the genes changed in PD change also in aging however there is no aging effect in PD. The authors further examine the involvement of ALP genes in PD using a mouse model expressing α -synuclein and combine it with a DSS colitis model. They show that DSS colitis triggers differential methylation of ALP genes and that this effect is enhanced in α -synuclein transgenics. There is an overlap between genes that are epigenetically programmed in the transgenic mouse overexpressing α -synuclein as well as mice infected in the cecal patch by an AAV overexpressing α -synuclein. However, while there is an overlap between ALP genes epigenetically programmed by overexpression of α -synuclein in mice and PD, the epigenetic program by DSS does not resemble the epigenetic changes in ALP genes in PD appendix. While DSS triggers mainly hypomethylation of ALP genes, in the PD appendix the response is predominantly hypermethylation. Overall, this paper provides strong evidence for involvement of epigenetic reprogramming of ALP genes in the gut and the brain in PD and provides a plausible mechanism.

This is an extremely important discovery that expands our understanding of the mechanisms involved in PD and provides a strong argument for involvement of epigenetic programming of ALP genes in the disease. The authors use a very robust method to examine the DNA methylation state of a large group of functionally linked genes at high resolution, the results are extremely convincing and provide evidence for reprogramming of lysosomal functions in PD. The overlap between the appendix and the brain and the replication in two prefrontal cortex samples coupled with evidence from proteomic and transcriptomic analyses as well as an animal model overexpressing α -synuclein provides compelling evidence for the involvement of epigenetic reprogramming of ALP genes in the molecular pathology of PD. There are however several issues that I believe could be clarified in a revised version of this paper.

- a. The authors claim that the main response in PD is hypermethylation. Indeed, although the majority of DNA methylation changes in PD ALP genes is hypermethylation, a large fraction of the genes are hypomethylated. What is the role of hypomethylation in regulating ALP genes? This shouldn't be ignored.
- b. The authors show that there is an overall silencing of ALP gene expression in PD their transcriptomic analysis, the correlation with methylation is significant but quite weak only 0.2. This suggests that for many ALP genes expression is reprogrammed by mechanisms that do not involve DNA methylation while other genes are differentially methylated with no effect on expression. What is the overlap between DNA methylation and expression? What do we know about those genes that are downregulated by mechanisms that don't involve DNA methylation while other genes in the same functional family seem to be regulated by DNA methylation? What is the role of altered DNA methylation that is not accompanied by changes in gene expression?
- c. The analysis reveals overall overlap in DNA methylation in genes between PD appendix and brain regions. Are the same sites methylated in both tissues? What is the overlap at the CG level?
- d. Correlation of protein abundance in brain and appendix is 0.26. 34 proteins are epigenetically altered in appendix, what about the state of expression of these proteins in brain? What fraction of the ALP proteins that are altered are also epigenetically regulated?
- e. Out of the ALP proteins that are concordant between brain and appendix, how many are epigenetically regulated?
- f. Proteins that are regulated by DNA methylation as well as those that are not should be discussed to appreciate the impact of epigenetic regulation on downregulation of ALP proteins.
- g. The authors show concordance between genes that are epigenetically altered in aging and those altered in PD. It would be expected therefore that aging is accelerated in PD. But it seems that there

is no effect of aging on ALP genes in PD. How is this possible? Do the ALP genes reach higher level of methylation earlier than in healthy patients? This should then be illustrated by a figure showing this in several gene methylation map examples.

h. The authors do not provide any gene maps with CG positions and their rate of change in methylation in PD and normal aging?

i. How large are the changes in methylation throughout the paper? The authors provide numbers of genes that change significantly but not the scope and range of changes.

j. The authors show that a-synuclein over expression in mice exacerbates the changes in ALP DNA methylation that occur during chronic inflammation. These changes are predominantly hypomethylation. They also show that the changes induced by DDS inflammation do not correspond to changes in methylation seen in PD patients and that they take an opposite direction, hypomethylation in DDS inflammatory response versus hypermethylation in PD. They show on the other hand that a-synuclein over expression triggers changes in methylation that are concurrent with the changes seen in PD. How is this possible? A-synuclein over expression =PD, a-synuclein =DDS, then one would expect that DDS=PD. Are we talking about one set of genes that are targeted by a-synuclein and DDS that is hypomethylated and then a totally different set of genes that is hypermethylated by a-synuclein and is also hypermethylated in PD? If this is the case, it might suggest a disconnect between the effects of DDS-inflammation and the link between a-synuclein and PD which is different from the model proposed by the authors. This needs to be discussed and the identity of the genes that are either hypomethylated or hypermethylated in DDS, a-synuclein over expression and PD should be revealed and perhaps presented and illustrated.

In summary, the epigenetic alterations in ALP genes in PD are robust and convincing but the effects of aging and DDS mediated inflammation are confusing and contradictory. These need to be analyzed in detail and explained more clearly. Alternatively, the paper could focus on PD and a synuclein overexpression in the mouse and the potential mechanistic implications of deregulation of ALP genes in the molecular pathology of PD.

Reviewer #3 (Remarks to the Author):

The authors presented an interesting and well conducted study in which they produced a comprehensive set epigenetic, transcriptomic and proteomic data from human and mouse tissues. They suggest that methylation and transcriptome differences in samples taken from the human appendix and brain (olfactory bulb and prefrontal cortex neurons) distinguish Parkinson's patients from controls in terms of methylation for a preselected number of genes contributing to the autophagy-lysosomal pathway (ALP) as well as some select known risk genes for PD (GWAS derived). Based on their data, they conclude that APL dysregulation through methylation and differential expression of these genes may play a key role in the pathogenesis of PD. This conclusion is generally in concordance with the current existing knowledge about the importance and the role of autophagy in neurodegenerative disorders especially Parkinson's disease.

This study was hypothesis driven with the ALP as the target. The ALP is considered an important pathway in PD (due to GWAS results, and also because GBA - a Mendelian gene - contributes to familial PD) that when disrupted may cause the accumulation of aggregation-prone α -syn, a hallmark protein that contributes to Lewy-bodies in PD. While this is not new as a hypothesis in general, the hypothesis and the tissues (appendix together with the olfactory bulb and brain (PFC)) these authors chose to evaluate together are very novel and interesting as the gut has only recently been implicated as a possible origin/source of α -syn that would be transported into the brain retrogradely. Also, α -syn aggregates were found in healthy humans and in PD in the appendix and if these protein aggregates are responsible for seeding brain aggregates, it is of great interest to determine what makes the difference between those who remain free of PD and those who eventually develop the disorder i.e. to find avenues for early intervention and prevention.

To determine the interaction of the ALP molecular system and α -syn aggregates in the gut and brain,

the authors used human samples and assessed local tissue pathologies i.e. comparing samples from people with and without PD presumably at one point in time at autopsy. In the text it is hard to tell whether and the authors need to spell out more clearly at what point in time these samples were taken. It seems they all were derived from autopsies? However, is this also true for appendix at younger ages or whether these appendices from surgeries in living individuals. Also, it is not clear whether some of the tissues from multiple organs are recovered from the same person or whether they all were derived from different donors postmortem? While the age at sampling and sex are reported, for PD patients not only the Braak stage but also the duration of disease in years since diagnosis or onset would be interesting to know if such information were available. Similarly, for control tissues it would be interesting to know whether the donors had died of an accident or some disease and what the predominant cause of deaths were - especially for the appendix samples; e.g. are these individuals all free of colitis etc..

The ultimate question - and the authors clearly state this - is whether there are processes such as transport of α -syn from the gut to the brain or vice versa to which and how the ALP and inflammation may contribute. Nevertheless, the authors should state more clearly that the direction cannot be determined in this study except maybe indirectly. Similarly, these tissues were necessarily taken from PD patients long after the onset of disease - which opens up the possibility that there is reverse causation i.e. that the localized changes in the gut methylome represent consequences and not causes of a disease known to affect the parasympathetic nervous system of the gut with constipation a known non-motor symptom; i.e. the question of what mechanisms underly the development of α -syn aggregates in the gut or what triggers its spread to the brain cannot necessarily be answered here except indirectly. The authors acknowledge this by stating that "Our human studies do not discern whether epigenetic changes at ALP genes are causal to PD or a consequence of PD pathophysiology" but there are many passages in which a presumed direction is implied without the necessary qualification. Indeed, figure 6 is most helpful to illustrate these points i.e. that we are seeing more or less a snapshot in time for multiple feedback loops between inflammation, ALP (dys)function, and α -syn aggregates in the gut and brain. I would like to ask the authors to consider pulling this figure up in the manuscript to the front to help the reader orient him/herself and also to carefully review the manuscript for wording about directionality such that this ambiguity about direction that implies causes is always clear to the reader. None of this diminishes the results, however, and the completeness of the story the authors are telling overall i.e. that there is widespread hypermethylation in the ALP of PD patient compared to controls in multiple tissues of interest in PD and in a number of overlapping genes across the interrogated tissues and that some of the genes affected are linked to immune function.

A second point the authors are trying to make is about the relevance of aging processes. To do so they investigate whether epigenetic perturbation of the ALP are similar to or related to the known epigenetic DNA remodeling that occurs with aging, as aging is the most consistent risk factor for PD, may be contributing to the impaired clearance, accumulation, and spread of aggregated α -syn in PD, and has a known effect on DNA methylation. Since they used targeted fine mapping in 521 preselected ALP genes and some well-known PD risk genes, however, they cannot assess epigenetic aging of the tissues overall but only age-related changes in this targeted system. In healthy controls they found a clear shift towards hypermethylation of sites with increasing age in ALP genes in prefrontal cortex neurons. The authors interpreted this as an aging related degradation of the machinery that removes aggregated proteins and invading pathogens therefor aging making the appendix and brain more vulnerable to protein aggregation and infections. I agree that a less well functioning system in older age may increase vulnerability. Interestingly, they however did not find any age-related changes in DNA methylation (an overall directional shift at ALP genes) in the aging appendix or the PFC neurons in PD patients (Fig. 4). This is further supported by the predicted younger age of the ALP machinery in PD compared with controls. The authors interpret this lack of an 'aging effect' of the ALP in PD patients as the attempt of the system to deal with aggregated α -syn in PD by upregulating the ALP, even though this is obviously inadequate as the disease is characterized by more and more aggregated α -syn. On the other hand does this observation not also suggest that the ALP in PD PFC neurons is indeed still able to upregulate in response to disease processes and this would then speak against the hypothesis that it is the ALP that is failing and causing the

aggregation of α -syn etc even though it is part of the picture as it may not be able to upregulate enough to stem the disease process? Also, is the ALP of the neurons only of interest or should we perhaps also pay attention to the microglia that are not part of this analysis. Finally, I wondered why the authors omitted showing any results for the olfactory bulb for aging in PD and controls.

The authors found far more differentially methylated sites and genes in appendix and olfactory bulb than neurons of prefrontal cortex. I wonder whether this would not be expected as the type of cells is much more homogeneous i.e. only neurons while in the appendix or olfactory tissues we have immune cells mixed and epithelium etc as well etc. Thus, wouldn't we also expect some signal from immune cells in gut and nose that we do not expect to see in neurons? In fact it is a bit surprising that the neurons showed immune tissue relevant methylation changes (TLR9 and GPNMB). The authors mention that even the neurons of PD patients that did not have Lewy pathology (Braak stage 3-4) showed greatly overlapping changes in methylation, suggesting that many of these may precede the arrival of Lewy pathology and imply that this may mean that the ALP 'dysfunction' is at the root cause for the eventually pathology; however, this may not necessarily be the case as not having Lewy bodies does not mean that α -syn is not upregulated yet, rather it might be and the ALP machinery might also be reacting to this only it is possibly still capable of keeping the pathology at bay.

Third, the authors compared two synuclein pathology mouse models with wild type mice and also induced gut inflammation. One model are mice overexpressing A30P α -syn and for the other they used virally induced expression of α -syn (rAAV-mediated α -syn overexpression through injection into the cecal patch of wild-type C57BL/6J mice). The virus induced overexpression was accompanied by a differential methylation in a large number of ALP genes in response to α -syn aggregation while the mice overexpressing A30P α -syn did not show any differences in ALP gene methylation compared to wild type mice unless they had undergone an inflammatory challenge. There were also positive correlations and overlap in epigenetically altered ALP genes in response to inflammation and those altered in response to α -syn aggregation by rAAV-mediated overexpression. Also, differentially methylated ALP genes in the PD appendix had gene homologs that were differentially methylated in the mice overexpressing A30P α -syn relative to wild-type mice. However, methylation changes in the ALP induced by gut inflammation did not resemble those seen in the PD appendix. Indeed, the strong ALP hypermethylation in PD was contrary to the ALP hypomethylation mediated by gut inflammation in mice. Nevertheless, pathway analysis revealed that α -syn overexpression in the A30P mouse model recapitulates abnormalities in lysosome function in the PD appendix and affected autophagy. The authors therefore concluded that aberrant α -syn expression and accumulation may contribute to the epigenetic abnormalities in the ALP in the PD appendix and in turn may trigger a hypersensitivity to gut inflammatory events.

Given this speculation, wouldn't we expect to see an inflammatory state in the PD gut or appendix compared with controls? Did the authors have any measure of inflammation available for their appendix samples?

The authors mention that in mice that overexpress α -syn they see a heightened responsivity of the ALP to an inflammatory stimulus. They call this an 'exaggerated response to an inflammatory event as an attempt to overcome deficient ALP function mediated by α -syn accumulation'. However, it is unclear why they consider this response exaggerated and what this would mean for the tissue, i.e. is there a point where too much autophagy would be detrimental? Similarly, they state that "Our results also suggest that the PD appendix is not overtly inflamed, but rather the elevated levels of α -syn may prime for exaggerated and potentially pathological ALP responses to inflammatory triggers". Again, what would a 'pathologic response of the ALP' look like in terms of the tissue i.e. what negative consequences would an upregulation of autophagy have?

Additionally:

- The authors are presenting odds ratios (OR) as a 'measure of the magnitude of enrichment of dysregulation' however, this measure is hard to interpret; i.e. it seems easier to understand an x-fold increase or decrease in methylation over control samples. Maybe the authors could explain more carefully what they are comparing when using such an odds ratio? I.e. what the ratio measure is a

ratio of. Also it is unclear how or why this measure as presented in Additional File 8 would suggest an overlap of methylation in genes across tissues (I might misinterpret the numbers in this file as there is no legend for the column titles provided but there are also ORs listed as 0 – does this mean the model did not converge or the ratio was actually zero; for example for MAN2B1 the ORs reported are 0.74, 0, 1.2 and 7.4 that is they are very different and on both sides of the null value of 1, so it is unclear how this can be considered a consistent result for this gene across tissues).

- Consider showing some Venn diagrams for the number of overlapping changes in methylation of genes across tissues etc.

- It is unclear what the y-axis of figure 4D stands for i.e. what kind of measure is meant by an 'absolute PD effect'?

- Also figure 5e is uninformative and can be removed

- The authors state that "epigenetic changes in the ALP in the appendix may be a potential biomarker for disease risk and progression by serving as a proxy for the epigenetic status of ALP genes in the brain"; it is not clear how this tissue (appendix) could indeed be a useful biomarker though as it is unlikely that someone concerned about PD would undergo an appendectomy and it also is unclear at what stage of PD the appendix and its methylation status would indeed reflect changes in the brain. I suggest to reword or strike this sentence.

Beate Ritz

Reviewer #4 (Remarks to the Author):

In this manuscript, Gordevicius et al. performed an extensive study of DNA methylation of the autophagy-lysosome pathway (ALP) in the Parkinson's disease appendix. Through the targeted deep DNA methylation sequencing of a relatively large number of samples from the appendix and brain of PD and control subjects as well as the mouse equivalent of the human appendix of control, A30P alpha-synuclein mice with and without gut inflammation, the team identified the synchronized hypermethylation at the ALP in the appendix and brain in PD. Specifically, they discovered:

1) ~63% of the 521 ALP genes were subject to DNA methylation dysregulation in the appendix in PD versus control and such DNA methylation dysregulation was specific to lysosome genes and affected predominantly their active promoter regions.

2) ~68% and 11% of the 521 ALP genes were subject to DNA methylation dysregulation in the PD olfactory neurons and prefrontal cortex, respectively. And those gene signatures were enriched in the ALP genes subject to DNA methylation dysregulation in the appendix, suggesting the DNA methylation dysregulation is shared by the appendix and brain.

3) Protein abundance changes in the appendix between PD and control were highly positively correlated with those in the prefrontal cortex.

4) A large number of genes were also subject to DNA methylation changes in ageing in the healthy appendix and prefrontal cortex and these changes were highly correlated with those between PD and control, suggesting ageing is one big contributor of PD pathogenesis.

5) in vivo experiments showed that gut inflammation substantially induced DNA methylation dysregulation (hypomethylation) in the cecal patch (the mouse equivalent of the human appendix), which is different from the widespread hypermethylation in the PD appendix.

Overall, this is a compelling study and presents a very clear picture of epigenetic regulation of the ALP genes in the PD appendix and brain. I have only a few minor concerns listed below:

1) How do the lysosome genes subject to DNA methylation dysregulation in the PD appendix overlap those differentially expressed between PD and control in the PD appendix? The two sets of genes correspond to the same pathway but they may not strongly overlap with each other.

2) There are many gene signatures from the experiments carried out in the study and often only pairwise comparisons were performed. It will be beneficial to prepare a heatmap to show the methylation patterns of the 521 ALP genes under all the experiments.

3) The authors should compare their findings with those from the genome wide DNA methylation data in PD. I suspect that many findings here may not be significant at the genome-wide analysis.

Response to Reviews

Reviewer #1

The manuscript “Epigenetic inactivation of the autophagy–lysosomal system in the Parkinson’s disease appendix” presents a comprehensive study of the methylation status of DNA in human appendices, olfactory bulbs and isolated neuronal nuclei from prefrontal cortex by comparing Parkinsons disease (PD) patients and controls. The methylation patterns associated to normal aging is also characterised and compared to the PD patterns. The functional relevance of hyper vs hypomethylation on mRNA expression and protein abundance is investigated in a limited semiquantitative proteomic analysis. Finally, the role of intestinal inflammation to the changes observed in rodent models of alpha-synuclein (AS) expression is investigated.

The main message that is carried by the paper is that epigenetic hypermethylation occurs in genes associated to the autophagic-lysosomal pathway (ALP) in PD appendix and CNS. These changes are present in neurons of the prefrontal cortex before apparent Lewy body pathology

occurs suggesting they represents early effects. Additionally, a hypomethylation of the AS expressing SNCA gene occurs that may increase AS expression.

The results are exciting but some points need to be addressed to fully substantiate the conclusions

Reviewer 1, Comment 1: The functional effect of the methylation patterns is investigated by transcriptomic analysis of appendix. To validate they also occur at the protein levels, a proteomic analysis is conducted in 3 samples of detergent extracts of appendix tissue from PD and ctl. The data are considered valid if 2 of 3 samples are suitable. These numbers are low. Moreover, there is not performed any real quantitative comparison using e.g. SILAC-like technology. They will have to validate their MS-based semi-quantitative results using Western blotting both for the ALP candidate proteins but also for AS that they hypothesise is upregulated.

Response 1: We thank the reviewer for their enthusiasm for our study and for encouraging a revision that extends the proteomics data. Considering the numerous dysregulated ALP proteins identified in our study we validated our label-free proteomic findings by performing a replication proteomic study with a separate cohort of PD and controls. This replication study was performed using a tandem mass tag (TMT) proteomic approach. Both the label-free (LFQ) and the TMT method used in our study are quantitative proteomic approaches, which have been used to examine postmortem tissue with similar sample sizes¹⁻⁵ (whereas the SILAC method is generally used in vitro cell culture studies). In TMT lysosome-enriched dataset we identified 1659 proteins that included 22% of ALP genes interrogated in this study. We found that differentially abundant proteins in lysosome-enriched TMT data (N = 229, p < 0.05) were enriched among differentially abundant proteins in label-free data (OR = 1.76, p = 0.04; Fisher's exact test). The proteins previously identified as differentially abundant in label-free data had a concordant direction of change in TMT data (r = 0.34, p = 5.43 x 10⁻¹⁶; weighted Pearson correlation). Overall, we conclude that the new dataset confirms the findings of the label-free approach despite technological and sample preparation differences. In addition, we have previously shown by immunoblotting that insoluble alpha-synuclein protein levels are significantly increased in the PD appendix⁶. Overall, our epigenetic, transcriptomic and two quantitative proteomics analyses demonstrate a dysregulation of the ALP in the PD appendix.

The following changes were made to the manuscript:

Results, pg. 7: In addition, we performed a replication study using a tandem mass tag (TMT) quantitative proteomic analysis to further validate our findings in a lysosome-enriched fraction of appendix tissue (n=5 control, 5 PD). We identified 2084 proteins including 132 (27%) of the proteins encoded by ALP genes interrogated in this study (Supplementary data 10). We found 175 differentially abundant proteins (FDR q < 0.05, robust linear regression), including 7 ALP proteins encoded by genes that were also epigenetically dysregulated in the human appendix. The overlap of differentially abundant proteins identified in both datasets was higher than

expected by chance (OR = 1.88, $p = 0.04$; Fisher's exact test), and the proteins identified as differentially abundant in label-free approach had a concordant fold change in the replication dataset ($r = 0.40$, $p = 5.99 \times 10^{-25}$; weighted Pearson correlation). Upregulation of SNCA and downregulation of NAMPT, HSPA8, GPNMB and VPS35 was confirmed by fold change estimates in the replicate data. Overall, this analysis confirms the validity of our previous findings.

Methods, pg. 26. A replication cohort of human appendix tissue ($n=5$ PD, 5 controls) was analyzed by the Whitehead Institute proteomics core facility using tandem mass tag (TMT) proteomics. Samples were prepared using the Minute Lysosome Isolation kit (Invitrogen Biotechnologies) following kit directions with the following modifications: Approximately 35 mg of cryopulverized appendix tissue was dounce homogenized in 500 μ L of buffer A with protease inhibitor cocktail (Roche). After initial filtration and centrifugation supernatant was centrifuged for 5 min at 11,000 X g. Remaining kit protocol steps were followed with the omission of the 2000 X g spin before adding buffer B. Final lysosome pellet was resuspended in 50 μ L PBS and quantified with BCA. Reduction, alkylation, proteolytic digestion and isotopic labeling were carried out using Pierce TMT 10-plex (catalog number 90110) according to kit specifications. The resulting labeled peptides were washed, extracted and concentrated by solid phase extraction using Waters Sep-Pak Plus C18 cartridges. Organic solvent was removed and the volumes were reduced to 80 μ L using a speed vac for subsequent analyses. First dimension of chromatography fractionation of the labeled peptides was performed using Pierce High pH Reversed-Phase Fractionation Kit (catalog number 84868) according to manufacturer's specifications. Each of the fractions were transferred to autosampler vials and reduced to a final volume of 20 μ L by SpeedVac with eight fractions destined for subsequent analysis. These chromatographic fractions were analyzed by reversed phase high performance liquid chromatography (HPLC) using Thermo EASY-nLC 1200 pumps and autosampler and a Thermo Exploris 480 Hybrid Quadrupole-Orbitrap mass spectrometer using a nano flow configuration. Samples were loaded on a 2 cm x 75 micron Thermo Pepmap100 C18 trapping column and washed with 4 μ L total volume to trap and wash peptides. These were then eluted onto the 15cm x 75 μ m Thermo EASY-Spray C18 analytical column attached to a spray emitter with a 5 micron tip. The gradient initial condition was 1% A Buffer (1% formic acid in water) at 300 nl min^{-1} with increasing B buffer (1% formic acid in acetonitrile) concentrations to 6% B at 1 minute, 21% B at 42.5 minutes, 36% B at 63 minutes and 50% B at 73 minutes. The column was washed with high percent B and re-equilibrated between analytical runs for a total cycle time of approximately 97 minutes. The mass spectrometer was operated in a dependent data acquisition mode where the 12 most abundant peptides detected in the Exploris using full scan mode with a resolution of 120,000 were subjected to daughter ion fragmentation using a resolution of 60,000. A running list of parent ions was tabulated to an exclusion list to increase the number of peptides analyzed throughout the chromatographic run.

The raw mass spectrometry data were searched using PEAKS Studio (Bioinformatics Solutions Inc., Waterloo, ON, Canada, version 10.5) against a combined protein database of Refseq human entries and common MS contaminants. De novo sequencing of peptides, database

search and characterizing specific PTMs were used to analyze the raw data; false discovery rate (FDR) was set to $\leq 0.5\%$, and $[-10 \cdot \log(P)]$ was calculated accordingly. The search parameters included a maximum of two missed cleavages; carbamidomethylation at cysteine and TMT10plex as fixed modifications with oxidation at methionine as a variable modification. Precursor tolerance was set to 10 ppm and MS/MS tolerance to ± 0.05 Da for both de novo and database searches. Purity correction factors for isotopic distribution of reporter ions was incorporated in the quantification. Only proteins with more than one unique peptide were considered further. Measured abundance values were log transformed and standardized across each sample. Robust linear regression was used to determine the differences between PD cases and controls by adjusting for sample sex, age, and postmortem interval. Proteins with FDR adjusted $q < 0.05$ were considered differentially abundant.

Supplementary Figure S1:

Supplementary Data 10 (.csv): TMT quantitative proteomic analysis of the PD appendix.

References

1. Carlyle BC, *et al.* A multiregional proteomic survey of the postnatal human brain. *Nat Neurosci* **20**, 1787-1795 (2017).
2. Li KW, Ganz AB, Smit AB. Proteomics of neurodegenerative diseases: analysis of human post-mortem brain. *J Neurochem* **151**, 435-445 (2019).
3. Pai S, *et al.* Differential methylation of enhancer at IGF2 is associated with abnormal dopamine synthesis in major psychosis. *Nat Commun* **10**, 2046 (2019).

4. Roy M, *et al.* Proteomic analysis of postsynaptic proteins in regions of the human neocortex. *Nat Neurosci* **21**, 130-138 (2018).
 5. Unwin RD, Griffiths JR, Whetton AD. Simultaneous analysis of relative protein expression levels across multiple samples using iTRAQ isobaric tags with 2D nano LC-MS/MS. *Nat Protoc* **5**, 1574-1582 (2010).
 6. Killinger BA, *et al.* The vermiform appendix impacts the risk of developing Parkinson's disease. *Sci Transl Med* **10**, (2018).
-

Reviewer 1, Comment 2: The animal studies focuses on the relation of gut inflammation and DNA methylation. Here is observed a general hypomethylation in contrast to the general hypermethylation in PD. The authors need to discuss their models better. The A30P-AS transgenic mice may be suboptimal as this mutation changes the vesicle binding of AS and thereby may affect other pathways than aggregation as hypothesised in the paper. The other model using AAV mediated overexpression of AS in the mouse cecal paths (equivalent of human appendix) does not convincingly demonstrate aggregation of AS although this is stated in the text. Fig S13 demonstrates staining of total AS and pS129-AS. The pS129 is used as a proxy for aggregated AS. The total AS demonstrates AS in beaded structures arranged along neurite or axon-like structures but also larger positive cells. These cells may well be non-neurons. The pS129 staining occurs predominantly in such larger structures and are not convincingly localised in neuronal structures. The AAV vectors used for expression of AS is likely not neuron specific. Hence the effects of AS expression cannot be associated to aggregation and not even to neuronal AS expression. An AAV vector using a neuron specific promoter may help solve the issue or double immunofluorescence microscopy should be employed to prove the AS expression is restricted to neurons. As presented, data does not demonstrate a correlation to aggregation but only AS expression.

Response 2: We agree with the reviewer on improving the presentation of our mouse models. In mouse models of synucleinopathy, we sought to examine whether there was an epigenetic dysregulation of the ALP the mouse appendix (the cecal patch) that was similar to that observed in the PD appendix. The A30P mouse model of synucleinopathy involves the overexpression of human A30P mutated α -syn under the neuronal cell type-selective Thy1-promoter. We are aware of the limitation of this and other transgenic models and agree that the A30P mutation has been implicated with stronger lipid membrane interaction. This is in contrast to human α -syn and murine α -syn. The latter coincidentally contains the A53T sequence, which notably is pathogenic in humans and leads as human A53T α -syn to much reduced binding to membranes and more aggregation of the recombinant protein in vitro compared to human wild type α -syn. To complement the findings in the inflammation models, we therefore independently employed the rAAV viral vector model to overexpressed human wild type α -syn in the cecal patch. This was accomplished via targeted subserosal delivery directly into the gut wall. As mentioned by the reviewer, rAAV-mediated α -syn overexpression is not exclusive to enteric neurons. However, using this particular delivery method we have demonstrated that this vector (AAV9) transduces murine enteric neurons, with no transduction of muscle or glia^{1,2}. We have revised the description of our models in the Results and Methods and specified the delivery method. We also clarify that Figure S13A and S13B involving the rAAV model shows an enteric plexus in the

cecal patch containing punctate human α -syn and pS129 α -syn inclusions, respectively. Overall, both the inflammation related A30P α -syn and wild type paradigm and the rAAV-induced human α -syn model serve to improve our understanding of consequences of excess α -syn accumulation on the ALP in the appendix.

Results, pg. 9. We and others have previously reported that wild-type and A30P α -syn mutant mice develop α -syn aggregates in enteric neurons triggered by this chronic DSS colitis paradigm [72, 73], though processes beyond α -syn aggregation (i.e., synaptic transmission, immunological responses) are also altered in these models [71, 74].

Results, pg. 10. We then sought to determine whether α -syn aggregation in enteric neurons was a key contributor to epigenetic dysregulation of the ALP induced by intestinal inflammation. In this experiment, we injected into the cecal patch of wild-type C57BL/6J mice a recombinant adeno-associated virus (rAAV) vector that overexpressed either human α -syn or GFP as control [74] (Fig. 5a) using direct subserosal delivery to the ENS. This targeted approach results in transduction of neurons per se, with no off-target transduction of support cells such as muscle or glia [74, 75])

References

1. Benskey MJ, *et al.* Targeted gene delivery to the enteric nervous system using AAV: a comparison across serotypes and capsid mutants. *Mol Ther* **23**, 488-500 (2015).
 2. Benskey MJ, Manfredsson FP. Gene Therapy of the Peripheral Nervous System: The Enteric Nervous System. *Methods Mol Biol.* 2016;1382:263-274. doi:10.1007/978-1-4939-3271-9_19
-

Reviewer 1, Comment 3: Interesting point that may be addressed:

In the Discussion, the potential as biomarkers are presented “epigenetic changes in the ALP in the appendix may be a potential biomarker for disease risk and progression by serving as a proxy for the epigenetic status of ALP genes in the brain”. It will be interesting to determine if such changes also occur on other sites in the intestinal tracts that are amenable for biopsy as this will improve their significance as biomarkers.

Response 3: We appreciate this reviewer comment. The appendiceal orifice is routinely identified during a total colonoscopic examination and can be biopsied¹. As recommended by Reviewer 3 we have revised this sentence to:

Discussion, pg. 13. The appendiceal orifice is routinely identified during a total colonoscopic examination and can be biopsied [91], and as such, is more accessible than the brain. Our findings suggest that epigenetic changes in the ALP in the appendix may serve as a proxy for ALP status in the brain, though this would require further investigation across disease stages.

Reference

1. Khawaja FI. Diseases of the appendix recognized during colonoscopy. *Saudi J Gastroenterol.* 2002;8(2):43-52. PubMed PMID: 19861790.

Reviewer #2

The paper by Gordevicius et al., performs a tour de force analysis of epigenetic and expression changes in the appendix olfactory bulb and prefrontal cortex in PD patients. Using padlock bisulfite sequencing the authors measure at high resolution at 521 genes belonging to the autophagy lysosome pathway (ALP) as well as genes associated with PD by GWAS studies. They show changes in methylation in 326 of these genes. Changes are enriched in lysosomal genes in promoters and enhancers. Examination of olfactory bulb and a replicate study in the prefrontal cortex shows parallel changes in DNA methylation in the brain. Transcriptome analysis of PD and control appendix shows downregulation of ALP genes and enrichment in lysosomal genes. Proteomic analysis shows as well downregulation of ALP genes as well as downregulation of several genes that are epigenetically regulated. The authors show that many of the genes changed in PD change also in aging however there is no aging effect in PD. The authors further examine the involvement of ALP genes in PD using a mouse model expressing α -synuclein and combine it with a DSS colitis model. They show that DSS colitis triggers differential methylation of ALP genes and that this effect is enhanced in α -synuclein transgenics. There is an overlap between genes that are epigenetically programmed in the transgenic mouse overexpressing α -synuclein as well as mice infected in the cecal patch by an AAV overexpressing α -synuclein. However, while there is an overlap between ALP genes epigenetically programmed by overexpression of α -synuclein in mice and PD, the epigenetic program by DSS does not resemble the epigenetic changes in ALP genes in PD appendix. While DSS triggers mainly hypomethylation of ALP genes, in the PD appendix the response is predominantly hypermethylation. Overall, this paper provides strong evidence for involvement of epigenetic reprogramming of ALP genes in the gut and the brain in PD and provides a plausible mechanism.

This is an extremely important discovery that expands our understanding of the mechanisms involved in PD and provides a strong argument for involvement of epigenetic programming of ALP genes in the disease. The authors use a very robust method to examine the DNA methylation state of a large group of functionally linked genes at high resolution, the results are extremely convincing and provides evidence for reprogramming of lysosomal functions in PD. The overlap between the appendix and the brain and the replication in two prefrontal cortex samples coupled with evidence from proteomic and transcriptomic analyses as well as an animal model overexpressing α -synuclein provides compelling evidence for the involvement of epigenetic reprogramming of ALP genes in the molecular pathology of PD. There are however several issues that I believe could be clarified in a revised version of this paper.

Response: We appreciate the reviewer's interest in our study and helpful comments to improve our manuscript.

Reviewer 2, Comment 1: The authors claim that the main response in PD is hypermethylation. Indeed, although the majority of DNA methylation changes in PD ALP genes is hypermethylation, a large fraction of the genes are hypomethylated. What is the role of hypomethylation in regulating ALP genes? This shouldn't be ignored.

Response 1: In PD appendix tissue, we identified DNA methylation abnormalities at 928 cytosine sites affecting 326 ALP genes, of which 192 were predominantly hypermethylated, while 134 were hypomethylated. We investigated the genomic locations enriched with differentially methylated sites. Gene promoters, CpG islands and active enhancers were found to be highly enriched with hypermethylation, which is strongly associated with gene silencing¹⁻⁵ (Figure 1c and S2). Hypermethylation most often occurred at genes in the lysosome pathway, with a down-regulation of genes in that pathway (Figure 1d and Figure S3). A fewer number of genes show hypomethylation, and we find that these are enriched in the autophagy pathway (Figure S3). We have added a text in the Results to describe this finding. We also list the genes that are more hypomethylated (as well as the more hypermethylated genes) in Supplementary data 17 and as part of the Reviewer 2, Response 10.

Results, pg. 3. DNA methylation abnormalities were detected at 928 cytosine sites affecting 326 ALP genes in the PD appendix relative to the control appendix (2.8 differentially methylated sites per affected ALP gene with 7% average methylation change; $q < 0.05$, robust linear regression; 192 and 134 genes had more hypermethylated and hypomethylated cytosines, respectively; **Fig. 1a**; Supplementary Data 1).

Results, pg. 4. Conversely, we found enrichment of hypomethylated cytosines among genes involved in autophagy (OR = 1.25; $p=0.04$; Supplementary Figure S3).

References

1. Ball MP, *et al.* Targeted and genome-scale strategies reveal gene-body methylation signatures in human cells. *Nat Biotechnol* **27**, 361-368 (2009).
2. Laurent L, *et al.* Dynamic changes in the human methylome during differentiation. *Genome Res* **20**, 320-331 (2010).
3. Varley KE, *et al.* Dynamic DNA methylation across diverse human cell lines and tissues. *Genome Res* **23**, 555-567 (2013).
4. Lister R, *et al.* Human DNA methylomes at base resolution show widespread epigenomic differences. *Nature* **462**, 315-322 (2009).
5. Greenberg MVC, Bourc'his D. The diverse roles of DNA methylation in mammalian development and disease. *Nat Rev Mol Cell Biol* **20**, 590-607 (2019).

Reviewer 2, Comment 2: The authors show that there is an overall silencing of ALP gene expression in PD their transcriptomic analysis, the correlation with methylation is significant but quite weak only 0.2. This suggests that for many ALP genes expression is reprogrammed by mechanisms that do not involve DNA methylation while other genes are differentially methylated with no effect on expression. What is the overlap between DNA methylation and expression?

What do we know about those genes that are downregulated by mechanisms that don't involve DNA methylation while other genes in the same functional family seem to be regulated by DNA methylation? What is the role of altered DNA methylation that is not accompanied by changes in gene expression?

Response 2: In our analysis, we find that 41 ALP genes exhibit DNA methylation alterations and nominal changes in expression. We have updated Additional File 3 to include the dominant direction of methylation of each ALP gene next to its differential expression fold change and p value.

In addition, the reviewer asks about the meaning of circumstances when DNA methylation status is not correlated to the transcript levels of its corresponding gene. The repressive role of DNA methylation in transcription has long been recognized with a correlation between DNA methylation and gene silencing, particular at gene promoters^{1,2}. However, gene expression begins with transcription and processing of messenger RNA (mRNA), followed by export to the cytoplasm for translation and ultimately decay. Included in this pathway are many regulatory steps that alter mRNA transcript levels, including quality control-based surveillance, splicing, RNA modification, translational control and decay processes³⁻⁵. In RNA-seq, we measure steady-state mRNA transcript levels (mRNA levels with all the synthesis and degradation regulatory steps). In our study, we observed that PD patients had a hypermethylation of lysosomal genes coupled with a downregulation of lysosome gene transcripts. However, the concordance between DNA methylation status and steady-state mRNA levels is likely dampened by the aforementioned processes that influences gene transcript levels.

References

1. Ball MP, *et al.* Targeted and genome-scale strategies reveal gene-body methylation signatures in human cells. *Nat Biotechnol* **27**, 361-368 (2009).
2. Greenberg MVC, Bourc'his D. The diverse roles of DNA methylation in mammalian development and disease. *Nat Rev Mol Cell Biol* **20**, 590-607 (2019).
3. Glisovic T, Bachorik JL, Yong J, Dreyfuss G. RNA-binding proteins and post-transcriptional gene regulation. *FEBS Lett* **582**, 1977-1986 (2008).
4. Popp MW, Maquat LE. Organizing principles of mammalian nonsense-mediated mRNA decay. *Annu Rev Genet* **47**, 139-165 (2013).
5. Herzel L, Ottoz DSM, Alpert T, Neugebauer KM. Splicing and transcription touch base: co-transcriptional spliceosome assembly and function. *Nat Rev Mol Cell Biol* **18**, 637-650 (2017).

Reviewer 2, Comment 3: The analysis reveals overall overlap in DNA methylation in genes between PD appendix and brain regions. Are the same sites methylated in both tissues? What is the overlap at the CG level?

Response 3: We provide the differentially methylated cytosines in the PD appendix and brain in Supplementary Files 1, 3, 4, and 6. There is not a significant overlap between the datasets at the cytosine site level, but this is not unexpected because human brain neurons have an

abundance of CpH methylation (CpH accounts for 53% of the methylated fraction and non-brain tissue lacks CpH methylation)^{1,2}. Furthermore, gene regulatory element locations are known to differ between cell/tissue types^{3,4}. Thus, DNA methylation changes do not have to occur at the same cytosine site to similarly impact gene activity across tissues. Instead, in our study we look across appendix and brain datasets for concordance in differentially methylated ALP genes and genomic elements (Figure 2c), finding that many genomic element types exhibit similar changes in DNA methylation in the PD appendix and PD brain.

References

1. Bernstein BE, *et al.* The NIH Roadmap Epigenomics Mapping Consortium. *Nat Biotechnol* **28**, 1045-1048 (2010).
 2. Luo C, Hajkova P, Ecker JR. Dynamic DNA methylation: In the right place at the right time. *Science* **361**, 1336-1340 (2018).
 3. Encode Project Consortium. An integrated encyclopedia of DNA elements in the human genome. *Nature* **489**, 57-74 (2012).
 4. Lister R, *et al.* Human DNA methylomes at base resolution show widespread epigenomic differences. *Nature* **462**, 315-322 (2009).
-

Reviewer 2, Comment 4: Correlation of protein abundance in brain and appendix is 0.26. 34 proteins are epigenetically altered in appendix, what about the state of expression of these proteins in brain? What fraction of the ALP proteins that are altered are also epigenetically regulated?

Response 4: In Table S1 we provide the ALP protein changes in the PD appendix and brain for the epigenetically altered ALP genes. Further, we find that the majority of ALP proteins altered in the PD appendix and prefrontal cortex are also epigenetically altered in their respective tissues (76% and 70% of ALP proteins altered in the PD appendix and brain, respectively, are epigenetically altered). The following text has been added to the Results:

Results, pg. 7. Overall, we found that the majority of ALP proteins altered in the PD appendix and prefrontal cortex are also epigenetically altered in their respective tissues (76%, 70%, 85.7% of ALP proteins altered in the PD appendix, prefrontal cortex, or both tissues, respectively, were epigenetically altered).

Reviewer 2, Comment 5: Out of the ALP proteins that are concordant between brain and appendix, how many are epigenetically regulated?

Response 5: Of the ALP proteins with altered levels in both the PD appendix and brain, 85.7% are epigenetically altered in PD. We have revised the Results as described in Reviewer 2, Comment 4. Also Table S1 details the epigenetically altered ALP genes exhibiting protein changes in the PD appendix and prefrontal cortex.

Reviewer 2, Comment 6: Proteins that are regulated by DNA methylation as well as those that are not should be discussed to appreciate the impact of epigenetic regulation on downregulation of ALP proteins.

Response 6: As mentioned in Reviewer 2, Comment 4 the majority of the ALP proteins altered in PD tissues are differentially methylated ($\geq 70\%$ of altered ALP proteins). We have amended additional files 9 and 10 to include all reliably detected proteins and indication for each gene whether it is differentially methylated and/or differentially expressed in appendix and whether it is differentially methylated in either of the prefrontal cortex datasets. ALP genes were marked as well, making it easy to verify the above findings.

Reviewer 2, Comment 7: The authors show concordance between genes that are epigenetically altered in aging and those altered in PD. It would be expected therefore that aging is accelerated in PD. But it seems that there is no effect of aging on ALP genes in PD. How is this possible? Do the ALP genes reach higher level of methylation earlier than in healthy patients? This should then be illustrated by a figure showing this in several gene methylation map examples.

Response 7: In order to show the aging effects in PD, we added to the manuscript an analysis of cytosine epigenetic aging rates among PD samples in appendix and prefrontal cortex neurons. In the appendix of PD samples there were 561 aging cytosines affecting 251 ALP genes. In prefrontal cortex neurons of PD samples there were 77 aging cytosines affecting 70 ALP genes. We computed mean aging rate for each ALP gene by weighted averaging of aging rates at individual cytosines (new Figure 4e). We found a strong overlap of genes affected by aging cytosines between healthy and PD samples in appendix ($OR = 3.43$, $p < 2.2 \times 10^{-16}$) but not in prefrontal cortex neurons ($OR = 1.58$, $p = 0.09$). Further, we found that in appendix the 118 ALP genes affected by age among both healthy and PD samples had concordant aging direction ($OR = 2.21$, $p = 0.05$; Fisher's exact test) and their absolute aging rate was higher in PD (1.58×10^{-15} ; paired t-test) indicating accelerated aging. There was no such agreement in prefrontal cortex neuron samples. This additional analysis indicates that epigenetic aging is present in PD samples, yet it is markedly different when compared to normal aging of ALP function.

The following changes have been made to the manuscript:

Results, pg. 9. Next, we compared epigenetic aging rates of ALP genes in PD patients to that in healthy controls. In the appendix of PD samples there were 561 aging cytosines affecting 251 ALP genes. In prefrontal cortex neurons of PD samples there were 77 aging cytosines affecting 70 ALP genes. For each ALP gene, we computed mean aging rate weighted by the aging p value of each cytosine pertaining to that gene and set the rate to zero if there were no

significantly aging cytosines (**Fig. 4e**). In the appendix, we found 118 ALP genes aging among both healthy control and PD samples, more than expected by chance (OR = 3.43, $p < 2.2 \times 10^{-16}$; Fisher's exact test), and among them there was an agreement of aging direction (OR = 2.21, $p = 0.05$; Fisher's exact test). The absolute aging rates of ALP genes were higher in PD than in control samples ($p = 1.58 \times 10^{-15}$; paired t-test), suggesting accelerated aging. In prefrontal cortex neurons, 35 ALP genes were aging in both healthy control and PD samples (OR = 1.58, $p = 0.09$; Fisher's exact test); there was no significant agreement of aging direction nor aging rate (OR = 0.31, $p = 0.55$ and $p = 0.35$; Fisher's exact test and paired t-test, respectively). Together, this suggests that in patients, PD disease processes disrupt normal aging changes in ALP function.

Methods, pg. 26. We also compared epigenetic aging patterns of the ALP genes in the healthy and PD appendix and prefrontal cortex. Summarized aging trend for each gene was computed as weighted mean aging rate of all cytosines pertaining to the gene where log transformed aging p value was used as weight. The aging trend was set to zero for the ALP genes that did not have a single significantly aging cytosine (FDR $q < 0.05$).

Figure 4 sections d and e, figure caption:

(d) Overlap of the ALP genes affected by mean methylation differences between control and PD samples or age-related methylation changes in healthy control or PD samples in the appendix (upper panel) and prefrontal cortex neurons (lower panel). A gene is affected if it has a significantly differentially modified cytosine (FDR $q < 0.05$). **(e)** Epigenetic aging rates of ALP genes in the appendix (upper panel) and prefrontal cortex neurons (lower panel) of healthy controls (blue points) and PD cases (purple points). For each ALP gene, the aging rate was computed as mean aging rate of cytosines pertaining to that gene weighted by log transformed p value of aging model fit. The aging rate was set to zero for ALP genes that did not have any aging cytosines with FDR $q < 0.05$. Genes were sorted by the sum of their aging rates in control and PD samples. PD related genes implicated in GWAS studies are marked by orange segments and labeled. Appendix $n = 51$ controls, 24 PD; prefrontal cortex $n = 42$ controls, 52 PD.

Reviewer 2, Comment 8: The authors do not provide any gene maps with CG positions and their rate of change in methylation in PD and normal aging?

Response 8: Supplementary Data 11 and 12 list age-associated DNA methylation changes of healthy individuals as well as PD patients in appendix and prefrontal cortex neurons, respectively. For each interrogated cytosine, the table includes its chromosomal location, the symbol of the gene to which it is assigned, and its aging rate and p-value in control samples and aging rate and p-value in PD samples. In addition, we added a Supplementary Figure S10 with visual representation of three genes mentioned in the manuscript: SNCA, LMX1B and MTOR. Finally, we added Supplementary Data 17 that summarizes significant findings at the level of each ALP gene across all analyses performed in this paper.

Supplementary Figure S10 has been added to the paper:

Figure S10. PD related and aging changes in individual cytosines interrogating three PD implicated genes in PD appendix and prefrontal cortex neurons. (a) SNCA. (b) LMX1B. (c) MTOR. Purple and green bars show methylation changes with aging in control and PD, respectively, and blue bars show methylation changes in PD compared to control. Bar heights correspond to signed log-transformed FDR q value. Red dashed lines indicate FDR $q < 0.05$ threshold. Transparent filled rectangles indicate the area corresponding to the gene transcript.

Reviewer 2, Comment 9: How large are the changes in methylation throughout the paper? The authors provide numbers of genes that change significantly but not the scope and range of changes.

Response 9: We agree that this revision will benefit the clarity of results and have added the percentage of DNA methylation change at each cytosine to the Supplementary Data 1, 3, 4, 5, 6, 11, 12 and 13. Also, in the results we specify the average DNA methylation change at affected cytosines as well as the average number of differentially modified cytosines per affected ALP gene. The results have been revised to:

Results, pg 3. DNA methylation abnormalities were detected at 928 cytosine sites affecting 326 ALP genes in the PD appendix relative to the control appendix (2.8 differentially methylated sites per affected ALP gene with 7% average methylation change; $q < 0.05$, robust linear regression; Fig. 1A; Supplementary Data 1).

Results, pg. 5. In the olfactory bulb, we identified 1,142 differentially methylated cytosines affecting 353 genes in PD relative to controls (3.2 differentially methylated sites per affected ALP gene with 18% average methylation change; $q < 0.05$, robust linear regression; Fig 2A; Supplementary Data 3).

Results, pg. 5. In prefrontal cortex neurons, we observed 70 differentially methylated sites affecting 58 genes in PD (1.2 differentially methylated sites per affected ALP gene with average methylation change 8% in CpG and 6% in CpH sites; $q < 0.05$, robust linear regression; Fig 2A; Supplementary Data 4)

Results, pg. 5. There were 1,131 differentially methylated cytosines in neurons of PD patients relative to those of controls (3.3 differentially methylated sites per affected ALP gene with 13% average methylation change; $q < 0.05$, robust linear regression; Fig 2A; Supplementary Data 6).

Results, pg. 8. In the healthy appendix, there were 285 cytosine sites at 170 ALP genes showing aging changes in DNA methylation (1.7 differentially methylated sites per affected ALP gene with 0.44% average methylation change per year; $q < 0.05$, robust linear regression after adjusting for sex, postmortem interval, batch, and other sources of variation by surrogate variables factor analysis; Supplementary Data 11).

Results, pg. 8. In prefrontal cortex neurons of controls, 304 differentially methylated sites affecting 200 ALP genes occurred with aging (1.5 differentially methylated sites per affected ALP gene with 0.42% average methylation change per year; $q < 0.05$, robust linear regression adjusting for sex, postmortem interval, and neuron subtype proportion; Supplementary Data 12).

Results, pg. 8. In olfactory bulb of controls there were 853 epigenetically aging sites affecting 325 ALP genes (2.6 differentially methylated sites per affected ALP gene with 0.75% average methylation change per year; $q < 0.05$, robust linear regression adjusting for sex, postmortem interval, and neuron subtype proportion; Supplementary Data 13).

Results, pg. 10. In wild-type mice, previous DSS colitis resulted in 1,104 differentially methylated cytosines affecting 397 ALP genes, relative to mice that did not experience colitis (2.8 differentially methylated sites per affected ALP gene with 8.4% average methylation change; $q < 0.05$, robust linear regression; Fig 5B; Supplementary Data 15). In A30P α -syn mice, there were 1,378 differentially methylated cytosines affecting 408 ALP genes (3.4 differentially methylated sites per affected ALP gene with 9% average methylation change; $q < 0.05$, robust linear regression; Fig. 5C).

Results, pg. 10: We found 896 differentially methylated cytosines affecting 365 ALP genes in response to α -syn aggregation (2.5 differentially methylated sites per affected ALP gene with 10.85% average methylation change; $q < 0.05$, robust linear regression; Additional File 1: Figure S14; Supplementary Data 16).

Methods, pg. 21. Change in methylation percentage was computed by converting fitted m values of each cytosine and each sample into beta values using formula $B = 2^m / (2^m + 1)$. Linear models were then fitted on the B values and model coefficients were extracted.

Reviewer 2, Comment 10: The authors show that a-synuclein over-expression in mice exacerbates the changes in ALP DNA methylation that occur during chronic inflammation. These changes are predominantly hypomethylation. They also show that the changes induced by DDS inflammation do not correspond to changes in methylation seen in PD patients and that they take an opposite direction, hypomethylation in DDS inflammatory response versus hypermethylation in PD. They show on the other hand that a-synuclein over-expression triggers changes in methylation that are concurrent with the changes seen in PD. How is this possible? A-synuclein over-expression =PD, a-synuclein =DDS, then one would expect that DDS=PD. Are we talking about one set of genes that are targeted by a-synuclein and DDS that is hypomethylated and then a totally different set of genes that is hypermethylated by a-synuclein and is also hypermethylated in PD? If this is the case, it might suggest a disconnect between the effects of DDS-inflammation and the link between a-synuclein and PD which is different from the model proposed by the authors. This needs to be discussed and the identity of the genes

that are either hypomethylated or hypermethylated in DDS, α -synuclein over-expression and PD should be revealed and perhaps presented and illustrated.

Response 10: We thank the reviewer for requesting this revision which greatly improved our study's investigation of the ALP changes in the mouse model of synucleinopathy and DSS-mediated gut inflammation. We examined the overlap of the genes exhibiting predominant hypermethylation or hypomethylation in the PD appendix, in mice with synucleinopathy, and in mice in response to DSS colitis. The results of this analysis are presented in Figure 5c and the genes within each overlapped domain are listed in Supplementary Data 17. Interestingly we examined whether there was significant pathway enrichment in each of the overlapping domains. We found that the PD appendix and mice with synucleinopathy shared a hypermethylation of lysosomal genes (OR = 1.9, $p = 0.03$; Fisher's exact test). Meanwhile, there was a hypomethylation of genes involved in autophagy that overlapped between PD appendix, mice with synucleinopathy and following DSS colitis (OR = 2.16, $p = 0.01$; Fisher's exact test). Therefore, even though there is a predominant hypermethylation of lysosomal genes in the PD appendix and induced by synucleinopathy, there are autophagy genes exhibiting hypomethylation that are in common between the PD appendix, mice with synucleinopathy, and mice exposed to gut inflammation. We have revised Results and Methods and include the analysis in new Figure 5 and Supplementary Data 17:

Results, pg. 11. Further, pathway analysis revealed that α -syn overexpression in the A30P mouse model recapitulated the abnormalities in lysosome function in the PD appendix and affected autophagy (OR = 2.63, $p = 0.001$ and OR = 1.85, $p = 0.04$, respectively, Fisher's exact test; Supplementary Figure S15). In addition, we examined the overlap of the genes exhibiting predominant hypermethylation or hypomethylation in the PD appendix, in mice with synucleinopathy, and in mice in response to DSS colitis (**Fig. 5d**; Supplementary Data 17). We performed a pathway analysis for overlapping genes, and again found that the PD appendix and mice with synucleinopathy shared a hypermethylation of lysosomal genes (OR = 1.9, $p = 0.03$; Fisher's exact test). Meanwhile, there was a hypomethylation of genes involved in autophagy that overlapped between PD appendix, mice with synucleinopathy and following DSS colitis (OR = 2.16, $p = 0.01$; Fisher's exact test). Therefore, even though there is a predominant hypermethylation of lysosomal genes in the PD appendix and induced by synucleinopathy, there are autophagy genes exhibiting hypomethylation that are in common between the PD appendix, mice with synucleinopathy, and mice exposed to gut inflammation.

Methods, pg. 24. For any ALP gene the direction of DNA methylation change was determined by the direction of the majority of significant cytosines, with ties resolved by the direction of weighted mean fold change of all cytosines pertaining to the gene. Log transformed p value was used as weight that way emphasizing the most significant cytosines.

Methods, pg. 28. ALP genes differentially methylated in appendix were overlapped with those differentially methylated in synucleinopathy or DSS colitis. The genes differentially methylated in either rAAV α -syn mice or between A30P and wild type mice treated with water were deemed as

affected by synucleinopathy. Similarly, genes affected by DSS colitis were those that showed difference between DSS and water treatments in either A30P or wild type mice. As before, dominant direction of methylation for a gene was determined by majority significantly affected cytosines with ties resolved by weighted mean fold change.

Figure 5:

Fig. 5. ALP changes in DNA methylation in response to experimental gut inflammation and α -syn aggregation. DNA methylation was fine-mapped in the cecal patch of mice exposed to chronic DSS colitis, examining both wild-type mice and a mouse model of synucleinopathy, A30P α -syn mice (n = 40 mice: 9 A30P/DSS colitis, 10 A30P/Water, 10 WT/DSS colitis, 11 WT/Water). DNA methylation changes in response to α -syn aggregation were examined using a rAAV-mediated α -syn overexpression mouse model (n = 5 control vector and 5 α -syn overexpression vector in wild-type mice). **(a)** Schema of experimental design for the gut inflammation (left panel) and vector-mediated α -syn aggregation (right panel) studies. **(b)** Comparison of ALP changes induced by rAAV vector-mediated α -syn aggregation to those mediated by gut inflammation in wild-type and A30P α -syn mice. ALP gene enrichment in differentially methylated cytosines was determined. Plot shows concordance of epigenetic changes at ALP genes between rAAV-mediated α -syn aggregation mice and the wild-type and A30P α -syn mice in the DSS colitis study. **(c)** Comparison of ALP changes occurring in the PD appendix to those mediated by α -syn aggregation or gut inflammation. Concordance of epigenetic changes at ALP genes between PD appendix study and mouse studies. * $p < 0.05$, ** $p < 0.01$, *** $p < 0.001$ Kendall's rank correlation coefficient. **(d)** Overlap of hyper- and hypomethylated genes differentially methylated in PD appendix, rAAV a-syn and A30P water treated

mice with synucleinopathy and A30P and wild type mice with DSS colitis. Genes in each overlapping domain were tested for enrichment by ALP pathway genes using Fisher's exact test.

Supplementary Figure S12:

Figure S12. Differential methylation of the ALP in the cecal patch of A30P α -syn mice. DNA methylation was profiled at 571 ALP genes in the cecal patch of mice overexpressing human α -syn with the heterozygote A30P mutation (A30P α -syn mice), a PD-relevant model of synucleinopathy. **(a)** In wild-type mice that experienced DSS-mediated gut inflammation there were 397 genes in the ALP exhibiting differential methylation relative to wild-type mice that did not experience DSS colitis (1,104 sites at $q < 0.05$, robust linear regression). **(b)** In A30P α -syn mice that experienced DSS-mediated gut inflammation there were 408 genes in the ALP exhibiting differential methylation relative to A30P mice that did not experience DSS colitis (1,378 sites at $q < 0.05$, robust linear regression). **(c)** In A30P α -syn mice treated with water there were 315 genes in the ALP exhibiting differential methylation relative to wild-type mice treated with water (591 sites at $q < 0.05$, robust linear regression). ALP genes implicated in both our study and PD risk by GWAS [23] (mapped to corresponding mouse genes) are labeled. SLP refers signed log p-value, with sign corresponding to the direction of DNA methylation change (hypermethylation, green or hypomethylation, blue). Barplots indicate the fraction of significantly hypermethylated (green) and hypomethylated (blue) sites. The enrichment of hypermethylated loci was determined by Fisher's exact test, OR < 1 indicates dominant hypomethylation.

Reviewer 2, Comment 11: In summary, the epigenetic alterations in ALP genes in PD are robust and convincing but the effects of aging and DDS mediated inflammation are confusing and contradictory. These need to be analyzed in detail and explained more clearly. Alternatively, the paper could focus on PD and a synuclein overexpression in the mouse and the potential mechanistic implications of deregulation of ALP genes in the molecular pathology of PD.

Response 11: We thank the reviewer for their recommendations and have revised the aging and DSS sections as detailed in Reviewer 2, Response 7, 8, and 10. We have retained the aging section, as well as the synucleinopathy and colonic inflammation models in the manuscript because of the relevance of these processes to PD risk. Our analysis helps us understand whether aging, synucleinopathy, and colonic inflammation could manifest disease risk via changes in the ALP. We observe an epigenetic inactivation of the ALP in healthy aging, involving a hypermethylation of gene promoters, CpG islands, and poised enhancers. Synucleinopathy induced changes in the ALP that are consistent with those of PD, which included hypermethylation particularly at lysosomal genes. Colonic inflammation (mediated by DSS) largely induces a hypomethylation of the ALP. This is consistent with evidence that activation of autophagy is important for resolving inflammation^{1,2}. Furthermore, DSS colitis effects on the ALP were amplified by synucleinopathy. This exaggerated response to an inflammatory event may be an increased activation of autophagy in an attempt to overcome deficient lysosomal function mediated by α -syn accumulation. Indeed, we find that autophagy genes exhibiting hypomethylation are in common between the PD appendix and mice models of synucleinopathy and gut inflammation, while lysosomal genes are hypermethylated in the PD appendix and by synucleinopathy. Our discussion has been revised as follows:

Discussion, pg. 13: There is a robust interplay between inflammation and the ALP [95]. Autophagy is involved in the induction and suppression of inflammation [95, 96]. It regulates the development, homeostasis, and survival of inflammatory cells and affects the transcription, processing, and secretion of cytokines [95]. Loss of autophagy has proinflammatory consequences, and in the gut, ALP suppression exacerbates the inflammatory effects of DSS colitis [96, 97]. In our study, gut inflammation mediated by DSS largely induced a hypomethylation of the ALP. This is consistent with evidence that activation of autophagy is important for resolving inflammation [95, 96]. Furthermore, DSS colitis effects on the ALP were amplified by synucleinopathy. Synucleinopathy in mice induced a hypermethylation of lysosomal genes which was consistent with that observed in PD. Thus, the exaggerated response to an inflammatory event may be an increased activation of autophagy in an attempt to overcome deficient lysosomal function mediated by α -syn accumulation. Indeed, we find that autophagy genes exhibiting hypomethylation are in common between the PD appendix and mice models of synucleinopathy and gut inflammation, while lysosomal genes are hypermethylated in the PD appendix and by synucleinopathy. Excessive or prolonged autophagy stimulation is detrimental as it can lead to cell death, including in the GI tract (known as autophagic cell death and autosis) [98-101]. Thus, our study suggests that epigenetically-mediated ALP abnormalities in

the PD appendix may, in part, be due to an accumulation of α -syn. Given the evidence supporting that inflammation (including in the GI tract) plays a key role in PD pathogenesis [69, 70], it is possible that normative ALP activation needed to resolve inflammation is incapacitated in PD.

References

1. Deretic V, Levine B. Autophagy balances inflammation in innate immunity. *Autophagy* **14**, 243-251 (2018).
 2. Saitoh T, *et al.* Loss of the autophagy protein Atg16L1 enhances endotoxin-induced IL-1 β production. *Nature* **456**, 264-268 (2008).
-

Reviewer #3 (Remarks to the Author):

The authors presented an interesting and well conducted study in which they produced a comprehensive set epigenetic, transcriptomic and proteomic data from human and mouse tissues. They suggest that methylation and transcriptome differences in samples taken from the human appendix and brain (olfactory bulb and prefrontal cortex neurons) distinguish Parkinson's patients from controls in terms of methylation for a preselected number of genes contributing to the autophagy-lysosomal pathway (ALP) as well as some select known risk genes for PD (GWAS derived). Based on their data, they conclude that ALP dysregulation through methylation and differential expression of these genes may play a key role in the pathogenesis of PD. This conclusion is generally in concordance with the current existing knowledge about the importance and the role of autophagy in neurodegenerative disorders especially Parkinson's disease.

This study was hypothesis driven with the ALP as the target. The ALP is considered an important pathway in PD (due to GWAS results, and also because GBA - a Mendelian gene - contributes to familial PD) that when disrupted may cause the accumulation of aggregation-prone α -syn, a hallmark protein that contributes to Lewy-bodies in PD. While this is not new as a hypothesis in general, the hypothesis and the tissues (appendix together with the olfactory bulb and brain (PFC)) these authors chose to evaluate together are very novel and interesting as the gut has only recently been implicated as a possible origin/source of α -syn that would be transported into the brain retrogradely. Also, α -syn aggregates were found in healthy humans and in PD in the appendix and if these protein aggregates are responsible for seeding brain aggregates, it is of great interest to determine what makes the difference between those who remain free of PD and those who eventually develop the disorder i.e. to find avenues for early intervention and prevention.

Response: We thank the reviewer for the time and effort taken in your detailed review and helpful suggestions to improve our study.

Reviewer 3, Comment 1: To determine the interaction of the ALP molecular system and α -syn aggregates in the gut and brain, the authors used human samples and assessed local tissue pathologies i.e. comparing samples from people with and without PD presumably at one point in time at autopsy. In the text it is hard to tell whether and the authors need to spell out more clearly at what point in time these samples were taken. It seems they all were derived from autopsies? However, is this also true for appendix at younger ages or whether these appendices from surgeries in living individuals. Also, it is not clear whether some of the tissues from multiple organs are recovered from the same person or whether they all were derived from different donors postmortem? While the age at sampling and sex are reported, for PD patients not only the Braak stage but also the duration of disease in years since diagnosis or onset would be interesting to know if such information were available. Similarly, for control tissues it would be interesting to know whether the donors had died of an accident or some disease and what the predominant cause of deaths were - especially for the appendix samples; e.g. are these individuals all free of colitis etc..

Response 1: All samples in the comparison between the PD and control appendix was performed with postmortem tissue. In our aging analysis, we included surgically-isolated, histologically normal appendix tissues from control (non-PD) individuals. Surgical samples were obtained from individuals undergoing a right hemicolectomy for intestinal cancer not involving the appendix (appendix incidentally removed and histologically confirmed to be normal). In our aging analysis, we adjusted for differences in sample source (postmortem or surgical). Sample source information has been clarified in the Methods 'Human Tissue Samples Section'. In addition, no individual had a known diagnosis of inflammatory bowel disease. All PD patients, including those in the appendix study, had pathologically confirmed Lewy pathology in the brain. Controls had no brain Lewy pathology. Finally, there is not sufficient overlap in the samples between the datasets to perform a meaningful comparison in the same subjects across tissues. For example, only 4 PD patients are the same across the appendix, prefrontal cortex neurons, and olfactory bulb datasets.

The following changes have been made to the manuscript:

Methods, pg. 18. Postmortem appendix tissue from PD patients and controls was obtained from the Oregon Brain Bank. In our aging analysis, we included surgically-isolated, histologically normal appendix tissue from control (non-PD) individuals obtained from the Spectrum Health Universal Biorepository and Cooperative Human Tissue Network (CHTN). Appendix surgical samples were from individuals undergoing a right hemicolectomy for intestinal cancer not involving the appendix (appendix incidentally removed and histologically confirmed to be normal). Prefrontal cortex tissue was obtained from the NIH NeuroBioBank, Parkinson's UK Brain Bank, Michigan Brain Bank (primary cohort), or the Oregon Brain Bank (replication cohort). Olfactory bulb tissue was obtained from the Oregon Brain Bank. For the study samples, we had information on demographics (age, sex), tissue quality (postmortem/surgical interval), and pathological staging (Additional File 19). Appendix, prefrontal cortex, and olfactory bulb postmortem tissue from PD patients have evident brain Lewy pathology (PD Braak stages

III-VI), whereas control individuals have no Lewy pathology in the brain. Sample information is detailed in Supplementary Data 18.

Reviewer 3, Comment 2: The ultimate question - and the authors clearly state this - is whether there are processes such as transport of α -syn from the gut to the brain or vice versa to which and how the ALP and inflammation may contribute. Nevertheless, the authors should state more clearly that the direction cannot be determined in this study except maybe indirectly. Similarly, these tissues were necessarily taken from PD patients long after the onset of disease – which opens up the possibility that there is reverse causation i.e. that the localized changes in the gut methylome represent consequences and not causes of a disease known to affect the parasympathetic nervous system of the gut with constipation a known non-motor symptom; i.e. the question of what mechanisms underly the development of α -syn aggregates in the gut or what triggers its spread to the brain cannot necessarily be answered here except indirectly. The authors acknowledge this by stating that “Our human studies do not discern whether epigenetic changes at ALP genes are causal to PD or a consequence of PD pathophysiology” but there are many passages in which a presumed direction is implied without the necessary qualification. Indeed, figure 6 is most helpful to illustrate these points i.e. that we are seeing more or less a snapshot in time for multiple feedback loops between inflammation, ALP (dys)function, and α -syn aggregates in the gut and brain. I would like to ask the authors to consider pulling this figure up in the manuscript to the front to help the reader orient him/herself and also to carefully review the manuscript for wording about directionality such that this ambiguity about direction that implies causes is always clear to the reader.

Response 2: The reviewer brings up an important point that since the PD patients investigated in this study all have the disease, the direction of the ALP changes and its effects on α -syn propagation (gut-to-brain or brain-to-gut) is not discerned by this study. Since we found an epigenetically-mediated silencing of the ALP in the PD appendix, we propose that this tissue site exhibits molecular changes that promotes α -syn aggregation and propagation, but this study does not determine whether ALP dysregulation in the PD appendix preceded that of the brain. We have added text in the Discussion to clarify this point. In addition, we have limited the mention of gut to brain α -syn propagation in the paper to where we hypothesize this is a potential effect of ALP dysfunction, and since this is a means by which the gut has been proposed to contribute to PD there is some discussion of the literature on this topic. Figure 6 summarizes our proposed model based on our findings and the literature. We agree with the reviewer that this figure helps the readers, though the style of Nature Communications is to present such proposed models after the Results.

Discussion, pg. 12. Epigenetically induced dysfunction of the ALP in the PD appendix may contribute to the accumulation of aggregated α -syn in the gut and brain. In the PD appendix, widespread epigenetic inactivation of the ALP signifies that this tissue site exhibits molecular changes capable of promoting synucleinopathy; however, this study does not delineate whether

the ALP dysregulation in the PD appendix preceded that of the PD brain. Nonetheless, imaging studies of prodromal PD patients support that pathology can occur in the gut prior to the brain [11]. Aggregated α -syn has also been detected in enteric neurons of individuals in the prodromal stage of PD [7, 11] — in some cases as early as 20 years prior to the onset of motor symptoms [4]. Studies using animal models have demonstrated that α -syn pathology is capable of propagating from the gut to the brain via the vagus nerve [8, 9], though there is also evidence for bidirectional transfer of aggregated α -syn between the gut and brain [88, 89]. There is an abundance of aggregated α -syn in both the healthy and PD appendix, although α -syn levels are up to three times greater in the PD appendix [12]. In combination with epigenetic perturbation of lysosomal function, hypomethylation of the α -syn gene in the PD appendix may propel α -syn pathology. Indeed, studies in the brain have found that endogenous α -syn levels influence the spread of synucleinopathy [90]. Thus, epigenetic changes in the PD appendix are consistent with an increased production and impaired clearance of α -syn pathology.

In patients in which PD has fully emerged, epigenetic changes at ALP genes in the PD appendix are similar to those occurring in the PD brain. Our proteomic analysis also identified consistently increased levels of α -syn protein (SNCA) and decreased levels of NAMPT, HSPA8, and VPS35 in both the PD appendix and brain. Similarities across the PD gut and brain signify that the epigenetically dysregulated genes that enable the development, progression, and transport of α -syn pathology in the appendix could also facilitate the propagation of pathology within the brain. The appendiceal orifice is routinely identified during a total colonoscopic examination and can be biopsied [91], and as such, is more accessible than the brain. Our findings suggest that epigenetic changes in the ALP in the appendix may serve as a proxy for ALP status in the brain, though this would require further investigation across disease stages.

Typical epigenetic aging of the ALP appears to be disrupted in PD. In the healthy aging appendix and prefrontal cortex, we found that hypermethylated cytosines were overrepresented among genes in the selective autophagy pathway. In the brain, hypermethylated cytosines were also overrepresented at macroautophagy genes. Selective autophagy, which involves the targeting of specific cargoes by ubiquitin tagging, includes the clearance of intracellular pathogens, also referred to as xenophagy [92]. The accumulation of suppressive epigenetic marks affecting selective autophagy with aging suggests that the brain may be more susceptible to infection in advanced age. Chaperone-mediated autophagy (CMA) and macroautophagy are thought to be key pathways through which physiological α -syn and aggregated α -syn, respectively, are cleared from the cell [14, 83]. The hypermethylation of macroautophagy genes in the healthy aging brain may consequently render older individuals more vulnerable to the accumulation of α -syn aggregates. Though advanced age places individuals at greater risk for PD, the ALP in PD patients fails to exhibit normative epigenetic changes with aging. It may be the case that, in PD, various ALP pathways (e.g., macroautophagy, selective autophagy) do not undergo the same extent of hypermethylation as in healthy aging, in a futile attempt to compensate for the decrease in autophagic flux induced by lysosomal dysfunction.

The causative factors of the epigenetic dysregulation of the ALP in PD remain unclear, although there is evidence for a bidirectional relationship between α -syn and the ALP [19, 21]. Decreased autophagic flux results in an accumulation of α -syn [19, 21], and misfolded α -syn itself appears to play an active role in suppressing the ALP [83, 84]. In this way, a genetic and/or epigenetic defects in the ALP leading to α -syn accumulation could lead to further (epigenetic) dysregulation of the ALP. This is supported by our finding that the same ALP genes are disrupted in a mouse model with α -syn overexpression as in the human PD appendix. In addition to the joint contribution of genetic risk factors [23, 25] and α -syn accumulation triggering epigenetic disruption of the ALP, environmental agents [93] and abnormal shifts in the microbiome [94] may play a role, especially because they can impact gut inflammation.

There is a robust interplay between inflammation and the ALP [95]. Autophagy is involved in the induction and suppression of inflammation [95, 96]. It regulates the development, homeostasis, and survival of inflammatory cells and affects the transcription, processing, and secretion of cytokines [95]. Loss of autophagy has proinflammatory consequences, and in the gut, ALP suppression exacerbates the inflammatory effects of DSS colitis [96, 97]. In our study, gut inflammation mediated by DSS largely induced a hypomethylation of the ALP. This is consistent with evidence that activation of autophagy is important for resolving inflammation [95, 96]. Furthermore, DSS colitis effects on the ALP were amplified by synucleinopathy. Synucleinopathy in mice induced a hypermethylation of lysosomal genes which was consistent with that observed in PD. Thus, the exaggerated response to an inflammatory event may be an increased activation of autophagy in an attempt to overcome deficient lysosomal function mediated by α -syn accumulation. Indeed, we find that autophagy genes exhibiting hypomethylation are in common between the PD appendix and mice models of synucleinopathy and gut inflammation, while lysosomal genes are hypermethylated in the PD appendix and by synucleinopathy. Excessive or prolonged autophagy stimulation is detrimental as it can lead to cell death, including in the GI tract (known as autophagic cell death and autosis) [98-101]. Thus, our study suggests that epigenetically-mediated ALP abnormalities in the PD appendix may, in part, be due to an accumulation of α -syn. Given the evidence supporting that inflammation (including in the GI tract) plays a key role in PD pathogenesis [69, 70], it is possible that normative ALP activation needed to resolve inflammation is incapacitated in PD.

Components of the Review: None of this diminishes the results, however, and the completeness of the story the authors are telling overall i.e. that there is widespread hypermethylation in the ALP of PD patient compared to controls in multiple tissues of interest in PD and in a number of overlapping genes across the interrogated tissues and that some of the genes affected are linked to immune function.

A second point the authors are trying to make is about the relevance of aging processes. To do so they investigate whether epigenetic perturbation of the ALP are similar to or related to the known epigenetic DNA remodeling that occurs with aging, as aging is the most consistent risk

factor for PD, may be contributing to the impaired clearance, accumulation, and spread of aggregated α -syn in PD, and has a known effect on DNA methylation. Since they used targeted fine mapping in 521 preselected ALP genes and some well-known PD risk genes, however, they cannot assess epigenetic aging of the tissues overall but only age-related changes in this targeted system. In healthy controls they found a clear shift towards hypermethylation of sites with increasing age in ALP genes in prefrontal cortex neurons. The authors interpreted this as an aging related degradation of the machinery that removes aggregated proteins and invading pathogens therefore aging making the appendix and brain more vulnerable to protein aggregation and infections. I agree that a less well functioning system in older age may increase vulnerability.

Reviewer 3, Comment 3: Interestingly, they however did not find any age-related changes in DNA methylation (an overall directional shift at ALP genes) in the aging appendix or the PFC neurons in PD patients (Fig. 4). This is further supported by the predicted younger age of the ALP machinery in PD compared with controls. The authors interpret this lack of an 'aging effect' of the ALP in PD patients as the attempt of the system to deal with aggregated α -syn in PD by upregulating the ALP, even though this is obviously inadequate as the disease is characterized by more and more aggregated α -syn. On the other hand does this observation not also suggest that the ALP in PD PFC neurons is indeed still able to upregulate in response to disease processes and this would then speak against the hypothesis that it is the ALP that is failing and causing the aggregation of α -syn, etc even though it is part of the picture as it may not be able to upregulate enough to stem the disease process? Also, is the ALP of the neurons only of interest or should we perhaps also to pay attention to the microglia that are not part of this analysis.

Response 3: Following this comment as well as reviewer 2 comment 7 we have made a substantial revision of our aging analysis. We found that epigenetic aging changes of ALP genes exist among the PD samples. In short we found accelerated aging of ALP genes in appendix but lack of normal aging processes in prefrontal cortex neurons (see reviewer 2 response 7 for details). This supports the hypothesis of a failing ALP system in the appendix while the lack of aging changes in the PFC may suggest an ability to upregulate in response to disease processes. We removed the age prediction analysis as we found that its results were misleading. The predicted younger age of the PD samples was an artifact of sample age distribution and the bias of the predictor.

In this study we examined brain neurons, as considerable evidence supports that ALP inactivation contributes to the neuronal α -syn pathology and neurodegeneration; the pathological hallmarks of PD¹⁻⁸. Microglia have a neuroinflammatory role in PD and resting state microglia can engulf α -syn that is released from neurons to reduce α -syn pathology in nigral dopaminergic neurons in mice^{9,10}. However, the effects of ALP dysregulation in microglia on PD pathology is not well understood. There is a recent study in mice demonstrating that activation of selective autophagy in microglia reduces neuron-released α -syn and is neuroprotective¹¹,

though whether this occurs in human brain microglia is unknown, and thus the premise for ALP deficiency in microglia as pathogenic mechanism in PD is not as established. In our experience, microglia are very difficult to isolate from postmortem brain tissue in sufficient numbers to perform an epigenetic analysis (consisting of only ~2% of the brain cell fraction). Thus, a separate, dedicated study of human microglia ALP gene contributions to PD pathology is an innovative (and technically challenging) future direction, though is beyond the scope of this current study.

References

1. Sato S, *et al.* Loss of autophagy in dopaminergic neurons causes Lewy pathology and motor dysfunction in aged mice. *Sci Rep* **8**, 2813 (2018).
2. Laguna A, *et al.* Dopaminergic control of autophagic-lysosomal function implicates Lmx1b in Parkinson's disease. *Nat Neurosci* **18**, 826-835 (2015).
3. Cuervo AM, Stefanis L, Fredenburg R, Lansbury PT, Sulzer D. Impaired degradation of mutant alpha-synuclein by chaperone-mediated autophagy. *Science* **305**, 1292-1295 (2004).
4. Xilouri M, Brekk OR, Polissidis A, Chrysanthou-Piterou M, Kloukina I, Stefanis L. Impairment of chaperone-mediated autophagy induces dopaminergic neurodegeneration in rats. *Autophagy* **12**, 2230-2247 (2016).
5. Ysselstein D, *et al.* LRRK2 kinase activity regulates lysosomal glucocerebrosidase in neurons derived from Parkinson's disease patients. *Nat Commun* **10**, 5570 (2019).
6. Burbulla LF, *et al.* Dopamine oxidation mediates mitochondrial and lysosomal dysfunction in Parkinson's disease. *Science* **357**, 1255-1261 (2017).
7. Dehay B, *et al.* Pathogenic lysosomal depletion in Parkinson's disease. *J Neurosci* **30**, 12535-12544 (2010).
8. Desplats P, *et al.* Inclusion formation and neuronal cell death through neuron-to-neuron transmission of alpha-synuclein. *Proc Natl Acad Sci U S A* **106**, 13010-13015 (2009).
9. George S, *et al.* Microglia affect alpha-synuclein cell-to-cell transfer in a mouse model of Parkinson's disease. *Mol Neurodegener* **14**, 34 (2019).
10. Hirsch EC, Hunot S. Neuroinflammation in Parkinson's disease: a target for neuroprotection? *Lancet Neurol* **8**, 382-397 (2009).
11. Choi I, *et al.* Microglia clear neuron-released alpha-synuclein via selective autophagy and prevent neurodegeneration. *Nat Commun* **11**, 1386 (2020).

Reviewer 3, Comment 4: Finally, I wondered why the authors omitted showing any results for the olfactory bulb for aging in PD and controls.

Response 4: Olfactory bulb sample size is the smallest of the datasets studied. We have added to our manuscript an aging analysis of the ALP in the olfactory bulb of PD patients. We identified 853 loci aging in healthy olfactory bulb samples affecting 325 ALP genes. Similarly to other datasets, we found that epigenetic aging rate correlates with absolute difference between healthy control and PD samples (Supplementary Figure S9).

The following changes were made to the manuscript:

Results, pg. 8. As a comparison, we also examined aging changes in DNA methylation in two brain sample datasets. In prefrontal cortex neurons of 42 control and 52 PD individuals, ages 55-93 we profiled a total 130,733 CpG and 696,665 CpH sites. In olfactory bulb tissue of 14 control and 9 PD individuals, ages 53-92 we profiled a total of 143,553 CpG sites.

Results, pg. 8: In olfactory bulb of controls there were 853 epigenetically aging sites affecting 325 ALP genes (2.6 differentially methylated sites per affected ALP gene with 0.75% average methylation change per year; $q < 0.05$, robust linear regression adjusting for sex and post mortem interval; Supplementary Data 13). No dominant direction of methylation change could be established (OR=1.06, $p = 0.41$; Fisher's exact test).

Results, pg. 9. Absolute cytosine methylation aging rates were significantly positively correlated to absolute methylation changes manifesting in PD, for the appendix, olfactory bulb, and prefrontal cortex neurons ($r > 0.70$, $p < 10^{-15}$; Pearson correlation; Supplementary Figure S9).

Supplementary Data 13 (.csv): DNA methylation changes in aging olfactory bulb of healthy individuals and PD patients.

Figure S9. The relationship between epigenetic changes occurring with aging and epigenetic changes occurring in PD, examining the appendix, olfactory bulb and prefrontal cortex neurons among healthy and PD samples. Each point represents a cytosine having a nominally significant ($p < 0.05$) methylation change in PD and nominally significant aging change in control or PD group. Correlation coefficients and their statistical significance determined using Pearson

correlation. Appendix n = 31 controls, 24 PD; prefrontal cortex n = 35 controls, 52 PD; olfactory bulb n = 14 controls, 9 PD.

Reviewer 3, Comment 5: The authors found far more differentially methylated sites and genes in appendix and olfactory bulb than neurons of prefrontal cortex. I wonder whether this would not be expected as the type of cells is much more homogeneous i.e. only neurons while in the appendix or olfactory tissues we have immune cells mixed and epithelium etc as well etc. Thus, wouldn't we also expect some signal from immune cells in gut and nose that we do not expect to see in neurons? In fact it is a bit surprising that the neurons showed immune tissue relevant methylation changes (TLR9 and GPNMB). The authors mention that even the neurons of PD patients that did not have Lewy pathology (Braak stage 3-4) showed greatly overlapping changes in methylation, suggesting that many of these may precede the arrival of Lewy pathology and imply that this may mean that the ALP 'dysfunction' is at the root cause for the eventually pathology; however, this may not necessarily be the case as not having Lewy bodies does not mean that α -syn is not upregulated yet, rather it might be and the ALP machinery might also be reacting to this only it is possibly still capable of keeping the pathology at bay.

Response 5: In our study, we examined DNA methylation changes at ALP genes in isolated prefrontal cortex neurons from PD patients and controls. In neurons, DNA methylation occurs at both CpG and CpH locations¹. Thus, the prefrontal cortex neuron dataset profiled CpG as well as CpH methylation, which greatly increased the number of statistical tests performed, thereby raising the *p*-value significance threshold after multiple testing correction. Consequently, it is not surprising that fewer cytosines were significantly altered in that analysis. In prefrontal cortex neurons, we profiled 130,733 CpG and 696,665 CpH sites at ALP genes, whereas we profiled 182,024 CpG sites and 143,553 CpG sites at ALP genes in the PD appendix and olfactory bulb, respectively.

Since *TLR9* and *GPNMB* are epigenetically altered in both the PD appendix and brain, one may speculate that this could signify a widespread dysregulation of these specific ALP genes in PD, which may have disease relevance in the brain and in peripheral tissues that highly express α -syn and are connected to the brain by the vagus nerve. In addition, though prevalent in immune cells, TLR9 is present in human neurons, and low levels of GPNMB were detected in degenerating neurons of neurodegenerative disease patients²⁻⁶; both genes are linked to PD and neurodegeneration^{3,5-7}.

We have revised the statement concerning the analysis of the Braak stage 3-4 patients to strictly mention that the DNA methylation changes at ALP genes are present before Lewy pathology onset. In agreement with the reviewer, our discussion mentions that α -syn abnormalities have the potential to drive ALP dysregulation.

Results, pg. 5. These changes significantly overlapped with those identified with the full cohort (OR = 349.36, $p = 1.24 \times 10^{-7}$, Fisher's exact test), suggesting that many of the DNA methylation changes observed precede the onset of Lewy pathology.

References

1. Lister R, *et al.* Global epigenomic reconfiguration during mammalian brain development. *Science* **341**, 1237905 (2013).
2. Mukherjee P, *et al.* SARM1, Not MyD88, Mediates TLR7/TLR9-Induced Apoptosis in Neurons. *J Immunol* **195**, 4913-4921 (2015).
3. Hanke ML, Kielian T. Toll-like receptors in health and disease in the brain: mechanisms and therapeutic potential. *Clin Sci (Lond)* **121**, 367-387 (2011).
4. Shintani Y, *et al.* TLR9 mediates cellular protection by modulating energy metabolism in cardiomyocytes and neurons. *Proc Natl Acad Sci U S A* **110**, 5109-5114 (2013).
5. Satoh JI, Kino Y, Yanaizu M, Ishida T, Saito Y. Microglia express GPNMB in the brains of Alzheimer's disease and Nasu-Hakola disease. *Intractable Rare Dis Res* **8**, 120-128 (2019).
6. Moloney EB, Moskites A, Ferrari EJ, Isacson O, Hallett PJ. The glycoprotein GPNMB is selectively elevated in the substantia nigra of Parkinson's disease patients and increases after lysosomal stress. *Neurobiology of disease* **120**, 1-11 (2018).
7. Nalls MA, *et al.* Identification of novel risk loci, causal insights, and heritable risk for Parkinson's disease: a meta-analysis of genome-wide association studies. *Lancet Neurol* **18**, 1091-1102 (2019).

Components of the Review: Third, the authors compared two synuclein pathology mouse models with wild type mice and also induced gut inflammation. One model are mice overexpressing A30P α -syn and for the other they used virally induced expression of α -syn (rAAV-mediated α -syn overexpression through injection into the cecal patch of wild-type C57BL/6J mice). The virus induced overexpression was accompanied by a differential methylation in a large number of ALP genes in response to α -syn aggregation while the mice overexpressing A30P α -syn did not show any differences in ALP gene methylation compared to wild type mice unless they had undergone an inflammatory challenge. There were also positive correlations and overlap in epigenetically altered ALP genes in response to inflammation and those altered in response to α -syn aggregation by rAAV-mediated overexpression. Also, differentially methylated ALP genes in the PD appendix had gene homologs that were differentially methylated in the mice overexpressing A30P α -syn relative to wild-type mice. However, methylation changes in the ALP induced by gut inflammation did not resemble those seen in the PD appendix. Indeed, the strong ALP hypermethylation in PD was contrary to the ALP hypomethylation mediated by gut inflammation in mice. Nevertheless, pathway analysis revealed that α -syn overexpression in the A30P mouse model recapitulates abnormalities in lysosome function in the PD appendix and affected autophagy. The authors therefore concluded that aberrant α -syn expression and accumulation may contribute to the epigenetic abnormalities in the ALP in the PD appendix and in turn may trigger a hypersensitivity to gut inflammatory events.

Reviewer 3, Comment 6: Given this speculation, wouldn't we expect to see an inflammatory state in the PD gut or appendix compared with controls? Did the authors have any measure of inflammation available for their appendix samples?

Response 6: There were no significant transcription level differences among inflammatory state markers¹ in the PD appendix RNA-seq data.

References

1. Kany S, et al. Cytokines in Inflammatory Disease. *Int J Mol Sci.* 2019;20(23):6008. doi:10.3390/ijms20236008

Reviewer 3, Comment 7: The authors mention that in mice that overexpress α -syn they see a heightened responsivity of the ALP to an inflammatory stimulus. They call this an 'exaggerated response to an inflammatory event as an attempt to overcome deficient ALP function mediated by α -syn accumulation'. However, it is unclear why they consider this response exaggerated and what this would mean for the tissue, i.e. is there a point where too much autophagy would be detrimental? Similarly, they state that "Our results also suggest that the PD appendix is not overtly inflamed, but rather the elevated levels of α -syn may prime for exaggerated and potentially pathological ALP responses to inflammatory triggers". Again, what would a 'pathologic response of the ALP' look like in terms of the tissue i.e. what negative consequences would an upregulation of autophagy have?

Response 7: Excessive or prolonged autophagy stimulation is detrimental as it can lead to cell death, including in the GI tract^{1,95-98}. This process is known as autophagic cell death and autosis⁹⁵⁻⁹⁸. Our discussion has been revised to:

Discussion, pg. 14. Thus, the exaggerated response to an inflammatory event may be an increased activation of autophagy in an attempt to overcome deficient lysosomal function mediated by α -syn accumulation. ... Excessive or prolonged autophagy stimulation is detrimental as it can lead to cell death, including in the GI tract (known as autophagic cell death and autosis) [98-101].

In addition, this comment has been addressed in Reviewer 2, Response 10 and 11.

References

1. Doherty J, Baehrecke EH. Life, death and autophagy. *Nature Cell Biology* **20**, 1110-1117 (2018).
95. Liu Y, et al. Autosis is a Na⁺,K⁺-ATPase-regulated form of cell death triggered by autophagy-inducing peptides, starvation, and hypoxia-ischemia. *Proc Natl Acad Sci U S A* **110**, 20364-20371 (2013).
96. Marino G, Niso-Santano M, Baehrecke EH, Kroemer G. Self-consumption: the interplay of autophagy and apoptosis. *Nat Rev Mol Cell Biol* **15**, 81-94 (2014).

97. Denton D, *et al.* Autophagy, not apoptosis, is essential for midgut cell death in *Drosophila*. *Curr Biol* **19**, 1741-1746 (2009).
 98. Shen T, *et al.* Erbin exerts a protective effect against inflammatory bowel disease by suppressing autophagic cell death. *Oncotarget* **9**, 12035-12049 (2018).
-

Reviewer 3: Comment 8: The authors are presenting odds ratios (OR) as a ‘measure of the magnitude of enrichment of dysregulation’ however, this measure is hard to interpret; i.e. it seems easier to understand an x-fold increase or decrease in methylation over control samples. Maybe the authors could explain more carefully what they are comparing when using such an odds ratio? I.e. what the ratio measure is a ratio of. Also it is unclear how or why this measure as presented in Additional File 8 would suggest an overlap of methylation in genes across tissues (I might misinterpret the numbers in this file as there is no legend for the column titles provided but there are also ORs listed as 0 – does this mean the model did not converge or the ratio was actually zero; for example for MAN2B1 the ORs reported are 0.74, 0, 1.2 and 7.4 that is they are very different and on both sides of the null value of 1, so it is unclear how this can be considered a consistent result for this gene across tissues).

Response 8: Our approach to rank genes based on the odds ratio uses the number of altered cytosines as the measure of importance instead of the magnitude of change. To calculate an odds ratio, we compare the odds of observing a significantly differentially modified site within a gene versus the odds of observing a significantly differentially modified site genome-wide. An odds ratio above 1 indicates that there are more differentially modified sites within a gene than the background rate. Genes that do not have any significant sites will get an odds ratio of 0. The odds ratio estimates may be unstable when only few observations are available. Therefore, we used one-sided Fisher’s exact test to obtain enrichment *p* values that take into account the number of observations available for that gene. We rank the genes by their enrichment *p* values and aggregate the rankings using an unbiased rank aggregation algorithm, which brings to the top the genes that rank consistently higher across all used datasets than expected by chance alone. In the case of MAN2B1, the ORs that are less than 1 do not indicate an opposite directional effect but rather a lack of enrichment. This gene is significantly modified in all but one tissue and, according to the robust rank aggregation algorithm, ranks quite high overall.

We have made the following adjustment to the methods:

Methods, pg. 25. For the PD appendix, prefrontal cortex, and olfactory bulb datasets, the odds ratio of enrichment of ALP genes with significant differentially methylated cytosines was determined. To calculate an odds ratio, we compare the odds of observing a significantly differentially modified site within a gene versus the odds of observing a significantly differentially modified site genome-wide. Odds ratios were tested for statistical significance of enrichment using one-sided Fisher’s exact test. Then, for each dataset, the genes were ranked according to their enrichment *p*-value. Using the robust rank aggregation algorithm [54] we obtained the ranking of ALP genes consistently disrupted across the PD appendix and brain tissues. The

approach brings to the top the genes that rank consistently higher across all used datasets than expected by chance alone.

Reviewer 3, Comment 9: Consider showing some Venn diagrams for the number of overlapping changes in methylation of genes across tissues etc.

Response 9:

We have added Supplementary Data 17 that lists for each gene the odds ratio of enrichment with significantly differentially methylated cytosines in each tissue and experiment that was performed. In addition, the sign of the odds ratio value indicates whether the gene is predominantly hypermethylated (positive) or hypomethylated (negative).

Reviewer 3, Comment 10: It is unclear what the y-axis of figure 4D stands for i.e. what kind of measure is meant by an 'absolute PD effect'?

Response 10: PD diagnosis effect is the mean methylation difference between healthy control and PD individuals and could also be described as absolute fold change (FC). Figure 4D has been moved to Supplementary Figure S9, and the axis labels updated to "Epigenetic aging rate, absolute FC" and "PD diagnosis effect, absolute FC".

Reviewer 3, Comment 11: Also figure 5e is uninformative and can be removed

Response 11: We have removed Fig. 5e from the manuscript.

Reviewer 3, Comment 12: The authors state that "epigenetic changes in the ALP in the appendix may be a potential biomarker for disease risk and progression by serving as a proxy for the epigenetic status of ALP genes in the brain"; it is not clear how this tissue (appendix) could indeed be a useful biomarker though as it is unlikely that someone concerned about PD would undergo an appendectomy and it also is unclear at what stage of PD the appendix and its methylation status would indeed reflect changes in the brain. I suggest to reword or strike this sentence.

Response 12: Please see response to Reviewer 1, Response 3.

Reviewer #4 (Remarks to the Author):

In this manuscript, Gordevicius et al. performed an extensive study of DNA methylation of the autophagy-lysosome pathway (ALP) in the Parkinson's disease appendix. Through the targeted deep DNA methylation sequencing of a relatively large number of samples from the appendix and brain of PD and control subjects as well as the mouse equivalent of the human appendix of control, A30P alpha-synuclein mice with and without gut inflammation, the team identified the synchronized hypermethylation at the ALP in the appendix and brain in PD. Specifically, they discovered:

- 1) ~63% of the 521 ALP genes were subject to DNA methylation dysregulation in the appendix in PD versus control and such DNA methylation dysregulation was specific to lysosome genes and affected predominantly their active promoter regions.
 - 2) ~68% and 11% of the 521 ALP genes were subject to DNA methylation dysregulation in the PD olfactory neurons and prefrontal cortex, respectively. And those gene signatures were enriched in the ALP genes subject to DNA methylation dysregulation in the appendix, suggesting the DNA methylation dysregulation is shared by the appendix and brain.
 - 3) Protein abundance changes in the appendix between PD and control were highly positively correlated with those in the prefrontal cortex.
 - 4) A large number of genes were also subject to DNA methylation changes in ageing in the healthy appendix and prefrontal cortex and these changes were highly correlated with those between PD and control, suggesting ageing is one big contributor of PD pathogenesis.
 - 5) in vivo experiments showed that gut inflammation substantially induced DNA methylation dysregulation (hypomethylation) in the cecal patch (the mouse equivalent of the human appendix), which is different from the widespread hypermethylation in the PD appendix.
- Overall, this is a compelling study and presents a very clear picture of epigenetic regulation of the ALP genes in the PD appendix and brain. I have only a few minor concerns listed below:

Response: We appreciate the reviewer's interest and positive review of our study

Reviewer 4, Comment 1: How do the lysosome genes subject to DNA methylation dysregulation in the PD appendix overlap those differentially expressed between PD and control in the PD appendix? The two sets of genes correspond to the same pathway but they may not strongly overlap with each other.

Response 1: Please see the Reviewer 2, Response 2.

Reviewer 4, Comment 2: There are many gene signatures from the experiments carried out in the study and often only pairwise comparisons were performed. It will be beneficial to prepare a heatmap to show the methylation patterns of the 521 ALP gene under all the experiments.

Response 2: Please see Reviewer 3 Response 9.

Reviewer 4, Comment 3: The authors should compare their findings with those from the genome wide DNA methylation data in PD. I suspect that many findings here may not be significant at the genome-wide analysis.

Response 3:

We used two approaches to answer the reviewer's question on whether our epigenetic changes at ALP genes are observed in genome-wide studies. First, we examined a genome-wide dataset comparing epigenetic changes in isolated prefrontal cortex neurons of PD patients and controls¹. This dataset profiled DNA methylation at 904,511 methylated cytosine sites, examining all brain enhancers and promoters identified in the NIH Epigenomics RoadMap in prefrontal cortex neurons of 57 PD patients and 48 controls. We found that there was a significant overlap of the top loci identified in our ALP dataset with those of the genome-wide dataset (OR = 4.49, $p = 1.23 \times 10^{-6}$; top 5000 loci examined in each dataset). Thus, genome-wide analysis in PD brain neurons supports an epigenetic disruption of ALP gene elements.

In addition, in our PD appendix ALP dataset, we determined that the DNA methylation alterations in the PD appendix were specifically enriched within 20 kb of lysosomal gene start sites (FDR $q = 2.9 \times 10^{-4}$) and were not enriched at distant locations. DNA methylation changes in the PD appendix were specific to lysosomal genes, as no enrichment was observed for non-lysosomal genes (Supplementary Figure S4). Hence, DNA methylation abnormalities in the ALP of the PD appendix particularly affect lysosomal genes. Finally, an unbiased analysis of the entire transcriptome shows prominent lysosomal pathway abnormalities in the PD appendix.

We have added the following text to the Results:

Results, pg. 6. We compared the differentially methylated sites at ALP genes identified in the PD prefrontal cortex neurons to those identified in a genome-wide analysis [53]]. The genome-wide dataset used profiled DNA methylation at all brain enhancers and promoters, examining 904,511 methylated cytosine sites in prefrontal cortex neurons of 57 PD patients and 48 controls. We found that there was a significant overlap of the top loci identified in our ALP dataset with those of the genome-wide dataset (OR = 4.49, $p = 1.23 \times 10^{-6}$; top 5000 loci examined in each dataset). Thus, comparison with a genome-wide analysis supports an epigenetic disruption of ALP gene elements in PD brain neurons.

References:

1. Marshall LL, Killinger BA, Ensink E, et al. Epigenomic analysis of Parkinson's disease neurons identifies Tet2 loss as neuroprotective [published online ahead of print, 2020 Aug 17]. *Nat Neurosci*. 2020. doi:10.1038/s41593-020-0690-y

REVIEWER COMMENTS:

Reviewer #2 (Remarks to the Author):

The authors have addressed in great detail the suggestions of the reviewers and have answered most of the issues raised. I believe that the manuscript is ready for publication.

Reviewer #3 (Remarks to the Author):

The authors have done a remarkable job at answering a long list of detailed questions and the manuscript has become stronger and the writing clearer. Particularly convincing are the additional details presented concerning ALP epigenetic, transcriptomic and proteomic differences between controls and PD cases as well as the complementary results based on the mouse models. I also agree with the other reviewer that a reduction in the scope of this paper would make it easier to understand and follow and I believe that the 'aging' angle is a distraction and not yet well conceived. I believe that I now understand better what the authors are referring when they use 'aging' in this article. However, some of what I learned concerns me enough to suggest to either drop the 'aging' related work from this manuscript or to do a more in-depth assessments of the 'aging' effect. In fact, it would be more than sufficient and less distracting if the authors would focus on the results that support the title "Epigenetic inactivation of the autophagy-lysosomal system in the Parkinson's disease 1 appendix".

First, the authors are actually not modeling 'aging' as they do not have multiple longitudinal samples from the same individual (as the person is aging) to assess methylation changes occurring with 'aging'. Rather they are using cross-sectionally collected samples with a broad age range and determine methylation levels according to age of the subject (not aging of the subject) i.e. basically exploring an 'age effect' not 'aging effect'. Also, they are doing this with rather small samples of controls in whom they attempt to determine 'healthy aging'. At the very least I would like to discourage the authors from using the term 'aging' anywhere and to rather refer to 'age at which the sample was taken' etc to remind the reader that this is a cross-sectional comparison simply of samples taken at different ages rather than the 'aging' of a tissue.

It is well-known that methylation changes occur with aging (as longitudinal epigenome wide methylation studies have shown) but how this in fact translates to hyper- or hypomethylation of individual genes is less well known. The hypothesis that the ALP would also be affected by changes in methylation with aging is a valid proposal. However, it is questionable whether a background 'aging' related change in specific genes can be determined from the small number of samples of 'healthy' controls the authors have available. It would be much more convincing if the authors had used a much larger sample – possibly publicly available EWAS data or longitudinal samples from 'aging' cohorts – to first establish an 'aging' pattern in these genes. Even though this approach would not have the deep sequencing data of the ALP system, existing arrays might still allow us to establish whether certain genes become generally hyper- or hypomethylated with advancing age.

Second, it is not clear from the current description of the samples whether the age range of the PD samples corresponds to the age range of the control samples or not. For example, the authors provide information on "51 healthy and 24 PD appendix samples, ages 18-92"; however, it is not clear whether any PD samples were obtained at ages younger than 50 years or how much the age range of controls overlaps with the age range of cases (the prefrontal cortex samples of 42 control and 52 PD individuals had an age range of 55-93)? In fact, Figure 4E suggests that controls and cases have a very different age range at both the lower and the upper end and it seems that the high correlations of methylation with age in control are at least partially driven by low and high age individuals. I recommend to restrict the controls to the exact same age range as the cases and show that there is still an 'age effect' on methylation in controls that is not seen in cases. Otherwise, what is now looking

like a large difference between controls and PD cases in the age relatedness of their methylation may just be an artifact of the different age ranges being compared. Also, it is unclear whether the age-related changes observed in controls can indeed be extrapolated out to patients without making major assumptions that cannot be validated such as whether or in what age range changes are linear. Aging is generally accompanied by global hypomethylation and some local hypermethylation. Thus, it is unclear whether it is adequate to consider an age-related loss of methylation – as done by the authors - a 'healthy' phenotype. I.e. the authors call a higher level of methylation observed in PD patients (i.e. higher than expected from control samples) a 'hypermethylation' of the ALP in PD and interpret this as a sign of disease rather than less of a loss of methylation with aging i.e. an increased ability of PD patients to maintain their original methylation status while aging; this is a possibility the authors even considered i.e. a compensatory mechanism due to disease related challenges that may or may not be adequate. Thus, there could be a confounding of true 'hypermethylation' of genes in response to disease and a lack in the loss of aging related methylation as a compensatory mechanism in PD.

Third, it is probably not justified to state that the appendix samples represent 'healthy aging' as these were surgical samples obtained from individuals undergoing a right hemicolectomy for intestinal cancer not involving the appendix. Even though the appendix was incidentally removed and histologically confirmed to be normal findings such as reported here for calcium-binding and coiled-coil domain-containing protein 2 (CALCOCO2) - a key regulator of inflammation in Crohn's disease – might just signify global changes affecting the colon of a cancer patient who may have suffered from Crohn's disease known to greatly increase the risk of colon cancer even if the appendix consisted of normal tissue (this is especially of concern for young onset colon cancer patients who would be expected to have suffered from Crohn's disease). Similarly, the finding that 'ALP genes most affected by aging in the human appendix (and cortical neurons) are related to selective autophagy, inflammation, and physiological neuronal activity and survival' might also be, at least partially, due to the fact that the 'healthy' appendix samples were obtained from colon cancer patients who might have had upregulated inflammation throughout the colon (outside of the cancerous tissue) including the appendix as it has also been shown that inflammation is often associated with DNA hypermethylation of specific genes reported for ulcerative colitis. Furthermore, these cancer patients and Crohn's patients may have been on special diets that could have been influencing methylation patterns among the controls.

While I agree with the authors' conclusion that it is possible that "advanced age may place individuals at greater risk for PD, it is not clear that they have the data to conclude that the ALP in PD patients fails to exhibit normative epigenetic changes with 'aging' because they 'do not undergo the same extent of hypermethylation as in 'healthy aging', in a futile attempt to compensate for the decrease in autophagic flux induced by lysosomal dysfunction'. An age effect might simply not be observed in PD cases due to a lack of statistical power within a narrower age range for the cases.

Finally, please add some numbers into the abstract, such as sample Ns and ORs or fold changes in methylation.

A minor point to correct: some of the odds ratios (OR) measures the authors are presenting (with ORs below 1 or negative values) are not ORs but 'log ORs' (or beta values from a regression estimating ORs) as ORs that have a null value of 1 and an OR below one would indeed reflect a negative (protective) association. I think the reply to my question on page 33 referring to the meaning of these 'ORs' is incorrect i.e. "Genes that do not have any significant sites will get an odds ratio of 0 " should read 'log OR of 0' and "In the case of MAN2B1, the ORs that are less than 1 do not indicate an opposite directional effect but rather a lack of enrichment." should read the 'log ORs that are less than 1'. Please carefully scan your text, tables, graphs, and data files (where I found negative values reported in columns titled 'OR' which should instead be labelled 'log OR') and correct these where needed.

Also Figure 4F seems to be lacking a legend description.

Beate Ritz

Reviewer #4 (Remarks to the Author):

In this study, Gordevicius et al. identified the epigenetic alterations in the autophagy-lysosome pathway (ALP) genes (predominantly in the promoter region) in the Parkinson's disease appendix that lead to transcriptional changes validated by RNAseq and proteomic analyses. Such alterations are largely recapitulated in PD and aging brains, and in mice with alpha-synuclein overexpression and induced gut inflammation. The amount of work is very impressive. However, I have several major and minor concerns listed below.

1. Given the massive epigenetic dysregulation found in the appendix and the neurons in the cortex/olfactory bulb between PD patients and controls, have the authors examined the enzymes related to DNA methylation (e.g., TETs and DNMTs)? Are there any significant difference in the protein or the methylation levels of these genes between PD and control?
2. Previous work (<https://www.nature.com/articles/nn.3607>) has demonstrated the similarity and differences between CpG and CpH methylations in adult mammalian brains. Are these two classes of methylations differentially regulated in PD and aging?
3. In Fig. 5b, the concordance is only significant between PD appendix and A30P_water_WT_water. Does this indicate that DSS-induced gut inflammation and rAAV synuclein overexpression didn't cause PD-like methylation changes? Since A30P is a whole-body overexpression model, does it indicate a bidirectional influence from the brain as local overexpression by rAAV did not recapitulate the the methylation changes in PD appendix?
4. The number of ALP genes was inconsistent. In line 101, "521 genes reported in publicly available human autophagy and lysosomal [40] databases as well as PD risk genes", while line 421, " We fine-mapped DNA methylation changes at 571 ALP genes in the mouse cecal patch".
5. In Fig 6, environmental factors should be included in the proposed model.
6. Fig 2c, it would be more informative if the hypermethylation and hypomethylation are presented separately.

Reviewer #5 (Remarks to the Author):

This is a very interesting manuscript that can have a major impact on PD research. It can change our views on how PD begins and develops. However, there are several issues require more work and revisions, as detailed in the numbered list below:

Majors:

1. As the authors said, "Our analysis of differential methylation in the PD appendix was controlled for sample age, sex, postmortem." However, the statistical analysis of demographic information between the different groups was not shown, such as sex, age, Hoehn-Yahr stage.
2. I don't quite understand why the authors choose prefrontal cortex in this study. In the Introduction section, they explained "We then identify whether similar changes are mirrored in neurons of the prefrontal cortex and the olfactory bulb, another proposed starting point for PD [41, 42]." However, in these two references, the involvement of prefrontal cortex pathology was shown at stages 5 and 6, which at the later phase of PD.
3. In the manuscript, the authors chose appendix, olfactory bulb, and prefrontal cortex as the study regions. The results are good enough, but I also want to know, when the methylation alteration was appeared in these regions, what are the changes of methylation levels in the substantia nigra and striatum, the most important pathological areas in PD.
4. In line 401-402, page 9, the author said, "Together, this suggests that in patients, PD disease processes disrupt normal aging changes in ALP function." Is it possible that the change in ALP function causes the occurrence of PD, rather than the consequences of PD, what do the authors think about this question?
5. In vivo, the authors demonstrated the methylation alteration at ALP genes in animal models, but have the authors detected the changes of autophagy and lysosomal markers, such as LC3, p62, and Beclin-1 to confirm whether there is a dysfunction of autophagy flow? And how do the authors confirm

the causal relationship between methylation alteration and dysfunction of autophagy flow?

6. In the manuscript, the authors want to explore the underlying mechanism of the transfer of α -syn aggregates from appendix to brain in PD. However, in vivo study only showed in situ changes, what about the α -syn pathology changes in brain? How do you prove this animal model is successful?

7. In the Methods section, the authors said, "Within each data set all measurements were taken from distinct samples." But as the authors showed, methylation alteration of ALP can occur in areas such as the appendix or olfactory bulb at early stage of PD, but where do the changes in the same population weigh more and where do they originate? How do you do the comparison if not choose the same population?

Minors:

1. The running title is "Widespread silencing of autophagy-lysosomal genes in the Parkinson's disease gut and brain", but the title is "Epigenetic inactivation of the autophagy-lysosomal system in the Parkinson's disease appendix". What do you want to emphasize in this manuscript? The different statements make me feel confused.

2. The authors demonstrated silencing of autophagy-lysosomal genes in PD appendix and brain, what about other α -synucleinopathies, including dementia with Lewy bodies and multiple system atrophy, etc.? Whether the same changes appeared in these diseases?

3. In vivo study, why the authors choose rAAV vector that overexpressed human α -syn instead of mouse α -syn?

4. In vivo study, why the authors choose mice with different ages in gut inflammation model (12 weeks old) and vector-mediated α -syn overexpression model (8 weeks old)?

5. The schematic diagram in Fig.6 is somewhat confusing. The authors should make it more clearly and shows the relationship between the appendix and brain.

6. In Figure S13B, the distribution of aggregated α -syn in enteric neurons is not displayed clearly, the nucleus and axons should also be co-stained with phosphor-serine 129.

7. In line 83-86, page 2, "Taken together, the apparent relationships between PD, the ALP, and the development and spread of α -syn pathology suggest that disruption of the ALP in the aging appendix could be an important mechanism underlying the transfer of α -syn aggregates from appendix to brain in PD." It is not appropriate to use "between" for comparison of three subjects.

8. In line 169, page 4, "OR = 2.12, p = 3.67 x 10⁻⁸", the "x" should change to "×", the same mistake was repeated in later pages.

9. In line 263, page 6, "genome-wide analysis [53].", the later "]" should be deleted.

10. In line 404, page 9, " α -Syn accumulation", the " α -Syn" should change to " α -syn", the same mistake was repeated in later pages.

REVIEWER COMMENTS

Reviewer #2 (Remarks to the Author):

The authors have addressed in great detail the suggestions of the reviewers and have answered most of the issues raised. I believe that the manuscript is ready for publication.

Response: We thank the reviewer for their enthusiasm for our study and the time taken to review and comment which led to improved quality of the manuscript.

Reviewer #3 (Remarks to the Author):

The authors have done a remarkable job at answering a long list of detailed questions and the manuscript has become stronger and the writing clearer. Particularly convincing are the additional details presented concerning ALP epigenetic, transcriptomic and proteomic differences between controls and PD cases as well as the complementary results based on the mouse models.

Response: We thank the reviewer for their enthusiasm for our study and the time taken to review and comment.

Reviewer 3 comment 1: I also agree with the other reviewer that a reduction in the scope of this paper would make it easier to understand and follow and I believe that the ‘aging’ angle is a distraction and not yet well conceived. I believe that I now understand better what the authors are referring when they use ‘aging’ in this article. However, some of what I learned concerns me enough to suggest to either drop the ‘aging’ related work from this manuscript or to do a more in-depth assessments of the ‘aging’ effect. In fact, it would be more than sufficient and less distracting if the authors would focus on the results that support the title “Epigenetic inactivation of the autophagy–lysosomal system in the Parkinson’s disease 1 appendix”.

Response 1: Epigenetic age acceleration is known to be associated with Parkinson’s disease¹. Decreasing activity of the ALP is closely related to aging, which itself is the strongest risk factor for idiopathic PD [28]. Therefore, it seems worthwhile to explore the relationships between PD, ALP, and age. In this context, we believe that it is necessary to investigate what age-related effects are present in our data. Following the suggestions and comments of all reviewers, the aging section has markedly improved. We found that the loci that correlate with chronological age in non-PD appendix samples are also more likely to be the ones affected in PD vs Control comparison. Our analysis is the largest methylation study involving appendix tissue and proposes that disruption of the ALP in the aging appendix could be an important mechanism underlying the accumulation and spread of α -syn aggregates from appendix to brain in PD.

References

1. S. Horvath, B. R. Ritz, Increased epigenetic age and granulocyte counts in the blood of Parkinson's disease patients. *Aging (Albany, NY)*. 7, 1130–1142 (2015). PMID 26655927.

Reviewer 3 comment 2: First, the authors are actually not modeling 'aging' as they do not have multiple longitudinal samples from the same individual (as the person is aging) to assess methylation changes occurring with 'aging'. Rather they are using cross-sectionally collected samples with a broad age range and determine methylation levels according to age of the subject (not aging of the subject) i.e. basically exploring an 'age effect' not 'aging effect'. Also, they are doing this with rather small samples of controls in whom they attempt to determine 'healthy aging'. At the very least I would like to discourage the authors from using the term 'aging' anywhere and to rather refer to 'age at which the sample was taken' etc to remind the reader that this is a cross-sectional comparison simply of samples taken at different ages rather than the 'aging' of a tissue.

Response 2: We appreciate the reviewer's observation that the age effects we are measuring are common to the sampled population and may possibly differ from those one would observe if samples were taken longitudinally. We have adjusted the manuscript to use 'age effect' throughout to clarify that this is a cross-sectional comparison study.

Changes made to the manuscript: Results, pg. 8-9; Discussion pg. 11, 13.
Supplementary Figures S7, S8, S9, S10. Supplementary Data 11, 12, 13, 14. Methods pg. 22, 27.

Reviewer 3 comment 3: It is well-known that methylation changes occur with aging (as longitudinal epigenome wide methylation studies have shown) but how this in fact translates to hyper- or hypomethylation of individual genes is less well known. The hypothesis that the ALP would also be affected by changes in methylation with aging is a valid proposal. However, it is questionable whether a background 'aging' related change in specific genes can be determined from the small number of samples of 'healthy' controls the authors have available. It would be much more convincing if the authors had used a much larger sample – possibly publicly available EWAS data or longitudinal samples from 'aging' cohorts – to first establish an 'aging' pattern in these genes. Even though this approach would not have the deep sequencing data of the ALP system, existing arrays might still allow us to establish whether certain genes become generally hyper- or hypomethylated with advancing age.

Response 3: While there is a body of work about age-related methylation changes, including evidence of DNA hypermethylation associated with a decrease of autophagic activity¹, it is also known that age-related methylation patterns are tissue-specific^{2,3}. Therefore, we agree with the reviewer that not much is known about age effects on the methylation of ALP genes specifically in the appendix. There is no publically available The EPIC array EWAS data for the appendix tissue. In this study we used 51 control appendix samples to establish patterns of methylation associated with age, and to the best of our knowledge, this is the largest study of methylation in the human appendix. The bisulfite padlock technology that we used provides deep coverage of

300kbp regions surrounding ALP genes which would not be possible with EPIC arrays. Thus we believe our age-effect methylation data in appendix is a valuable contribution to our understanding of DNA methylation changes with age despite its size limitations.

References:

1. Jiang S, Guo Y. Epigenetic Clock: DNA Methylation in Aging. *Stem Cells Int.* 2020 Jul 8;2020:1047896. doi: 10.1155/2020/1047896. PMID: 32724310.
2. Jung SE, Shin KJ, Lee HY. DNA methylation-based age prediction from various tissues and body fluids. *BMB Rep.* 2017 Nov;50(11):546-553. doi: 10.5483/bmbrep.2017.50.11.175. PMID: 28946940.
3. Sliker RC, Relton CL, Gaunt TR, Slagboom PE, Heijmans BT. Age-related DNA methylation changes are tissue-specific with ELOVL2 promoter methylation as exception. *Epigenetics Chromatin.* 2018 May 30;11(1):25. doi: 10.1186/s13072-018-0191-3. PMID: 29848354.

Reviewer 3 comment 4: Second, it is not clear from the current description of the samples whether the age range of the PD samples corresponds to the age range of the control samples or not. For example, the authors provide information on “51 healthy and 24 PD appendix samples, ages 18-92”; however, it is not clear whether any PD samples were obtained at ages younger than 50 years or how much the age range of controls overlaps with the age range of cases (the prefrontal cortex samples of 42 control and 52 PD individuals had an age range of 55-93)? In fact, Figure 4E suggests that controls and cases have a very different age range at both the lower and the upper end and it seems that the high correlations of methylation with age in control are at least partially driven by low and high age individuals. I recommend to restrict the controls to the exact same age range as the cases and show that there is still an ‘age effect’ on methylation in controls that is not seen in cases. Otherwise, what is now looking like a large difference between controls and PD cases in the age relatedness of their methylation may just be an artifact of the different age ranges being compared. Also, it is unclear whether the age-related changes observed in controls can indeed be extrapolated out to patients without making major assumptions that cannot be validated such as whether or in what age range changes are linear.

Response 4: We thank the reviewer for the opportunity to clarify the age distribution in our sample cohorts. Our age effect analysis was performed on 51 control samples with age ranging from 18 to 92 years to identify the loci that correlate with age. It was a separate analysis from PD case / control study where we investigated 19 control samples and 24 age-matched PD samples and we also controlled for age in the models. We then overlapped the loci identified by the two analyses and found considerable concordance. This analysis approach does not require age matching of samples but benefits from the wider age range of the control samples.

Next, we compared the magnitude of the age effect in control and PD sample groups. In this case, the age range of PD samples was narrower than that of controls. We found more and stronger magnitude age effects in appendix PD than in control samples. Following the reviewer’s suggestion, we restricted the age of control samples to those older than 62 (N=15) and repeated the analysis. We confirmed hypermethylation of promoters and again found that

absolute age effects are stronger among PD than among control samples (Supplementary Data 20, Supplementary Figure S18).

We made the following updates to the manuscript:

Results, pg. 8: For this analysis, we examined DNA methylation at the 521 ALP genes in 51 healthy (ages 18-92) and 24 PD (ages 62-91) appendix samples and profiled a total of 181,151 CpG sites.

Results, pg. 10: To ascertain that our findings are not influenced by the wide age range of appendix controls, we repeated the same analysis using only appendix samples older than 62 years. We confirmed hypermethylation of promoters and again found absolute age effects to be stronger among PD samples (Supplementary Data 20, Supplementary Figure S18). Taken together, this suggests that in patients, PD disease processes disrupt normal age related changes in ALP function.

Reviewer 3 comment 5: Aging is generally accompanied by global hypomethylation and some local hypermethylation. Thus, it is unclear whether it is adequate to consider an age-related loss of methylation – as done by the authors - a ‘healthy’ phenotype. I.e. the authors call a higher level of methylation observed in PD patients (i.e. higher than expected from control samples) a ‘hypermethylation’ of the ALP in PD and interpret this as a sign of disease rather than ~~less of a~~ loss of methylation with aging i.e. an increased ability of PD patients to maintain their original methylation status while aging; this is a possibility the authors even considered i.e. a compensatory mechanism due to disease related challenges that may or may not be adequate. Thus, there could be a confounding of true ‘hypermethylation’ of genes in response to disease and a lack in the loss of aging related methylation as a compensatory mechanism in PD.

Response 5: The reviewer asks whether the increased methylation in PD compared to controls could reflect a loss of normal hypomethylation with age rather than “true” hypermethylation with disease. This is an interesting suggestion. In the manuscript we find that the control appendix does not have a dominant direction of age-related modification change although promoters are hypermethylated. The control prefrontal cortex neurons do exhibit dominant hypermethylation with age (Fig. 4a). As the reviewer has observed, some genes experience hyper-methylation with age (specifically selective autophagy and macroautophagy), whereas others may experience hypo-methylation; thus we do not observe a global age-related loss of methylation in healthy individuals. In addition, when comparing the control and PD samples, we observe global hyper-methylation in both PD appendix and PD brain; by controlling for age in this model we suggest that this increase in methylation is disease-related, not reflective of normal aging. In the discussion pg. 13 we conclude that It may be the case that, in PD, various ALP pathways (e.g., macroautophagy, selective autophagy) do not undergo the same extent of hypermethylation as in healthy aging, in a futile attempt to compensate for the decrease in autophagic flux induced by lysosomal dysfunction.

Reviewer 3 comment 6: Third, it is probably not justified to state that the appendix samples represent 'healthy aging' as these were surgical samples obtained from individuals undergoing a right hemicolectomy for intestinal cancer not involving the appendix. Even though the appendix was incidentally removed and histologically confirmed to be normal findings such as reported here for calcium-binding and coiled-coil domain-containing protein 2 (CALCOCO2) - a key regulator of inflammation in Crohn's disease – might just signify global changes affecting the colon of a cancer patient who may have suffered from Crohn's disease known to greatly increase the risk of colon cancer even if the appendix consisted of normal tissue (this is especially of concern for young onset colon cancer patients who would be expected to have suffered from Crohn's disease). Similarly, the finding that 'ALP genes most affected by aging in the human appendix (and cortical neurons) are related to selective autophagy, inflammation, and physiological neuronal activity and survival" might also be, at least partially, due to the fact that the 'healthy' appendix samples were obtained from colon cancer patients who might have had upregulated inflammation throughout the colon (outside of the cancerous tissue) including the appendix as it has also been shown that inflammation is often associated with DNA hypermethylation of specific genes reported for ulcerative colitis. Furthermore, these cancer patients and Crohn's patients may have been on special diets that could have been influencing methylation patterns among the controls.

Response 6: The proposal that the appendix control samples may not correspond to healthy appendix is valid yet difficult to verify with the data at hand. We were limited by the tissue available, and as the reviewer notes, we controlled for confounding factors as much as possible by confirming that the appendix removal was incidental and the tissue was histologically normal. To address the reviewers concerns, we compared whether the age-associated genes in the control appendix tissue correlated with age-associated genes in prefrontal cortex neurons of healthy individuals (who, to the best of our knowledge, were not all suffering from colon cancer and/or Crohn's disease). We verified that the age-associated genes overlap (OR=2.89, $p = 5 \times 10^{-8}$; Fisher's exact test) in appendix and prefrontal cortex neurons and their absolute age effect magnitudes correlate ($r = 0.31$, $p = 0.002$; Spearman correlation). This suggests that, at least in part, the control appendix samples do exhibit healthy aging. We have added these results and a discussion of this limitation to the manuscript, and thank the reviewer for the opportunity to strengthen the discussion of this section of our results.

We made the following revisions to the manuscript:

Results, pg. 9. Interestingly, age-associated genes in appendix and prefrontal cortex neurons overlapped (OR=2.89, $p = 0.5 \times 10^{-7}$; Fisher's exact test) and their absolute age effect magnitudes correlated ($r = 0.31$, $p = 0.002$; Spearman correlation).

Discussion, pg 14. Our study of normal age-related methylation changes in appendix is somewhat limited by potential confounding factors in the individuals from which it was obtained. Although the control appendix tissue was confirmed to be histologically normal, the patient diagnosis of intestinal cancer leading to incidental appendix removal

may have impacted some of the methylation changes observed, particularly those related to inflammatory pathways. Nevertheless, we verified that age-related genes observed in these appendices overlap with those seen in prefrontal cortex neurons from other control individuals, suggesting that the control appendix samples exhibit healthy aging, at least in part. Follow-up studies in a second cohort of normal appendix would further validate our results and strengthen our understanding of normal age-related changes of methylation in the appendix.

Reviewer 3 comment 7: While I agree with the authors' conclusion that it is possible that "advanced age may place individuals at greater risk for PD, it is not clear that they have the data to conclude that the ALP in PD patients fails to exhibit normative epigenetic changes with 'aging' because they 'do not undergo the same extent of hypermethylation as in 'healthy aging', in a futile attempt to compensate for the decrease in autophagic flux induced by lysosomal dysfunction'. An age effect might simply not be observed in PD cases due to a lack of statistical power within a narrower age range for the cases.

Response 7: We thank the reviewer for this comment and the opportunity to strengthen our claim. The prefrontal cortex neurons data was generated from a comparable number of control and PD samples (42 control, 52 PD). In the control PFC neurons, we do see an age effect (hypermethylation), specifically at selective autophagy and macroautophagy genes, while a hypermethylation trend could not be established among PD samples. We agree with the reviewer that the age-related hypermethylation in the appendix of PD samples may have not been observed due to a smaller sample size and more narrow age range of cases compared to controls. In order to show that the sample size is sufficient, we repeated our age effect analysis using only the control samples above 62 years of age (N = 15) which resulted in roughly the same age range of control and PD samples. We confirmed hypermethylation of promoters and again found absolute age effects to be stronger among PD samples (Supplementary Data 20, Supplementary Figure S18, see also **Reviewer 3 Response 4**).

Reviewer 3 comment 8: Finally, please add some numbers into the abstract, such as sample Ns and ORs or fold changes in methylation.

Response 8: We appreciate the reviewer's suggestion and have made the following change to the abstract:

Abstract, pg. 2: We systematically examined epigenetic alterations in the ALP by deep sequencing DNA methylation at 521 ALP genes in the appendix of 24 PD patients and 19 controls as well as in neurons isolated from the prefrontal cortex of 52 PD patients and 42 controls. We identified aberrant methylation at 928 cytosines affecting 326 ALP genes in the PD appendix and widespread hypermethylation that is recapitulated in the PD brain.

Reviewer 3 comment 9: A minor point to correct: some of the odds ratios (OR) measures the authors are presenting (with ORs below 1 or negative values) are not ORs but 'log ORs' (or beta values from a regression estimating ORs) as ORs that have a null value of 1 and an OR below one would indeed reflect a negative (protective) association.

Response 9: Thank you for your observation. We took care to report ORs in the text of the manuscript, and we used logOR (labeled as "Odds ratio, log") in the figures where it was more appropriate for visualization. Log transformed ORs are symmetric around 0 and are thus easier to interpret visually. We have carefully reviewed the paper and supplementary material and made corresponding updates to attribute names in Supplementary Data 17.

Reviewer 3 comment 10: I think the reply to my question on page 33 referring to the meaning of these 'ORs' is incorrect i.e. "Genes that do not have any significant sites will get an odds ratio of 0" should read 'log OR of 0' and "In the case of MAN2B1, the ORs that are less than 1 do not indicate an opposite directional effect but rather a lack of enrichment." should read the 'log ORs that are less than 1'.

Response 10: We regret the confusion. To compute an odds ratio, one has to establish the odds of observing a significantly differentially modified site within a gene versus the odds of observing significantly differentially modified site genome-wide. For a gene that has no significant sites, the odds of observing a significant site is 0. Consequently, the odds ratio is 0.

Reviewer 3 comment 11: Please carefully scan your text, tables, graphs, and data files (where I found negative values reported in columns titled 'OR' which should instead be labelled 'log OR') and correct these where needed.

Response 11: We are thankful for the observation which will help increase the quality of the manuscript.

The following changes were made:

In **Supplementary Data 17** column names were fixed.

Reviewer 3 comment 12: Also Figure 4F seems to be lacking a legend description.

Response 12: Figure 4F was dropped from the manuscript in the most recent iteration; in our response to reviews, the older version of the figure was crossed off with the tracked changes function, but perhaps the reviewer accidentally commented on this older figure? We have confirmed that the figure 4 caption includes a description for Figure 4a-e.

Reviewer #4 (Remarks to the Author):

In this study, Gordevicius et al. identified the epigenetic alterations in the autophagy-lysosome pathway (ALP) genes (predominantly in the promoter region) in the Parkinson's disease appendix that lead to transcriptional changes validated by RNAseq and proteomic analyses. Such alterations are largely recapitulated in PD and aging brains, and in mice with alpha-synuclein overexpression and induced gut inflammation. The amount of work is very impressive. However, I have several major and minor concerns listed below.

Response: We thank the reviewer for their enthusiasm for our study and the time taken to review and comment.

Reviewer 4, comment 1: Given the massive epigenetic dysregulation found in the appendix and the neurons in the cortex/olfactory bulb between PD patients and controls, have the authors examined the enzymes related to DNA methylation (e.g., TETs and DNMTs)? Are there any significant difference in the protein or the methylation levels of these genes between PD and control?

Response 1: This is an interesting question, and we appreciate the opportunity to examine this aspect of our results in more detail. Our findings with methylation padlock assay do not necessarily indicate genome-wide dysregulation. This is illustrated in supplementary figure S4 where differentially methylated cytosines are most likely to be found within 20 kbp away from Lysosome gene starts. The methylation assay included TET2 and TET3 genes which were not differentially methylated in the appendix. RNA-seq of appendix tissue did not indicate differential expression of either TET nor DNMT genes. On the other hand, TET3 was differentially methylated in the olfactory bulb and prefrontal cortex replication cohort but not the primary cohort. None of these genes were captured by proteomics analysis. Thus we do not have clear evidence of a widespread dysregulation that is being driven by differences in these enzymes.

Reviewer 4, comment 2: Previous work (<https://www.nature.com/articles/nn.3607>) has demonstrated the similarity and differences between CpG and CpH methylations in adult mammalian brains. Are these two classes of methylations differentially regulated in PD and aging?

Response 2: We appreciate the suggestion to parse out our data in more detail. As the reviewer notes, in isolated neurons, DNA methylation occurs at both CpG and CpH (i.e., CpA, CpT, CpC) locations [36], and thus in neurons of the prefrontal cortex, we investigated a total of 130,733 CpG and 696,665 CpH sites at ALP genes. There was a clear hypermethylation trend among CT and CA dinucleotides (OR = 5.01, $p = 0.005$, OR = 13.41, $p = 0.0004$, respectively; Fisher's exact test). We made the following modification to the manuscript:

Results, pg 6. In prefrontal cortex neurons, we observed 70 differentially methylated sites affecting 58 genes in PD (1.2 differentially methylated sites per affected ALP gene with average methylation change 8% in CpG and 6% in CpH sites; $q < 0.05$, robust linear regression; Fig 2a; Supplementary Data 4), which again were mostly hypermethylated (OR = 2.41, $p = 7.13 \times 10^{-4}$, Fisher's exact test; Fig. 2b, PFC). Specifically, we observed strong hypermethylation trend among CT and CA dinucleotides (OR = 5.01, $p = 0.005$, OR = 13.41, $p = 0.0004$, respectively; Fisher's exact test).

Reviewer 4, comment 3: In Fig. 5b, the concordance is only significant between PD appendix and A30P_water_WT_water. Does this indicate that DSS-induced gut inflammation and rAAV synuclein overexpression didn't cause PD-like methylation changes? Since A30P is a whole-body overexpression model, does it indicate a bidirectional influence from the brain as local overexpression by rAAV did not recapitulate the the methylation changes in PD appendix?

Response 3: The reviewer makes a valid observation that only methylation changes in the A30P overexpression model show significant concordance with those seen in the PD appendix. We agree that this may suggest the importance of whole body changes induced by a-syn (including possible bidirectional influence from the brain) to impact the methylation changes we see in PD. However, we also show in Fig 5b that there is significant correlation between rAAV and A30P induced changes in methylation, showing that a local a-syn accumulation does have a similar effect to some extent. Thus we suggest that the lack of such correlation with human appendix in the other models is more likely due to differences between human and mouse.

Reviewer 4, comment 4: The number of ALP genes was inconsistent. In line 101, "521 genes reported in publicly available human autophagy and lysosomal [40] databases as well as PD risk genes", while line 421, " We fine-mapped DNA methylation changes at 571 ALP genes in the mouse cecal patch".

Response 4: We thank the reviewer for the attention to detail. There is a different number of genes known to be associated with autophagy in human and mouse, resulting in a different number of ALP genes assessed in the respective species.

Reviewer 4, comment 5: In Fig 6, environmental factors should be included in the proposed model.

Response 5: In the manuscript we demonstrate molecular dysregulation of the ALP system , making it a potential culprit for PD initiation and progression as shown in Fig. 6. The causative factors of such dysregulation remain unclear and could be addressed in future work. Please

note the inclusion of environmental factors addressed at the end of the following paragraph in the Discussion, pg. 14:

The causative factors of the epigenetic dysregulation of the ALP in PD remain unclear, although there is evidence for a bidirectional relationship between α -syn and the ALP [19, 21]. Decreased autophagic flux results in an accumulation of α -syn [19, 21], and misfolded α -syn itself appears to play an active role in suppressing the ALP [83, 84]. In this way, genetic and/or epigenetic defects in the ALP leading to α -syn accumulation could lead to further (epigenetic) dysregulation of the ALP. This is supported by our finding that the same ALP genes are disrupted in a mouse model with α -syn overexpression as in the human PD appendix. In addition to the joint contribution of genetic risk factors [23, 25] and α -syn accumulation triggering epigenetic disruption of the ALP, environmental agents [93] and abnormal shifts in the microbiome [94] may play a role, especially because they can impact gut inflammation.

We also made the following adjustment to the caption of Figure 6 (marked in italic):

Fig 6. Proposed model of ALP changes in the PD appendix and brain. Model based on our study and the literature [14, 95] illustrating the interplay between the ALP, aging, inflammation, and α -syn aggregates, and their contribution to the development and progression of PD. The healthily functioning ALP is responsible for the breakdown of physiological and aggregated α -syn [14]. In PD, widespread epigenetic silencing of ALP genes leads to decreased lysosomal functioning. This promotes an accumulation of α -syn aggregates, which reciprocally furthers ALP dysfunction in PD. In aging, there is an epigenetic inactivation of macroautophagy and selective autophagy genes, with concomitant decline in ALP activity, which places individuals of advanced age at greater risk of developing PD. The ALP also moderates inflammatory responses [95]. PD patients may exhibit heightened responses to inflammation as result of α -syn accumulation and ALP dysregulation. Loss of ALP function in PD also enables the secretion and cell-to-cell transfer of aggregated α -syn [85]. Hence, epigenetic disruption of the ALP in the gut and brain may contribute to the development and progression of α -syn pathology. *While the causative factors of the epigenetic dysregulation of the ALP in PD remain unclear, joint contribution of genetic risk factors [23, 25] and α -syn accumulation triggering epigenetic disruption of the ALP, environmental agents [93] and abnormal shifts in the microbiome [94] may play a role, especially because they can impact gut inflammation.* The appendix, a potential initiation site for synucleinopathy in idiopathic PD, is circled in red. Red arrows indicate direction of change in PD relative to controls. Dotted lines indicate interactions that are weakened by the epigenetic dysregulation of the ALP in PD.

Reviewer 4, comment 6: Fig 2c, it would be more informative if the hypermethylation and hypomethylation are presented separately.

Response 6: We thank the reviewer for the suggestion to add more detail. We added a supplementary figure in which the analysis is stratified by the direction of methylation change.

Figure S19. Genomic elements exhibiting similar changes in DNA methylation in the PD brain and PD appendix stratified by the direction of methylation change. For each category, the overlap of genomic elements with differentially methylated cytosines between PD appendix and brain datasets was determined. Poised and active enhancers, and active promoters were identified using appendix ChIP-seq (n = 3 individuals each for H3K27ac and H3K4me1) and prefrontal cortex (n = 9 individuals, PsychENCODE data). Filled circles represent *p < 0.05, **p < 0.01, and ***p < 0.001, Fisher's exact test examining overlap with PD appendix. Error bars indicate 95% confidence intervals.

Reviewer #5 (Remarks to the Author):

This is a very interesting manuscript that can have a major impact on PD research. It can change our views on how PD begins and develops. However, there are several issues require more work and revisions, as detailed in the numbered list below:

Response: We thank the reviewer for their enthusiasm for our study and the time taken to review and comment.

Majors:

Reviewer 5, comment 1: As the authors said, “Our analysis of differential methylation in the PD appendix was controlled for sample age, sex, postmortem.” However, the statistical analysis of demographic information between the different groups was not shown, such as sex, age, Hoehn-Yahr stage.

Response 1: We thank the reviewer for the observation which will help to improve the manuscript. In our statistical analysis, we fitted a binomial regression model with group as the dependent variable and the study covariates as the independent covariates. We have amended the Supplementary Data 18 with an additional sheet detailing the relationship of each covariate and the sample group for each study of human samples. While Hoehn-Yahr stage was not available, tissue samples had evident brain Lewy pathology (PD Braak stages III-VI) as described in Supplementary data 15.

Changes made to manuscript: Added sheet to Supplementary Data 18.

Reviewer 5, comment 2: I don't quite understand why the authors choose prefrontal cortex in this study. In the Introduction section, they explained “We then identify whether similar changes are mirrored in neurons of the prefrontal cortex and the olfactory bulb, another proposed starting point for PD [41, 42].” However, in these two references, the involvement of prefrontal cortex pathology was shown at stages 5 and 6, which at the later phase of PD.

Response 2: We selected prefrontal cortex neurons because of their relevance to PD, but most importantly, because prefrontal cortex neurons still exist in the postmortem PD brain. In contrast, substantia nigra neurons have largely degenerated and therefore are no longer present for isolation⁵. On a technical level, neuronal nuclei isolation has been fully optimized for the prefrontal cortex (as shown by our publications^{1,2}), and the prefrontal cortex yields sufficient numbers of neurons for DNA methylation analysis^{1,2}. Epigenetic (DNA methylation) changes in prefrontal cortex neurons can occur early in neurodegenerative diseases^{1,3,4}. Hence, for our study of molecular changes in PD brain neurons, the prefrontal cortex offers a disease-relevant and available source of neurons, in combination with technical feasibility.

We also made the following clarification in the manuscript:

Introduction, pg. 3. We then identify whether similar changes are mirrored in neurons of the prefrontal cortex, a region affected in later disease stages, and the olfactory bulb, another proposed starting point for PD [41, 42]

References:

1. Li, P. et al. Epigenetic dysregulation of enhancers in neurons is associated with Alzheimer's disease pathology and cognitive symptoms. *Nat Commun* **10**, 2246 (2019).
2. Pai, S. et al. Differential methylation of enhancer at IGF2 is associated with abnormal dopamine synthesis in major psychosis. *Nat Commun* **10**, 2046 (2019).

3. De Jager, P.L. et al. Alzheimer's disease: early alterations in brain DNA methylation at ANK1, BIN1, RHBDF2 and other loci. *Nat Neurosci* **17**, 1156-63 (2014).
4. Lunnon, K. et al. Methylomic profiling implicates cortical deregulation of ANK1 in Alzheimer's disease. *Nat Neurosci* **17**, 1164-70 (2014).
5. Kordower JH, Olanow CW, Dodiya HB, Chu Y, Beach TG, Adler CH, Halliday GM, Bartus RT. Disease duration and the integrity of the nigrostriatal system in Parkinson's disease. *Brain*. 2013 Aug;136(Pt 8):2419-31. doi: 10.1093/brain/awt192. PMID: 23884810.

Reviewer 5, comment 3: In the manuscript, the authors chose appendix, olfactory bulb, and prefrontal cortex as the study regions. The results are good enough, but I also want to know, when the methylation alteration was appeared in these regions, what are the changes of methylation levels in the substantia nigra and striatum, the most important pathological areas in PD.

Response 3: In **Response 2** we mentioned why substantia nigra could not be interrogated and point to a few studies that have shown that methylation changes occur early in disease. However, we agree with the reviewer that it would be interesting to investigate longitudinal methylation profiles in various brain regions. We believe that this study lays good groundwork for further investigation.

Reviewer 5, comment 4: In line 401-402, page 9, the author said, "Together, this suggests that in patients, PD disease processes disrupt normal aging changes in ALP function." Is it possible that the change in ALP function causes the occurrence of PD, rather than the consequences of PD, what do the authors think about this question?

Response 4: Our study investigates correlative rather than causative relationship of ALP and PD. We agree that this sentence inadvertently implies causation. We have amended the sentence to avoid confusion.

Results, pg. 10: Taken together, this suggests that in patients, PD disease processes are associated with disrupted normal aging changes in ALP function.

Reviewer 5, comment 5: In vivo, the authors demonstrated the methylation alteration at ALP genes in animal models, but have the authors detected the changes of autophagy and lysosomal markers, such as LC3, p62, and Beclin-1 to confirm whether there is a dysfunction of autophagy flow? And how do the authors confirm the causal relationship between methylation alteration and dysfunction of autophagy flow?

Response 5: The reviewer raises an important question. It is important to point out that similar analyses have been performed. For example, it has been shown that α -syn over expression leads to decreased autophagy flux in neurons, including in the A30P mouse model used in our study^{1,2}. The analysis presented in our manuscript revealed that α -syn overexpression in the A30P mouse model recapitulated the epigenetic abnormalities in lysosome function in the PD

appendix and affected autophagy (OR = 2.63, p = 0.001 and OR = 1.85, p = 0.04, respectively, Fisher's exact test; Supplementary Figure S15). Accordingly, we believe it can be inferred that these epigenetic changes correlate with changes in autophagy and lysosomal function, akin to what is described in the above referenced manuscripts.

While further data on autophagy and lysosomal markers in our study would be valuable to more firmly establish a causal relationship between methylation and decreased autophagy flux, it would be difficult to generate it for two main reasons. First, there is no enteric tissue left from the initial subjects, therefore, new animals would have to be generated and new DSS treatment experiments have to be performed, a time-consuming process which would be further delayed due to Covid restrictions. Second, the lab led by Dr. Labrie has disbanded after her unexpected death, accordingly there are no resources to run such experiments.

References:

1. Lei Z, Cao G, Wei G. A30P mutant α -synuclein impairs autophagic flux by inactivating JNK signaling to enhance ZKSCAN3 activity in midbrain dopaminergic neurons. *Cell Death Dis.* 2019 Feb 12;10(2):133. doi: 10.1038/s41419-019-1364-0. PMID: 30755581.
2. Pupyshev AB, Korolenko TA, Akopyan AA, Amstislavskaya TG, Tikhonova MA. Suppression of autophagy in the brain of transgenic mice with overexpression of A53T-mutant α -synuclein as an early event at synucleinopathy progression. *Neurosci Lett.* 2018 Apr 13;672:140-144. doi: 10.1016/j.neulet.2017.12.001. Epub 2017 Dec 2. PMID: 29203207.

Reviewer 5, comment 6: In the manuscript, the authors want to explore the underlying mechanism of the transfer of α -syn aggregates from appendix to brain in PD. However, in vivo study only showed in situ changes, what about the α -syn pathology changes in brain? How do you prove this animal model is successful?

Response 6: The reviewer brings up an important question of whether the α -syn pathology would be present in the brain in our in vivo model. Transfer of α -syn pathology from the enteric nervous system to the CNS has been observed in a variety of α -syn models. We have done a thorough investigation into this before¹, and the data clearly showed (both in rats and non human primates) that regardless of injection site, that you at most get transient CNS pathology in the absence of additional drivers (e.g. in M83 overexpressing animals where pathology is maintained and potentially propagated). Similar findings have also been published by other groups^{2,3}. In a paper by Grathwohl et al.⁴ that is on bioRxiv, and which is currently in revision for publication, our collaborators show that α -syn pathology develops in the brain 18 months post colitis (Fig 5 and 6) but not at 6 months post colitis. However, we feel that the important point is that the data shows that α -syn pathology can induce local ALP changes in the appendix, and that these could subserve additional "hits/insults". In other words, such early changes could represent the initial seed of α -syn pathology. Further experimentation to directly test this hypothesis is underway in our various laboratories, but we feel that this is beyond the scope of this manuscript.

References:

1. Manfredsson FP, Luk KC, Benskey MJ, Gezer A, Garcia J, Kuhn NC, Sandoval IM, Patterson JR, O'Mara A, Yonkers R, Kordower JH. Induction of alpha-synuclein pathology in the enteric nervous system of the rat and non-human primate results in gastrointestinal dysmotility and transient CNS pathology. *Neurobiol Dis.* 2018 Apr;112:106-118. doi: 10.1016/j.nbd.2018.01.008. Epub 2018 Jan 16. PMID: 29341898.
2. Uemura N, Yagi H, Uemura MT, Yamakado H, Takahashi R. Limited spread of pathology within the brainstem of α -synuclein BAC transgenic mice inoculated with preformed fibrils into the gastrointestinal tract. *Neurosci Lett.* 2020 Jan 18;716:134651. doi: 10.1016/j.neulet.2019.134651. Epub 2019 Nov 26. PMID: 31783082.
3. Uemura N, Yagi H, Uemura MT, Hatanaka Y, Yamakado H, Takahashi R. Inoculation of α -synuclein preformed fibrils into the mouse gastrointestinal tract induces Lewy body-like aggregates in the brainstem via the vagus nerve. *Mol Neurodegener.* 2018 May 11;13(1):21. doi: 10.1186/s13024-018-0257-5. Erratum in: *Mol Neurodegener.* 2019 Jul 26;14(1):31. PMID: 29751824.
4. Stefan Grathwohl et al. Experimental colitis drives enteric alpha-synuclein accumulation and Parkinson-like brain pathology. *bioRxiv* 505164; doi: <https://doi.org/10.1101/505164> (<https://www.biorxiv.org/content/10.1101/505164v4>)

Reviewer 5, comment 7: In the Methods section, the authors said, “Within each data set all measurements were taken from distinct samples.” But as the authors showed, methylation alteration of ALP can occur in areas such as the appendix or olfactory bulb at early stage of PD, but where do the changes in the same population weigh more and where do they originate? How do you do the comparison if not choose the same population?

Response 7: The intention of the sentence was to explain that technical replicate measurements were combined into one biological measurement within each dataset. The question of the reviewer is very important nonetheless.

First, while we show methylation changes in all tissues, it is difficult to know where these changes result in stronger phenotypic effects. With the data at hand we cannot claim that methylation changes in the appendix matter more than methylation changes of the same magnitude in olfactory bulb. An entirely different experimental setup would be necessary to investigate such a hypothesis. However, it is known from earlier epidemiological studies that suggest the link between early removal of appendix and late onset of PD^{1,2}. Thus, in our work we propose that the ALP system in the appendix is a potential early stage initiator of Parkinson's disease.

Second, it was not technically feasible to collect appendix and brain samples from the same individuals as the latter were obtained post mortem and from multiple different tissue repositories. We have made every effort to make sure that the samples are drawn from demographically comparable populations which would allow us to make statistically meaningful inferences about the whole population.

References:

1. Killinger BA, Madaj Z, Sikora JW, Rey N, Haas AJ, Vepa Y, Lindqvist D, Chen H, Thomas PM, Brundin P, Brundin L, Labrie V. The vermiform appendix impacts the risk of developing Parkinson's disease. *Sci Transl Med*. 2018 Oct 31;10(465):eaar5280. doi: 10.1126/scitranslmed.aar5280. PMID: 30381408.
 2. Mendes A, Gonçalves A, Vila-Chã N, Moreira I, Fernandes J, Damásio J, Teixeira-Pinto A, Taipa R, Lima AB, Cavaco S. Appendectomy may delay Parkinson's disease Onset. *Mov Disord*. 2015 Sep;30(10):1404-7. doi: 10.1002/mds.26311. Epub 2015 Jul 30. PMID: 26228745.
-

Minors:

Reviewer 5, comment 8: The running title is “Widespread silencing of autophagy–lysosomal genes in the Parkinson’s disease gut and brain”, but the title is “Epigenetic inactivation of the autophagy–lysosomal system in the Parkinson’s disease appendix”. What do you want to emphasize in this manuscript? The different statements make me feel confused.

Response 8: Thank you for the observation, we have amended the running title as follows:

TT: Epigenetic inactivation of the autophagy–lysosomal system in the Parkinson’s disease appendix

RT: Epigenetic inactivation of the autophagy–lysosomal system in PD appendix

Reviewer 5, comment 9: The authors demonstrated silencing of autophagy–lysosomal genes in PD appendix and brain, what about other α -synucleinopathies, including dementia with Lewy bodies and multiple system atrophy, etc.? Whether the same changes appeared in these diseases?

Response 9: We appreciate the suggestion that epigenetic silencing may also play a role in other synucleinopathies. In this paper we focused on PD. Other α -synucleinopathies, though outside the scope of this study, would certainly be interesting to investigate in future work.

Reviewer 5, comment 10: In vivo study, why the authors choose rAAV vector that overexpressed human α -syn instead of mouse α -syn?

Response 10: The AAV synucleinopathy model exclusively uses human or mutant α -syn as we¹ and others² have shown that overexpression of rat α -syn, for whatever reason, does not result in the same toxicity as human or mutant forms.

References:

1. Gorbatyuk OS, Li S, Nash K, Gorbatyuk M, Lewin AS, Sullivan LF, Mandel RJ, Chen W, Meyers C, Manfredsson FP, Muzyczka N. In vivo RNAi-mediated alpha-synuclein silencing induces nigrostriatal degeneration. *Mol Ther*. 2010 Aug;18(8):1450-7. doi: 10.1038/mt.2010.115. Epub 2010 Jun 15. PMID: 20551914.

2. Polinski NK, Manfredsson FP, Benskey MJ, Fischer DL, Kemp CJ, Steece-Collier K, Sandoval IM, Paumier KL, Sortwell CE. Impact of age and vector construct on striatal and nigral transgene expression. *Mol Ther Methods Clin Dev.* 2016 Dec 7;3:16082. doi: 10.1038/mtm.2016.82. PMID: 27933309.

Reviewer 5, comment 11: In vivo study, why the authors choose mice with different ages in gut inflammation model (12 weeks old) and vector-mediated α -syn overexpression model (8 weeks old)?

Response 11: We aimed to perform the experimentation on the same age-range of animals. The ages used in these assays would all fall under the range of young adult. There is no differential in terms of the effects of α -syn overexpression or inflammatory readouts unless the subjects are aged^{1,2}.

References:

1. Fischer DL, Gombash SE, Kemp CJ, Manfredsson FP, Polinski NK, Duffy MF, Sortwell CE. Viral Vector-Based Modeling of Neurodegenerative Disorders: Parkinson's Disease. *Methods Mol Biol.* 2016;1382:367-82. doi: 10.1007/978-1-4939-3271-9_26. PMID: 26611600.
2. Polinski NK, Manfredsson FP, Benskey MJ, Fischer DL, Kemp CJ, Steece-Collier K, Sandoval IM, Paumier KL, Sortwell CE. Impact of age and vector construct on striatal and nigral transgene expression. *Mol Ther Methods Clin Dev.* 2016 Dec 7;3:16082. doi: 10.1038/mtm.2016.82. PMID: 27933309.

Reviewer 5, comment 12: The schematic diagram in Fig.6 is somewhat confusing. The authors should make it more clearly and shows the relationship between the appendix and brain.

Response 12: As per reviewer's suggestion we have amended the Fig. 6 to indicate the proposed relationship between the appendix and brain.

Reviewer 5, comment 13: In Figure S13B, the distribution of aggregated α -syn in enteric neurons is not displayed clearly, the nucleus and axons should also be co-stained with phosphor-serine 129.

Response 13: We thank the reviewer for the attention to detail. While we do not have a different figure, the intended purpose of Figure S13B is to show that rAAV-mediated human α -syn overexpression of the mouse cecal patch results in α -syn aggregation. The issue has been addressed in more detail in our previous as well as other work^{1,2}.

References:

1. Manfredsson FP, Luk KC, Benskey MJ, Gezer A, Garcia J, Kuhn NC, Sandoval IM, Patterson JR, O'Mara A, Yonkers R, Kordower JH. Induction of alpha-synuclein pathology in the enteric nervous system of the rat and non-human primate results in gastrointestinal dysmotility and transient CNS pathology. *Neurobiol Dis.* 2018 Apr;112:106-118. doi: 10.1016/j.nbd.2018.01.008. Epub 2018 Jan 16. PMID: 29341898
2. Uemura N, Yagi H, Uemura MT, Hatanaka Y, Yamakado H, Takahashi R. Inoculation of α -synuclein preformed fibrils into the mouse gastrointestinal tract induces Lewy body-like aggregates in the brainstem via the vagus nerve. *Mol Neurodegener.* 2018 May 11;13(1):21. doi: 10.1186/s13024-018-0257-5. Erratum in: *Mol Neurodegener.* 2019 Jul 26;14(1):31. PMID: 29751824

Reviewer 5, comment 14: In line 83-86, page 2, "Taken together, the apparent relationships between PD, the ALP, and the development and spread of α -syn pathology suggest that disruption of the ALP in the aging appendix could be an important mechanism underlying the

transfer of α -syn aggregates from appendix to brain in PD.” It is not appropriate to use “between” for comparison of three subjects.

Response 14: We have replaced *between* with *among*:

Introduction, pg. 3: Taken together, the apparent relationships among PD, the ALP, and the development and spread of α -syn pathology suggest that disruption of the ALP in the aging appendix could be an important mechanism underlying the transfer of α -syn aggregates from appendix to brain in PD.

Reviewer 5, comment 15: In line 169, page 4, “OR = 2.12, p = 3.67 x 10⁻⁸”, the “x” should change to “×”, the same mistake was repeated in later pages.

Response 15: We fixed the manuscript accordingly.

Reviewer 5, comment 16: In line 263, page 6, “genome-wide analysis [53]].”, the later “]” should be deleted.

Response 16: We deleted this extra bracket.

Reviewer 5, comment 17: In line 404, page 9, “ α -Syn accumulation”, the “ α -Syn” should change to “ α -syn”, the same mistake was repeated in later pages.

Response 17: . We appreciate the reviewers attention to detail. Here “Syn” was capitalized since it was the first word in a sentence. We have changed this per the reviewer’s request and leave this to the editor’s discretion. We also checked for consistency in using “ α -syn” in the rest of the paper.

REVIEWER COMMENTS:

Reviewer #4 (Remarks to the Author):

The authors have addressed all the concerns in the revised manuscript.

Reviewer #5 (Remarks to the Author):

Response :

Thanks for making the modification according to my suggestion.

This manuscript has improved since the last revision. However, the authors did not sufficiently address the reviewers' comments because of some objective reasons. To make the readers understand your experiments better, I suggest the authors address these reasons in the section of "Discussion", for example, why don't you choose substantia nigra in this study?

REVIEWERS' COMMENTS

Reviewer #4 (Remarks to the Author):

The authors have addressed all the concerns in the revised manuscript.

Response: We thank the reviewer for their time taken to review and comment which led to improved quality of the manuscript.

Reviewer #5 (Remarks to the Author):

Thanks for making the modification according to my suggestion.

This manuscript has improved since the last revision. However, the authors did not sufficiently address the reviewers' comments because of some objective reasons. To make the readers understand your experiments better, I suggest the authors address these reasons in the section of "Discussion", for example, why don't you choose substantia nigra in this study?

Response: We thank the reviewer. We have made the following additions to the manuscript:

Discussion, pg 14: For this study we selected prefrontal cortex neurons because of their relevance in later stages of PD and because prefrontal cortex neurons still exist in the analyzed early stage (Braak stage 3-4) postmortem PD brain. In contrast, substantia nigra neurons have largely degenerated and therefore are insufficiently present or are too advanced in the degenerative process for isolation [96]. On a technical level, neuronal nuclei isolation has been fully optimized for the prefrontal cortex, and the prefrontal cortex yields sufficient numbers of neurons for DNA methylation analysis [97, 98]. Epigenetic (DNA methylation) changes in prefrontal cortex neurons can occur early in neurodegenerative diseases [97-99]. Hence, for our study of molecular changes in the early PD brain neurons, the prefrontal cortex offers a disease-relevant and available source of neurons, in combination with technical feasibility.

Discussion, pg. 13: Transfer of α -syn pathology from the enteric nervous system to the CNS has been observed in a variety of α -syn models [72, 90, 91]. While it is unclear, what triggers the enteric α -syn pathology and its propagation to the brain, recent studies in α -syn transgenic mice demonstrate that α -syn pathology develops in the brain 18 months post an experimental form of colitis but not at 6 months post colitis which was accompanied by dopaminergic neuronal loss in the substantia nigra [92]. In humans, colitis and the prodromal appearance of enteric α -syn pathology has also been implicated as risk factor for PD as well as several genes related to immune function [92]. We previously reported that there can be an abundance of aggregated α -syn in both the healthy and PD appendix, although α -syn levels are up to three times greater in the PD appendix [12]. In combination with epigenetic perturbation of lysosomal function, hypomethylation of the α -syn gene in the PD appendix may propel α -syn pathology. Indeed, studies in the brain have found that endogenous α -syn levels influence the spread of synucleinopathy [90]. Thus, epigenetic

changes in the PD appendix are consistent with an increased production and impaired clearance of α -syn pathology.